# Drop-Muon: Update Less, Converge Faster

## Abstract

Conventional wisdom in deep learning optimization dictates updating all layers at every step–a principle followed by all recent state-of-the-art optimizers such as Muon. In this work, we challenge this assumption, showing that full-network updates can be fundamentally suboptimal, both in theory and in practice. We introduce a non-Euclidean Randomized Progressive Training method—Drop-Muon—a simple yet powerful framework that updates only a subset of layers per step according to a randomized schedule, combining the efficiency of progressive training with layer-specific non-Euclidean updates for top-tier performance. We provide rigorous convergence guarantees under both layer-wise smoothness and layer-wise $(L^0, L^1)$-smoothness, covering deterministic and stochastic gradient settings, marking the first such results for progressive training in the stochastic and non-smooth regime. Our cost analysis further reveals that full-network updates are not optimal unless a very specific relationship between layer smoothness constants holds. Through experiments on `NanoGPT`-124M, a `ResNet50` image-classification task, and small controlled CNN studies, we empirically demonstrate that Drop-Muon consistently outperforms full-network Muon, achieving the same accuracy up to $1.4\times$ faster in wall-clock time. Together, our results suggest a shift in how large-scale models can be efficiently trained, challenging the status quo and offering a highly efficient, theoretically grounded alternative to full-network updates.

## 1 Introduction

Since their debut, Adam and related methods (Kingma & Ba, 2015; Loshchilov & Hutter, 2019) have dominated deep learning optimization. Yet, the field is now at an inflection point. Recent advances highlight a new generation of algorithms designed to better capture the geometry of modern models, with Muon (Jordan et al., 2024) and its successors—Scion (Pethick et al., 2025) and Gluon (Riabinin et al., 2025b)—emerging as promising alternatives. Fueled by state-of-the-art performance in large language models (LLMs) training (Liu et al., 2025; Shah et al., 2025; Thérien et al., 2025; Moonshot AI, 2025; Wen et al., 2025) and emerging theoretical developments (Pethick et al., 2025; Kovalev, 2025; Riabinin et al., 2025b), these methods are on track to disrupt entrenched practices.

Central to their design are the layer-specific *linear minimization oracles* (LMOs) over non-Euclidean norm balls, enabling better alignment with the highly anisotropic loss landscapes of neural networks. Concretely, let $X = [X_1, \ldots, X_b]$ denote the parameters of a $b$-layer model, with $X_i$ indexing the parameters of layer $i \in [b] := \{1, \ldots, b\}$. Each of the aforementioned optimizers can be viewed as an instance of the general update rule

$$X_i^{k+1} = X_i^k + \text{LMO}_{\mathcal{B}_i(0, t_i^k)}(M_i^k), \qquad i \in \{1, \ldots, b\} \tag{1}$$

(see Section A.1). Here, $\mathcal{B}_i(X_i, t_i) := \{Z_i \in \mathcal{X}_i : \|X_i - Z_i\|_{(i)} \leq t_i\}$ is a ball of radius $t_i$ centered at $X_i$ in the vector space $\mathcal{X}_i$, where $\|\cdot\|_{(i)}$ is a norm chosen for the $i$th layer, $M_i^k$ is a momentum term, and the linear minimization oracle is defined as $\text{LMO}_{\mathcal{B}_i(X_i, t_i)}(M_i) := \arg\min_{Z_i \in \mathcal{B}_i(X_i, t_i)} \langle M_i, Z_i \rangle$. Different choices of $\|\cdot\|_{(i)}$ produce different algorithms (for example, Muon uses the spectral norm for hidden layers). Crucially, however, all these updates share a common characteristic: all layers are updated at every iteration. In this work, we question this design choice. Our central hypothesis is simple yet fundamental:

*Updating the entire network at every step may **not** be optimal.*

---

**Algorithm 1** Drop-Muon

---

1: **Input:** initial iterate $X^0 = [X_1^0, \ldots, X_b^0] \in \mathcal{X}$; momentum $M^0 = [M_1^0, \ldots, M_b^0] \in \mathcal{X}$; stepsizes $\gamma_i^k > 0$; momentum parameters $\beta^k \in [0, 1)$
2: **for** $k = 0, \ldots, K - 1$ **do**
3:     Sample $\xi^k \sim \mathcal{P}$ and the set of *active* layers $S^k \sim \mathcal{D}$
4:     **for** $i \notin S^k$ **do**                    ▷ Freeze layers not selected as *active*
5:         $M_i^k = M_i^{k-1}$
6:         $X_i^{k+1} = X_i^k$
7:     **end for**
8:     **for** $i \in S^k$ **do**                    ▷ Update *active* layers
9:         Update momentum $M_i^k = (1 - \beta_i)M_i^{k-1} + \beta_i \nabla_i f(X^k; \xi^k)$
10:        Update parameters via

$$X_i^{k+1} = \text{LMO}_{\mathcal{B}(X_i^k, t_i^k)}(M_i^k) = X_i^k - \gamma_i^k \left(M_i^k\right)^\sharp \tag{3}$$

11:    **end for**
12: **end for**

---

In the remainder of this paper, we demonstrate that the default practice of full-network updates is indeed not universally the best choice, from both *theoretical* and *practical* points of view, calling into question a core principle of standard training protocols.

### 1.1 BACKGROUND

Let us begin by formalizing the setup. We consider the optimization problem

$$\min_{X \in \mathcal{X}} \{f(X) := \mathbb{E}_{\xi \sim \mathcal{P}}[f(X; \xi)]\}, \tag{2}$$

where $X \in \mathcal{X}$ represents the collection of trainable parameters of a neural network. Specifically, $X$ is composed of block variables $X_i \in \mathcal{X}_i := \mathbb{R}^{m_i \times n_i}$ corresponding to layer $i \in [b]$; we write $X = [X_1, \ldots, X_b]$. In this context, $\mathcal{X}$ is the $d$-dimensional product space

$$\mathcal{X} := \bigotimes_{i=1}^b \mathcal{X}_i \equiv \mathcal{X}_1 \otimes \ldots \otimes \mathcal{X}_b,$$

where $d := \sum_{i=1}^b m_i n_i$. Each function $f(\cdot; \xi) : \mathcal{X} \to \mathbb{R}$ is continuously differentiable, potentially nonconvex and non-smooth, and represents the loss of the model evaluated at a data point $\xi$ sampled from the probability distribution $\mathcal{P}$. We denote by $\nabla_i f(X) \in \mathcal{X}_i$ the gradient component corresponding to the $i$th layer, so that $\nabla f(X) = [\nabla_1 f(X), \ldots, \nabla_b f(X)] \in \mathcal{X}$. Each space $\mathcal{X}_i$ is equipped with the trace inner product, defined as $\langle X_i, Y_i \rangle_{(i)} := \text{tr}(X_i^\top Y_i)$ for $X_i, Y_i \in \mathcal{X}_i$, which induces the standard Euclidean norm, denoted by $\|\cdot\|_2$. In addition, each space is endowed with an arbitrary norm $\|\cdot\|_{(i)}$ (which need not be induced by this inner product). We let $\|\cdot\|_{(i)\star}$ be the dual norm associated with $\|\cdot\|_{(i)}$ (i.e., $\|X_i\|_{(i)\star} := \sup_{\|Z_i\|_{(i)} \leq 1} \langle X_i, Z_i \rangle_{(i)}$ for any $X_i \in \mathcal{X}_i$).

## 2 THE ALGORITHM

Before summarizing our main contributions (see the end of Section 3), we dive directly into presenting our method. Motivated by the considerations in Section 1, we propose Drop-Muon (Algorithm 1)–a non-Euclidean layer-wise optimizer for deep learning based on the idea of sub-network training. At each iteration $k$, instead of updating the entire network as in standard Muon, Drop-Muon samples a random subset $S^k \subseteq [b]$ of layers according to a user-defined distribution $\mathcal{D}$ and updates only the parameters of layers in $S^k$, keeping all other layers frozen. As the reader may have noticed, the main LMO update step (3) admits two equivalent formulations. The alternative representation of (1) uses the *sharp operator* (Nesterov, 2012; Kelner et al., 2014), defined for any $M \in \mathcal{X}$ by $M^\sharp := \arg\max_{X \in \mathcal{X}} \{\langle M, X \rangle - \frac{1}{2}\|X\|^2\}$. It is well known that $M^\sharp$ relates to the LMO via $M^\sharp = -\|M\|_\star \text{LMO}_{\mathcal{B}(0,1)}(M)$, and hence

$$X_i^{k+1} = X_i^k + t_i^k \text{LMO}_{\mathcal{B}_i(0,1)}(M_i^k) = X_i^k - \frac{t_i^k}{\|M_i^k\|_{(i)\star}}\left(M_i^k\right)^\sharp, \tag{4}$$

---

**Algorithm 2** Drop-Muon (deterministic gradient variant)

---

1: **Input:** initial iterate $X^0 = [X_1^0, \ldots, X_b^0] \in \mathcal{X}$; stepsizes $\gamma_i^k > 0$, $i \in [b]$, $k \geq 0$
2: **for** $k = 0, \ldots, K - 1$ **do**
3:     Sample $S^k \sim \mathcal{D}$
4:     Freeze the layers $\{X_i\}_{i \notin S^k}$ ($X_i^{k+1} = X_i^k$ for $i \notin S^k$)
5:     Update the layers $\{X_i\}_{i \in S^k}$ via

$$X_i^{k+1} = X_i^k - \gamma_i^k \left( \nabla_i f(X^k) \right)^\sharp \tag{5}$$

6: **end for**

---

which corresponds to a layer-wise normalized *steepest descent* step with stepsize $\gamma_i^k := t_i^k / \|M_i^k\|_\star$. When $\|\cdot\|_{(i)} = \|\cdot\|_2$ is the standard Euclidean norm, the sharp operator reduces to the identity mapping, so that $M^\sharp = M$, and the update coincides with Stochastic Gradient Descent with momentum (SGDM) (Cutkosky & Mehta, 2020), though here performed layer-wise. These equivalent formulations will be repeatedly invoked in the proofs of the results from Sections 4.1 and 4.2.

## 3 COST MODEL

Our key theoretical contribution is that strategically skipping some layer updates may lead to performance gains. To isolate the core phenomenon, we first study a simplified variant of the method without stochasticity in the gradients and momentum, described in Algorithm 2. This deterministic version follows the same fundamental principles as Algorithm 1, with the difference that momentum terms $M_i^k$ are replaced by the components of the full gradient $\nabla_i f(X^k)$. When $\|\cdot\|_{(i)} = \|\cdot\|_2$ is the Euclidean norm, the update (5) coincides with that of layer-wise Gradient Descent (GD).

How expensive is one step of Algorithm 2? The answer is governed by the sampling distribution $\mathcal{D}$. Consider iteration $k$ and denote $s^k := \min S^k$ (the smallest index of an active layer). The operations performed by the algorithm can be summarized as follows:

(i) **Backward pass:** Backpropagate gradients through layers $[X_{s^k}^k, \ldots, X_b^k]$. Since layers $1, \ldots, s^k - 1$ are frozen, no gradients are computed for them, effectively truncating the gradient flow at the first active layer.

(ii) **Forward pass:** To evaluate the loss, activations must in principle be propagated through all layers $1, \ldots, b$. However, since only layers $[X_{s^k}^k, \ldots, X_b^k]$ are updated at iteration $k$, the activations up to layer $s^k - 1$ may be cached and reused in the next step.

(iii) **Gradient transformation:** Given the gradients $\{\nabla_i f(X^k)\}_{i \in S^k}$, compute the corresponding sharp operators $\{(\nabla_i f(X^k))^\sharp\}_{i \in S^k}$ (or, equivalently, the LMOs; see (4)).

(iv) **Parameter updates:** Update the parameters of layers $\{X_i^k\}_{i \in S^k}$ using their computed (transformed) gradients, while keeping the frozen layers unchanged.

To model the total computational effort of the optimization procedure, we associate a *cost* with each step (measured, for example, in FLOPs or wall-clock time). Let $c_{\mathrm{ov}} \geq 0$ denote the fixed per-iteration overhead (e.g., data loading). As noted above, backpropagation must be performed from the last layer $b$ down to layer $s^k$, while forward-pass activations up to layer $s^k - 1$ can be cached and reused in subsequent iterations. Hence, the costs of steps (i) and (ii) can be aggregated into a single per-layer constant $c_i > 0$ for each $i \in [b]$. In the non-Euclidean setting (where $M^\sharp \neq M$), an additional cost arises from computing sharp operators. We denote by $c_i^\sharp \geq 0$ the combined cost of evaluating this operator and performing the corresponding parameter update for layer $i$ (steps (iii) and (iv)). Under this model, the total compute cost of iteration $k$ is

$$\mathrm{cost}(S^k) := c_{\mathrm{ov}} + \sum_{i=s^k}^{b} c_i + \sum_{i \in S^k} c_i^\sharp, \tag{6}$$

and, consequently, for a fixed target accuracy $\varepsilon > 0$, the expected cost of the entire optimization procedure can be expressed as

$$\text{cost}_\varepsilon(\mathcal{D}) := K \times \mathbb{E}\left[\text{cost}(\hat{S})\right], \tag{7}$$

where we write $\hat{S} \sim \mathcal{D}$ to denote a random variable with the same distribution as that of the samplings (since $\{S^k\}_{k \geq 0}$ are i.i.d.), and $K$ is the number of iterations to reach convergence (interpreted in the nonconvex case as reaching an $\varepsilon$-stationary point in expectation). In the remainder of this paper, we compute $K$ under two smoothness regimes: layer-wise smoothness (Assumption 4.2) and layer-wise $(L^0, L^1)$–smoothness (Assumption 4.3). We then evaluate various layer-update strategies and compute their expected costs as in (7). These results converge to a single conclusion:

> Full-network updates are *not* optimal unless a very specific relationship
> between layer smoothness constants holds.

We provide a rigorous statement and justification for this claim in the sections that follow. The main takeaway, however, is simple: as this condition is highly unlikely to be realized in practice, updating only a subset of layers at each iteration should be seen as more efficient than the default strategy of updating all parameters at each iteration.

Our **contributions** can be summarized as follows:

1. **Challenging full network updates.** We provide, to our knowledge, the first systematic investigation of the practice of updating *all layers* of a network at every iteration. We argue—and rigorously demonstrate both in theory and in practice—that this design choice is *generally suboptimal*.

2. **General framework for sub-network optimization.** We introduce Drop-Muon (Algorithm 1 and its deterministic gradient counterpart Algorithm 2), a principled layer-wise optimization framework with randomized layer subsampling. Drop-Muon strictly generalizes LMO-type methods (including Muon (Jordan et al., 2024), Scion (Pethick et al., 2025), and Gluon (Riabinin et al., 2025b)) by allowing random subsets of layers to be updated per step, with full-network training as a special case. Drop-Muon supports virtually any layer sampling scheme (Section C). In the main part of this paper, we focus on Randomized Progressive Training (RPT) (Szlendak et al., 2024), a natural strategy aligned with backpropagation mechanics that avoids redundant gradient computations and reduces compute cost while maintaining strong convergence guarantees.

3. **Tight iteration complexity guarantees under novel smoothness regimes.** We establish convergence guarantees for Drop-Muon under two regimes: *layer-wise smoothness* (Theorem 4.1) and *layer-wise $(L^0, L^1)$–smoothness* (Theorems 4.2 and 4.4). Our rates recover the state of the art for SGD- and Muon-type methods, and, to our knowledge, provide the first convergence guarantees for progressive training-style methods in the non-smooth setting.

4. **Theoretical compute-optimality results.** To isolate the key phenomena, we first consider a deterministic gradient variant of Drop-Muon (Algorithm 2). Using a simple yet expressive cost model (Equation (6)) accounting for per-layer forward/backward passes, gradient transformations, and parameter updates, we prove that full-network updates are *not* optimal unless a very specific condition on layer smoothness constants holds (Theorems 4.3 and E.2), which is unlikely in practice. This formally justifies selective layer updates as the compute-optimal default.

5. **Empirical validation.** Experiments on `NanoGPT`-124M, a `ResNet50` image-classification task, and small controlled CNN experiments on `MNIST`, `Fashion-MNIST`, and `CIFAR-10` (Sections 6 and G) show that Drop-Muon consistently outperforms standard full-network Muon, achieving the same accuracy up to $1.4\times$ faster in wall-clock time.

## 4 RANDOMIZED PROGRESSIVE TRAINING

The general framework in Algorithm 2 allows virtually any sampling strategy. However, due to the mechanics of backpropagation, it is most natural to update all layers from the last one down to some sampled minimal index. Specifically, if the smallest sampled index at iteration $k$ is $s^k$, then computing the gradient $\nabla_{s^k} f(X^k)$ requires backpropagating from the last layer $b$ up to layer $s^k$, which automatically produces all gradient components $[\nabla_{s^k} f(X^k), \ldots, \nabla_b f(X^k)]$.

Formally, we can define the sampling distribution $\mathcal{D}$ as follows: at each iteration $k$, sample $s^k \in [b]$ with probabilities $p_i := \mathbb{P}\left(s^k = i\right)$, where $\sum_{i=1}^b p_i = 1$ and $p_1 > 0$, and set $S^k = \{s^k, \ldots, b\}$. Algorithms 1 and 2 then update the layers $[X_{s^k}^k, \ldots, X_b^k]$, while $[X_1^k, \ldots, X_{s^k-1}^k]$ remain frozen at their previous values (with the convention that $[X_1^k, X_0^k] = \emptyset$ when $s^k = 1$). We refer to this sampling scheme as *Randomized Progressive Training* (RPT), or Drop-training for short (see Section 5).

### 4.1 ITERATION COMPLEXITY – DETERMINISTIC GRADIENT SETTING

We now analyze the iteration complexity of Algorithm 2 under two smoothness regimes; the proofs are deferred to the Appendix. Throughout, we make the standard assumption that the objective function is lower-bounded.

**Assumption 4.1.** There exist $f^\star \in \mathbb{R}$ such that $f(X) \geq f^\star$ for all $X \in \mathcal{X}$.

This ensures the existence of an approximately stationary point for any desired level of accuracy.

**Smooth case.** We first establish convergence under the *layer-wise smoothness* assumption.

**Assumption 4.2** (supp($\mathcal{D}$)–layer-wise smoothness)**.** The function $f : \mathcal{X} \mapsto \mathbb{R}$ is supp($\mathcal{D}$)–layer-wise $L^0$–smooth with constants $L^0 := \{(L_{1,S}^0, \ldots, L_{b,S}^0)\}_{S \in \text{supp}(\mathcal{D})}$, $(L_{1,S}^0, \ldots, L_{b,S}^0) \in \mathbb{R}_+^b$, i.e., for any $S \in \text{supp}(\mathcal{D})$,

$$f(X + \Gamma) - f(X) - \langle \nabla f(X), \Gamma \rangle \leq \sum_{i \in S} \frac{L_{i,S}^0}{2} \|\Gamma_i\|_{(i)}^2 .$$

for all $X = [X_1, \ldots, X_b] \in \mathcal{X}$ and $\Gamma = [\Gamma_1, \ldots, \Gamma_b] \in \mathcal{X}$ such that $\Gamma_i = 0$ for all $i \notin S$.

We take $L^0$ to be the smallest collection of constants satisfying the above. Throughout, we use supp($\mathcal{D}$) to denote the subsets of $[b]$ assigned positive probability mass by $\mathcal{D}$. In the progressive training setting, these supported sets take the form supp($\mathcal{D}$) $= \{\{j, \ldots, b\}, j \in [b]\}$. By definition, $L_{i,S}^0 = 0$ whenever $i \notin S$. Moreover, if $S_1, S_2 \in \text{supp}(\mathcal{D})$ are such that $S_1 \subseteq S_2$, then $L_{i,S_1}^0 \leq L_{i,S_2}^0$ (see Lemma B.1).

The assumption is inspired by the coordinate descent (CD) literature (Wright, 2015), reducing to the standard block-wise Lipschitz continuity of the gradient (Richtárik & Takáč, 2014a; Fercoq & Richtárik, 2015; Qu & Richtárik, 2016) in the special case when supp($\mathcal{D}$) $= \{\{1\}, \{2\}, \ldots, \{b\}\}$. Assumption 4.2 captures the intuition that each subset of layers can have its own effective smoothness constant, allowing tighter bounds on the local curvature of $f$ and better reflecting the structure of the model. Importantly, it is *not* more restrictive than standard smoothness–rather, it offers a richer parametric description by assigning separate constants to different layer subsets, allowing a more precise analysis without shrinking the function class.

With the assumptions in place, we are now ready to state the first formal convergence result.

**Theorem 4.1.** Let Assumptions 4.1 and 4.2 hold, and let $\{X^k\}_{k=0}^{K-1}$ be the iterates of Algorithm 2 run with stepsizes $\gamma_i^k = 1/L_{i,S^k}^0$. Then

$$\frac{1}{K} \sum_{k=0}^{K-1} \sum_{i=1}^b \frac{w_i}{\frac{1}{b}\sum_{j=1}^b w_j} \mathbb{E}\left[\left\|\nabla_i f(X^k)\right\|_{(i)\star}^2\right] \leq \frac{f(X^0) - f^\star}{K\left(\frac{1}{b}\sum_{j=1}^b w_j\right)},$$

where $w_i := \sum_{s=1}^i \frac{p_s}{2L_{i,\{s,\ldots,b\}}^0}$.

Theorem 4.1 establishes an $\mathcal{O}(K^{-1})$ convergence rate for a weighted sum of squared gradient component norms, matching the theoretical rates previously established for Muon-type methods under classical smoothness (Li & Hong, 2025; Pethick et al., 2025; Kovalev, 2025). For the result to be meaningful, every gradient component must contribute to the weighted average. In other words, the sampling distribution must satisfy $w_i > 0$ for all $i \in [b]$. This is a natural requirement, equivalent to ensuring that $p_1 > 0$, i.e., that all layers are updated with nonzero probability. Obviously, if this was not the case, the first layer would be completely ignored, making convergence impossible.

**Generalized smooth case.** Layer-wise optimizers considered here are designed for deep learning, where the classical smoothness assumption is often violated (Zhang et al., 2020; Riabinin et al., 2025b). Consequently, the layer-wise smoothness model in Assumption 4.2 may not accurately capture the local geometry of the loss. To address this, we adopt a more expressive framework building upon $(L^0, L^1)$–smoothness (Zhang et al., 2020; Chen et al., 2023b). Assumption 4.3 below generalizes Assumption 4.2 by letting the local curvature of each layer depend not only on fixed constants $L_{i,S}^0$, but also on the magnitude of the layer's gradient via additional terms $L_{i,S}^1 \|\nabla_i f(X)\|_{(i)\star}$.

**Assumption 4.3** (supp($\mathcal{D}$)–layer-wise $(L^0, L^1)$–smoothness). The function $f : \mathcal{X} \mapsto \mathbb{R}$ is supp($\mathcal{D}$)–layer-wise $(L^0, L^1)$–smooth with constants $L^\alpha := \{(L_{1,S}^\alpha, \dots, L_{b,S}^\alpha)\}_{S \in \text{supp}(\mathcal{D})}$, $(L_{1,S}^\alpha, \dots, L_{b,S}^\alpha) \in \mathbb{R}_+^b$, $\alpha \in \{0, 1\}$, i.e., for any $S \in \text{supp}(\mathcal{D})$,

$$f(X + \Gamma) - f(X) - \langle \nabla f(X), \Gamma \rangle \leq \sum_{i \in S} \frac{L_{i,S}^0 + L_{i,S}^1 \|\nabla_i f(X)\|_{(i)\star}}{2} \|\Gamma_i\|_{(i)}^2 ,$$

for all $X = [X_1, \dots, X_b] \in \mathcal{X}$ and $\Gamma = [\Gamma_1, \dots, \Gamma_b] \in \mathcal{X}$ such that $\Gamma_i = 0$ for all $i \notin S$.

As with Assumption 4.2, assigning separate constants to each subset of layers $S$ allows for tighter, subset-specific bounds, reflecting the interactions among layers.

**Theorem 4.2.** Let Assumptions 4.1 and 4.3 hold, fix $\varepsilon > 0$, and let $\{X^k\}_{k=0}^{K-1}$ be the iterates of Algorithm 2 run with stepsizes $\gamma_i^k = \left(L_{i,S^k}^0 + L_{i,S^k}^1 \|\nabla_i f(X^k)\|_{(i)\star}\right)^{-1}$. Then, to guarantee that

$$\min_{k=0,\dots,K-1} \sum_{i=1}^{b} \left[ \frac{w_i}{\frac{1}{b}\sum_{l=1}^{b} w_l} \mathbb{E}\left[\|\nabla_i f(X^k)\|_{(i)\star}\right] \right] \leq \varepsilon,$$

it suffices to run the algorithm for

$$K = \left\lceil \frac{2\delta^0 \sum_{i=1}^{b} \frac{\left(\sum_{s=1}^{i} p_s\right)^2 \left(\sum_{s=1}^{i} p_s L_{i,\{s,\dots,b\}}^0\right)}{\left(\sum_{s=1}^{i} p_s L_{i,\{s,\dots,b\}}^1\right)^2}}{\varepsilon^2 \left(\frac{1}{b}\sum_{l=1}^{b} w_l\right)^2} + \frac{2\delta^0}{\varepsilon\left(\frac{1}{b}\sum_{l=1}^{b} w_l\right)} \right\rceil$$

iterations, where $\delta^0 := f(X^0) - f^\star$ and $w_i := \frac{\left(\sum_{s=1}^{i} p_s\right)^2}{\sum_{s=1}^{i} p_s L_{i,\{s,\dots,b\}}^1}$.

Similar to Theorem 4.1, Theorem 4.2 guarantees an $\mathcal{O}(K^{-1/2})$ convergence rate for a weighted sum of gradient component norms.[1] Again, the sampling distribution must ensure that $w_i > 0$ for all $i \in [b]$, which amounts to requiring that $p_1 > 0$. In the extreme case when $(p_1, p_2, \dots, p_b) = (1, 0, \dots, 0)$, corresponding to full-network training, the weights simplify to $w_i = 1/L_{i,[b]}^1$, exactly recovering the convergence rate of deterministic Gluon (Riabinin et al., 2025b, Theorem 1), demonstrating the tightness of our guarantees. Importantly, the stepsizes naturally scale inversely with the layer-specific smoothness constants and gradient magnitudes. This automatic adaptation to local geometry prevents overshooting, ensures stable convergence, and allows more aggressive updates when the $(L^0, L^1)$ constants and gradient norms are small.

### 4.1.1 COST OPTIMIZATION

Let us now make clear why performing full-network updates in all iterations is *not* in general an optimal strategy. We first consider the layer-wise smooth case. According to Theorem 4.1, under Assumption 4.2, Algorithm 2 guarantees that $\frac{1}{K}\sum_{k=0}^{K-1}\sum_{i=1}^{b} \mathbb{E}\left[\|\nabla_i f(X^k)\|_{(i)\star}^2\right] \leq \varepsilon$ after

$$K = \left\lceil \frac{f(X^0) - f^\star}{\varepsilon \times \min_{i \in [b]}\left[\sum_{s=1}^{i} \frac{p_s}{2L_{i,\{s,\dots,b\}}^0}\right]} \right\rceil$$

---

[1]Theorem 4.2 bounds the gradient norms directly, while Theorem 4.1 bounds their squares; this difference naturally arises from the distinct smoothness models used in each analysis.

iterations. Since in this case $\mathbb{E}\left[\mathrm{cost}(\hat{S})\right] = c_{\mathrm{ov}} + \sum_{j=1}^{b}(c_j + c_j^\sharp)\sum_{i=1}^{j} p_i$, the expected total cost of the optimization procedure satisfies

$$\mathrm{cost}_\varepsilon(\mathcal{D}) \overset{(7)}{=} K \times \mathbb{E}\left[\mathrm{cost}(\hat{S})\right] \propto \frac{c_{\mathrm{ov}} + \sum_{i=1}^{b}(c_i + c_i^\sharp)\sum_{s=1}^{i} p_s}{\min_{i\in[b]}\left[\sum_{s=1}^{i} \frac{p_s}{2L_{i,\{s,\ldots,b\}}^0}\right]} \tag{8}$$

(see Section E.1.1 for the details of the derivation). Finding the sampling distribution that minimizes $\mathrm{cost}_\varepsilon(\mathcal{D})$ is equivalent to optimizing over $\{p_i\}_{i\in[b]}$. Letting $(p_1, p_2, \ldots, p_b) = (1, 0, \ldots, 0)$ recovers full-network training, which serves as a natural baseline. Yet, can we do better? The following theorem shows that this configuration is optimal under very specific conditions only.

**Theorem 4.3.** The cost (8) is minimized by $(p_1, p_2, \ldots, p_b) = (1, 0, \ldots, 0)$ **if and only if**

$$L_{1,\{1,\ldots,b\}} = \max_{i\in[b]} L_{i,\{1,\ldots,b\}}.$$

Note that the condition in Theorem 4.3 is entirely independent of the cost parameters! In fact, one can derive a *recursive construction of the optimal probabilities* that depends solely on the smoothness constants, from which Theorem 4.3 follows as a simple corollary (see Remark E.1).

The layer-wise $(L^0, L^1)$–smooth case (Theorem E.2) leads to the same conclusion: **full-network updates are optimal only if the first layer is associated with the largest smoothness constant**– a restrictive and rarely observed condition, as confirmed by experiments on `NanoGPT` (Riabinin et al., 2025b). While broader validation is needed, there is no reason to expect this phenomenon to hold broadly. Overall, from the theoretical standpoint, *the prevalent practice of always updating all layers is fundamentally flawed*.

## 4.2 ITERATION COMPLEXITY – STOCHASTIC GRADIENT SETTING

We now turn to the convergence analysis of the practical variant of Drop-Muon with stochastic gradients and momentum (Algorithm 1), within the general layer-wise $(L^0, L^1)$–smoothness framework.

**Assumption 4.4** (supp$(\mathcal{D})$–layer-wise $(L^0, L^1)$–smoothness II). The function $f : \mathcal{X} \mapsto \mathbb{R}$ is supp$(\mathcal{D})$–layer-wise $(L^0, L^1)$–smooth with constants $L^\alpha := \{(L_{1,S}^\alpha, \ldots, L_{b,S}^\alpha)\}_{S\in\mathrm{supp}(\mathcal{D})}$, $(L_{1,S}^\alpha, \ldots, L_{b,S}^\alpha) \in \mathbb{R}_+^b, \alpha \in \{0, 1\}$, i.e., for any $S \in \mathrm{supp}(\mathcal{D})$,

$$\|\nabla_i f(X + \Gamma) - \nabla_i f(X)\|_{(i)\star} \leq \left(L_{i,S}^0 + L_{i,S}^1 \|\nabla_i f(X)\|_{(i)\star}\right)\|\Gamma_i\|_{(i)}$$

for all $X = [X_1, \ldots, X_b] \in \mathcal{X}$ and $\Gamma = [\Gamma_1, \ldots, \Gamma_b] \in \mathcal{X}$ such that $\Gamma_i = 0$ for all $i \notin S$.

Assumption 4.4 is a slightly stronger variant of Assumption 4.3 used in the analysis of Algorithm 2 (see Lemma H.3). Consequently, all results proven under Assumption 4.3 carry over to this setting. This stronger form is necessary to rigorously establish convergence in the stochastic case.

Furthermore, we assume access to a standard stochastic gradient oracle with bounded variance.

**Assumption 4.5.** The stochastic gradient estimator $\nabla f(\cdot; \xi) : \mathcal{X} \mapsto \mathcal{X}$ is unbiased and has bounded variance, i.e., $\mathbb{E}_{\xi\sim\mathcal{D}}[\nabla f(X; \xi)] = \nabla f(X)$ for all $X \in \mathcal{X}$ and there exist $\sigma_i \geq 0$ such that $\mathbb{E}_{\xi\sim\mathcal{D}}\left[\|\nabla_i f(X; \xi) - \nabla_i f(X)\|_2^2\right] \leq \sigma_i^2$ for all $X \in \mathcal{X}$ and $i = 1, \ldots, p$.

Note that, to facilitate the proofs, the variance bound in Assumption 4.5 is defined with respect to the Euclidean norm, consistent with prior analyses (Pethick et al., 2025; Kovalev, 2025; Riabinin et al., 2025b). This is not restrictive: since $\mathcal{X}$ is finite-dimensional, there exist $\underline{\rho}_i, \bar{\rho}_i > 0$ such that $\underline{\rho}_i \|X_i\|_{(i)} \leq \|X_i\|_2 \leq \bar{\rho}_i \|X_i\|_{(i)}$ for all $X_i \in \mathcal{X}_i$ (equivalently, $\underline{\rho}_i \|X_i\|_2 \leq \|X_i\|_{(i)\star} \leq \bar{\rho}_i \|X_i\|_2$).

**Theorem 4.4.** Let Assumptions 4.1, 4.4, and 4.5 hold. Let $\{X^k\}_{k=0}^{K}$, $K \geq 0$, be the iterates of Algorithm 1 initialized with $M_i^0 = \nabla_i f(X^0; \xi^0)$ and run with $t_i^k \equiv t_i = \eta_i/(K+1)^{3/4}$, where

$$0 < \eta_i^2 \leq \min\left\{\tfrac{1}{4}(K+1)^{1/2}\left(\sum_{s=1}^{i} p_s L_{i,\{s,\ldots,b\}}^1\right)^{-1}\left(\sum_{s=1}^{b} p_s \max_{i\in[b]} L_{i,\{s,\ldots,b\}}^1\right)^{-1},\right.$$

$$\frac{\rho_i p_1}{16\bar{\rho}_i(1-\beta_i)} \left( \sum_{s=1}^{i} p_s \right)^{-1} \left( \sum_{s=1}^{i} p_s L^1_{i,\{s,\dots,b\}} \right)^{-1} \left( \sum_{s=1}^{b} p_s \max_{i\in[b]} L^1_{i,\{s,\dots,b\}} \right)^{-1}, 1 \Bigg\}, \quad \text{and}$$

$\beta_i \equiv \beta = (K+1)^{-1/2}$. Then

$$\min_{k=0,\dots,K} \sum_{i=1}^{b} \frac{(\sum_{s=1}^{i} p_s)\eta_i}{\frac{1}{b}\sum_{l=1}^{b}\sum_{s=1}^{l} p_s \eta_l} \mathbb{E}\left[ \left\| \nabla_i f(X^k) \right\|_{(i)\star} \right]$$

$$\leq \frac{3\delta^0}{(K+1)^{1/4}\frac{1}{b}\sum_{l=1}^{b}\sum_{s=1}^{l} p_s \eta_l} + \frac{6}{(K+1)^{1/2}} \sum_{i=1}^{b} \frac{\eta_i \bar{\rho}_i \sigma_i}{\frac{1}{b}\sum_{l=1}^{b}\sum_{s=1}^{l} p_s \eta_l} + \sum_{i=1}^{b} \frac{2\bar{\rho}_i \sigma_i (\sum_{s=1}^{i} p_s)\eta_i}{(K+1)^{1/4}\frac{1}{b}\sum_{l=1}^{b}\sum_{s=1}^{l} p_s \eta_l}$$

$$+ \sum_{i=1}^{b} \frac{2\bar{\rho}_i \eta_i^2}{\rho_i(K+1)^{1/4}\frac{1}{b}\sum_{l=1}^{b}\sum_{s=1}^{l} p_s \eta_l} \left( \sum_{s=1}^{i} p_s L^0_{i,\{s,\dots,b\}} + \left( \sum_{s=1}^{i} p_s L^1_{i,\{s,\dots,b\}} \right) \left( \sum_{s=1}^{i} p_s \frac{L^0_{i,\{s,\dots,b\}}}{L^1_{i,\{s,\dots,b\}}} \right) \right)$$

$$+ \sum_{i=1}^{b} \frac{\eta_i^2}{2(K+1)^{3/4}\frac{1}{b}\sum_{l=1}^{b}\sum_{s=1}^{l} p_s \eta_l} \left( \sum_{s=1}^{i} p_s L^0_{i,\{s,\dots,b\}} + \left( \sum_{s=1}^{i} p_s L^1_{i,\{s,\dots,b\}} \right) \left( \sum_{s=1}^{i} p_s \frac{L^0_{i,\{s,\dots,b\}}}{L^1_{i,\{s,\dots,b\}}} \right) \right).$$

Theorem 4.4 establishes an $\mathcal{O}(K^{-1/4})$ convergence rate for a weighted sum of gradient component norms, in line with the state-of-the-art results for SGD- and Muon-type methods (Cutkosky & Mehta, 2020; Sun et al., 2023; Kovalev, 2025; Riabinin et al., 2025b). As in the deterministic setting, one could attempt a cost analysis; however, the stochastic bound's complexity necessitates a case-by-case treatment depending on which term dominates. While this prevents deriving as clean analytic results as in Section 4.2, our experiments (Section 6) clearly confirm that the approach remains effective in the stochastic setting, too.

## 5 PRIOR WORK ON PROGRESSIVE TRAINING

The concept of partial network updates has been explored in several prior works. Progressive Training (PT) (Karras et al., 2017) was first introduced in the context of Generative Adversarial Networks (GANs) (Goodfellow et al., 2014). The central idea is to begin with a shallow network trained on low-resolution inputs and gradually increase both the input resolution and network depth by adding layers over time. This progressive growing strategy has been shown to improve computational efficiency and stabilize the training process. Despite its empirical success, PT has remained a largely heuristic method without formal convergence guarantees. An early attempt to provide such guarantees was made by Wang et al. (2022) with ProgFed, which also extended progressive training to distributed settings. However, the accompanying theoretical analysis has been criticized as vacuous (Szlendak et al., 2024). A more rigorous treatment was proposed by Szlendak et al. (2024), who introduced Randomized Progressive Training (RPT). This method can be viewed as a randomized proxy for PT and was the first to establish theoretical convergence rates for progressive training on general smooth objectives, building on the framework of Randomized Coordinate Descent (RCD) (Nesterov, 2012; Richtárik & Takáč, 2014b). Conceptually, RPT is closely related to our approach: like Drop-Muon, it updates a random subnetwork at each iteration. However, the method of Szlendak et al. (2024) is simply a form of sketched GD with a particular choice of sketch operator, and has several limitations that restrict its practical utility. First, it treats network parameters as a flat vector, ignoring the layer-wise structure. Consequently, it fails to exploit layer-specific, non-Euclidean update rules–failing, to recover methods such as Muon as special cases. Second, RPT requires non-stochastic gradient computation, making it computationally infeasible for large-scale architectures.

## 6 EXPERIMENTS

We evaluate Drop-Muon on five benchmarks: `NanoGPT` (Karpathy, 2023) trained on the `FineWeb10B` dataset (Penedo et al., 2024), `ResNet50` on `Tiny ImageNet`, and 3-layer convolutional neural networks (CNNs) of varying capacity on `MNIST`, `Fashion-MNIST`, and `CIFAR-10` (the latter three experiments are presented in Section G). Additional experiments and details on the hyperparameter-selection procedure are provided in Section G.

Our aim is to design experiments that directly support and validate our theory. To this end, we use as our primary baseline the most natural comparison point: the standard full-network Muon.

**Results on `NanoGPT`.** We train a `NanoGPT` model (Karpathy, 2023) with 124M parameters on the `FineWeb10B` dataset (Penedo et al., 2024), using input sequences of length 1024 and a batch

size of 256. Optimization is performed with Muon and Drop-Muon, employing spectral-norm LMOs for hidden layers and $\ell_\infty$-norm LMOs for the embedding and output layers, following Pethick et al. (2025). For spectral norm LMOs, updates are approximated computed using five Newton–Schulz iterations (Kovarik, 1970; Björck & Bowie, 1971), as in Jordan et al. (2024).

Learning rates are tuned for each optimizer and experimental setting, initialized from the configurations in (Riabinin et al., 2025a). We use the same learning-rate schedule as Karpathy (2023) and fix the momentum parameter to 0.9. Model and optimizer hyperparameters are summarized in Tables 1 and 2.

The results are shown in Figures 1 and 2. Across all runs, Drop-Muon outperforms the full-network Muon baseline in wall-clock time (Figure 1) and achieves consistent speedups across all loss-threshold targets (Figure 2).

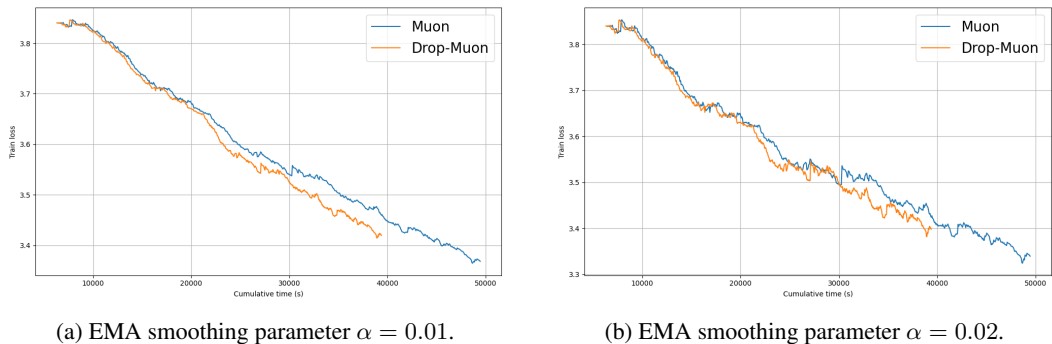

(a) EMA smoothing parameter $\alpha = 0.01$.  (b) EMA smoothing parameter $\alpha = 0.02$.

Figure 1: Training losses of Muon and Drop-Muon using the Gaussian-shift layer sampling with parameter $\sigma = 0.35$ and a late-wake-up parameter of $0.45$ (i.e., Drop-Muon activates after 45% of the first epoch). Both plots use a down-sampling factor of 10.

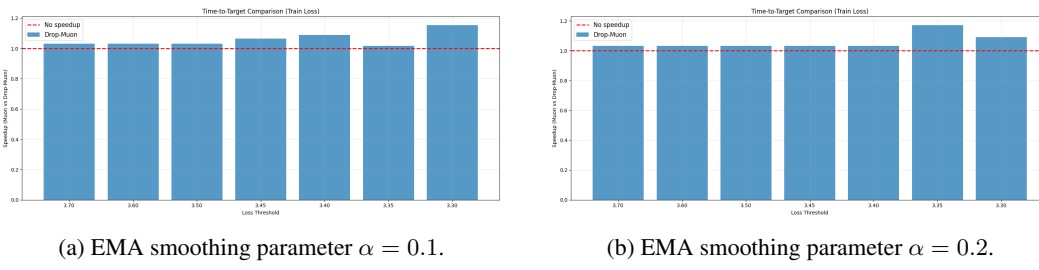

(a) EMA smoothing parameter $\alpha = 0.1$.  (b) EMA smoothing parameter $\alpha = 0.2$.

Figure 2: `NanoGPT` speedup of Drop-Muon with the Gaussian-shift layer sampling over Muon.

**Results on `ResNet50`.** We next evaluate Drop-Muon on an image-classification task, training `ResNet50` on `Tiny ImageNet`. For each setting, we report the mean performance + three standard deviations across five runs.

The results in Figure 3 again show that Drop-Muon achieves faster training than the full-network Muon baseline. While the per-epoch training accuracy of Drop-Muon is initially lower than that of full-network Muon (left panels of Figures 3a and 3b), the wall-clock view tells a different story: Drop-Muon consistently outperforms the baseline. In practical terms, Drop-Muon reaches a given accuracy threshold sooner, resulting in faster training.

Analogous trends are observed in the additional experiments reported in Section G.

In summary, even though Drop-Muon trains on partial gradients, it reaches high accuracy levels earlier than Muon in wall-clock time. Importantly, Drop-Muon is simple to implement, requiring only a few lines of code. Overall, our results demonstrate that Drop-Muon is an effective, practical, and easily implementable strategy that, with a well-chosen sampling strategy, consistently accelerates training while retaining high accuracy, making it a compelling choice for modern neural network optimization.

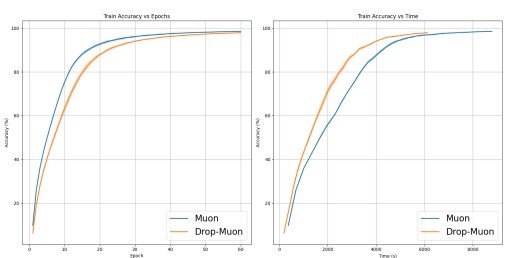 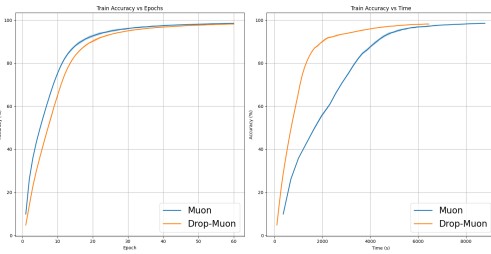

(a) Drop-Muon with uniform layer sampling distribution; learning rate = 0.05, batch size= 256.

(b) Drop-Muon with decreasing linear layer sampling distribution; learning rate = 0.02, batch size= 256.

Figure 3: Training accuracy of Muon and Drop-Muon on `Resnet50`.

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

# APPENDIX

# CONTENTS

# A  ADDITIONAL LITERATURE REVIEW

## A.1  Muon (AND FRIENDS)

Recent work has revisited how neural network parameters are updated, moving beyond simple element-wise gradient steps toward more structured update rules. Notable examples include the Muon optimizer (Jordan et al., 2024), as well as the closely related Scion (Pethick et al., 2025) and Gluon (Riabinin et al., 2025b). All of these methods can be interpreted as instances of algorithms driven by a *linear minimization oracle* (LMO) over norm balls.

Muon, introduced by Jordan et al. (2024), is an optimizer for the hidden layers of neural networks (with first and last layers typically trained using AdamW (Loshchilov & Hutter, 2019)). Given a layer $X_i$ and the associated (stochastic) gradient $G_i$, the update is obtained by solving the constrained optimization problem

$$\min_{\Delta X_i} \ \langle G_i, \Delta X_i \rangle \quad \text{subject to} \quad \|\Delta X_i\|_{2\to 2} \leq t_i, \tag{9}$$

where $t_i > 0$ plays the role of a trust-region radius or stepsize. The solution is characterized by the singular vectors of $G_i$: if $G_i = U_i \Sigma_i V_i^\top$ is its singular value decomposition, then

$$\Delta X_i = -t_i U_i V_i^\top, \qquad X_i^{k+1} = X_i^k + \Delta X_i. \tag{10}$$

In other words, Muon moves in the direction of steepest descent measured in the spectral norm.

Since computing an exact SVD can be expensive, Muon uses the Newton-Schulz method (Kovarik, 1970; Björck & Bowie, 1971) to approximate the orthogonalization. In practice, the algorithm also incorporates momentum, yielding the update

$$M_i^k = (1 - \beta_i) M_i^{k-1} + \beta_i G_i^k,$$
$$X_i^{k+1} = X_i^k - t_i^k \text{NewtonSchulz}(M_i^k),$$

where $\beta_i \in (0, 1]$ is the momentum parameter and $M_i^k$ is the running average of past gradients.

The update rule in (10) can be interpreted as a special case of the more general LMO-based update described in Section 1. By selecting different norms for the LMO ball constraint, one obtains different algorithmic variants. Since our primary focus is on deep learning applications, we are especially interested in matrix norms. Within this setting, a particularly important family is that of *operator norms*, defined for any $A \in \mathbb{R}^{m\times n}$ as

$$\|A\|_{\alpha\to\beta} := \sup_{\|Z\|_\alpha = 1} \|AZ\|_\beta,$$

where $\|\cdot\|_\alpha$ and $\|\cdot\|_\beta$ denote norms on $\mathbb{R}^n$ and $\mathbb{R}^m$, respectively.

In particular, Muon's update in (9) corresponds to an LMO over the spectral norm ball $\mathcal{B}_i^{2\to 2}(0, t_i) := \{Z_i \in \mathcal{X}_i : \|Z_i\|_{2\to 2} \leq t_i\}$, leading to the equivalent update

$$X_i^{k+1} = X_i^k + \text{LMO}_{\mathcal{B}_i^{2\to 2}(0, t_i)}(G_i).$$

Hence, Muon is just one member of a broader family of methods parameterized by the geometry of the update set.

Building on this perspective, Pethick et al. (2025) introduced Scion, which extends LMO-based updates to *all* network layers, rather than being limited to the hidden matrix-shaped layers as in Muon. To better capture layer-specific behavior, Scion employs different norms depending on layer type (scaled spectral norms for the weight matrices of transformer blocks and the $\|\cdot\|_{1\to\infty}$ norm for embedding and output layers).

Finally, Gluon (Riabinin et al., 2025b) generalizes the theory behind both Muon and Scion. It provides a convergence framework for updates over *arbitrary* norm balls. The analysis relies on a novel layer-wise smoothness model (closely related to our Assumptions 4.2, 4.3, and 4.4), capturing heterogeneity through $(L^0, L^1)$ parameters, that more accurately reflects the non-uniform characteristics of deep learning models (see more details in Section A.2). As such, Gluon can be seen as the theoretical foundation unifying the entire class of LMO-based optimizers for training deep networks.

---

**Algorithm 3** Randomized Block Coordinate Descent

---

1: **Input:** Initial iterate $X^0 = [X_1^0, \ldots, X_b^0] \in \mathcal{X}$; stepsizes $\gamma_i^k > 0$, $i \in [b]$, $k \geq 0$
2: **for** $k = 0, \ldots, K - 1$ **do**
3:      Choose index $i_k \in [b]$ randomly                                       ▷ Serial sampling
4:      Set $X_i^{k+1} = X_i^k$ for $i \neq i_k$
5:      Update $X_{i_k}^{k+1} = X_{i_k}^k - \gamma_{i_k}^k \nabla_{i_k} f(X^k)$
6: **end for**

---

### A.2 GENERALIZED SMOOTHNESS

Gradient-based methods are traditionally analyzed under the assumption of Lipschitz smoothness of the gradient. Yet, many modern deep learning tasks violate this assumption (Zhang et al., 2020; Crawshaw et al., 2022), prompting the development of alternative smoothness models. One such model is $(L^0, L^1)$–smoothness, initially introduced by Zhang et al. (2020) for twice continuously differentiable functions and later generalized beyond this setting (Li et al., 2023; Chen et al., 2023b).

This framework has been further tailored to deep learning through the non-Euclidean layer-wise $(L^0, L^1)$–smoothness assumption (Riabinin et al., 2025b), which closely relates to our Assumptions 4.2, 4.3, and 4.4. Concretely, Riabinin et al. (2025b) assume that

$$\|\nabla_i f(X) - \nabla_i f(Y)\|_{(i)\star} \leq \left( L_i^0 + L_i^1 \|\nabla_i f(X)\|_{(i)\star} \right) \|X_i - Y_i\|_{(i)} \tag{11}$$

for all $i \in [b]$ and all $X = [X_1, \ldots, X_p] \in \mathcal{X}$, $Y = [Y_1, \ldots, Y_p] \in \mathcal{X}$. Condition (11) can be interpreted as a global version of our Assumption 4.4. However, it is less precise because it does not allow each subset of layers to have its own effective smoothness constants. By adopting Assumption 4.4 instead of (11), we can derive tighter bounds without restricting the function class under consideration.

Accounting for heterogeneous parameter structures is not novel and has been studied in the context of coordinate descent (Nesterov, 2012; Richtárik & Takáč, 2014b) and in the analysis of algorithms such as signSGD (Bernstein et al., 2018; Crawshaw et al., 2022), AdaGrad (Jiang et al., 2024; Liu et al., 2024), and Adam (Xie et al., 2024).

### A.3 RANDOMIZED BLOCK COORDINATE DESCENT

Coordinate Descent (CD) algorithms have a long history (Southwell, 1940; Powell, 1973; Nesterov, 2012; Wright, 2015) and are among the most widely studied methods for large-scale optimization. Their randomized variants, known as Randomized Block Coordinate Descent (RBCD), have emerged as a powerful class of first-order methods, particularly in high-dimensional problems. Instead of updating the entire parameter vector, RBCD updates only a subset ("block") of variables at each iteration, reducing per-iteration computational cost and improving scalability.

In the context of CD, the parameters are viewed as a flat vector in $\mathbb{R}^d$. The general procedure of RBCD is summarized in Algorithm 3. At each iteration, one block is selected at random and updated, while the remaining blocks remain unchanged. Clearly, Algorithm 3 is a special case of Algorithm 2 with the *serial sampling* strategy (i.e., sampling a single coordinate per iteration) and standard Euclidean updates.

We define a family of linear maps $\{\mathbf{U}_i : \mathcal{X}_i \to \mathcal{X} : i \in [b]\}$ (where $\mathcal{X} = \mathbb{R}^d$ and $\mathcal{X}_i = \mathbb{R}^{d_i}$, with $\sum_{i=1}^b d_i = d$), forming a partition of the identity map, i.e., $X = \sum_{i \in [b]} \mathbf{U}_i(X_i)$. The standard analysis of RBCD relies on the assumption of *blockwise Lipschitz continuity* of the gradient:

$$\|\nabla_i f(X + \mathbf{U}_i(\Gamma_i)) - \nabla_i f(X)\|_{(i)\star} \leq L_i \|\Gamma_i\|_{(i)}, \quad \forall i \in [b], \ X \in \mathcal{X}, \ \Gamma_i \in \mathcal{X}_i. \tag{12}$$

From (12), one obtains the standard quadratic upper bound

$$f(X + \mathbf{U}_i(\Gamma_i)) - f(X) - \langle \nabla_i f(X), \Gamma_i \rangle \leq \frac{L_i}{2} \|\Gamma_i\|_{(i)}^2. \tag{13}$$

This inequality plays a pivotal role in establishing the iteration complexity of RBCD. Moreover, it explains why CD-type methods can be advantageous compared to full gradient descent: they allow

for larger stepsizes, which in turn leads to faster convergence. Indeed, the standard analysis of GD relies on *global* $L$-smoothness, enforcing stepsizes on the order of $1/L$. By contrast, the blockwise smoothness assumption (12) bounds the function's variation when only a single block of coordinates is updated. Consequently, in CD, the safe stepsize for block $i$ scales with $1/L_i$. Since $L$ must capture worst-case variations across all directions (including cross-block interactions), it is often much larger than the individual $L_i$'s.

Complexity analyses of RBCD in the convex setting can be found in Nesterov (2012); Wright (2015). Generalizations to parallel and arbitrary sampling variants were developed by Richtárik & Takáč (2016); Qu & Richtárik (2016). For the nonconvex case considered in this paper, iteration complexity results of Algorithm 2 are only known for serial sampling (Patrascu & Necoara, 2015; Dang & Lan, 2015). Otherwise, some work has focused on the global convergence properties of randomized CD methods (Chen et al., 2023a), rather than the iteration complexity analysis we develop here. It is worth noting that serial sampling is not suitable in the deep neural network context due to the special entanglement structure and the requirements of backpropagation for gradient computation.

## B MORE ON SMOOTHNESS ASSUMPTIONS

Recall that a function $f : \mathcal{X} \mapsto \mathbb{R}$ is supp($\mathcal{D}$)–layer-wise $L^0$–smooth with constants $L^0 := \{(L^0_{1,S}, \ldots, L^0_{b,S})\}_{S \in \text{supp}(\mathcal{D})}, (L^0_{1,S}, \ldots, L^0_{b,S}) \in \mathbb{R}^b_+$ (Assumption 4.2) if for any $S \in \text{supp}(\mathcal{D})$,

$$f(X + \Gamma) - f(X) - \langle \nabla f(X), \Gamma \rangle \leq \sum_{i \in S} \frac{L^0_{i,S}}{2} \|\Gamma_i\|^2_{(i)}$$

for all $X = [X_1, \ldots, X_b] \in \mathcal{X}$ and $\Gamma = [\Gamma_1, \ldots, \Gamma_b] \in \mathcal{X}$ such that $\Gamma_i = 0$ for all $i \notin S$.

**Lemma B.1.** Let $S_1, S_2 \in \text{supp}(\mathcal{D})$ satisfy $S_1 \subseteq S_2$, and suppose Assumption 4.2 holds. Then one can choose the constants $\{L^0_{i,S_1}\}_{i \in S_1}$ so that

$$L^0_{i,S_1} \leq L^0_{i,S_2}.$$

*Proof.* Let $X = [X_1, \ldots, X_b] \in \mathcal{X}$ be arbitrary. By Assumption 4.2, for any $\Gamma = [\Gamma_1, \ldots, \Gamma_b] \in \mathcal{X}$ such that $\Gamma_i = 0$ for all $i \notin S_1$, we have

$$f(X + \Gamma) - f(X) - \langle \nabla f(X), \Gamma \rangle \leq \sum_{i \in S_1} \frac{L^0_{i,S_1}}{2} \|\Gamma_i\|^2_{(i)}.$$

Now, since $S_1 \subseteq S_2$ and $\Gamma_i = 0$ for all $i \in S_2 \setminus S_1$, Assumption 4.2 also gives

$$f(X + \Gamma) - f(X) - \langle \nabla f(X), \Gamma \rangle \leq \sum_{i \in S_2} \frac{L^0_{i,S_2}}{2} \|\Gamma_i\|^2_{(i)} = \sum_{i \in S_1} \frac{L^0_{i,S_2}}{2} \|\Gamma_i\|^2_{(i)}.$$

Therefore, one can always choose $L^0_{i,S_1} \leq L^0_{i,S_2}$. $\square$

**Remark B.1.** We can show that, without further assumptions, there is no finite function $C : [b] \to (0, \infty)$ such that

$$L^0_{i,S} \leq C(|S|) L^0_{i,\{i\}} \tag{14}$$

holds for all $f$. Consider the case $b = 2$ with scalar blocks and the Euclidean norm, suppose that $\text{supp}(\mathcal{D}) = \{\{1\}, \{2\}, \{1, 2\}\}$, and define

$$f(x, y) = \alpha x y$$

for some $\alpha > 0$. For a single-block perturbation in the $x$-coordinate only, we have

$$f(x + \gamma_1, y) - f(x, y) - \partial_x f(x, y) \gamma_1 = \alpha(x + \gamma_1) y - \alpha x y - \alpha \gamma_1 y = 0,$$

which shows that $L^0_{1,\{1\}} = 0$. By symmetry, $L^0_{2,\{2\}} = 0$ as well. However, for a joint perturbation $\Gamma = (\gamma_1, \gamma_2)$, we obtain

$$\begin{aligned} f(x + \gamma_1, y + \gamma_2) - f(x, y) - \langle \nabla f(x, y), \Gamma \rangle &= \alpha(x + \gamma_1)(y + \gamma_2) - \alpha x y - \alpha \gamma_1 y - \alpha \gamma_2 x \\ &= \alpha \gamma_1 \gamma_2. \end{aligned}$$

Now, suppose for a contradiction that (14) holds for some function $C : [b] \to \mathbb{R}$. Then, since $L^0_{1,\{1\}} = 0$, it must be that $L^0_{1,\{1,2\}} = 0$. By Assumption 4.2, this requires

$$\alpha \gamma_1 \gamma_2 \leq \frac{L^0_{1,\{1,2\}}}{2} \gamma_1^2 + \frac{L^0_{2,\{1,2\}}}{2} \gamma_2^2 = \frac{L^0_{2,\{1,2\}}}{2} \gamma_2^2.$$

But the right-hand side is independent of $\gamma_1$, whereas the left-hand side grows without bound as $\gamma_1 \to \infty$ whenever $\gamma_2 \neq 0$. Hence no such constant $L^0_{2,\{1,2\}}$ can exist. This proves that no uniform bound of the form (14) can hold without additional assumptions.

**Remark B.2.** The above argument extends to any two sets with one contained in the other (not just singletons). Let $S_1, S_2 \in \text{supp}(\mathcal{D})$ satisfy $S_1 \subseteq S_2$. We show that, without further assumptions, there is no finite function $C : [b] \times [b] \to (0, \infty)$ such that

$$L^0_{i,S_2} \leq C(|S_1|, |S_2|) L^0_{i,S_1}. \tag{15}$$

Without loss of generality we can restrict attention to the coordinates indexed by $S_2$ (so that $S_2 = [b]$). Let $S_1 = \{1, \ldots, s\}$ and $S_2 \setminus S_1 = \{s+1, \ldots, s+u\}$, so $|S_1| = s$, $|S_2| = s+u$, and $|S_2 \setminus S_1| = u$. Again, consider scalar blocks and define a function $f : \mathbb{R}^b \to \mathbb{R}$ via

$$f(x_1, \ldots, x_s, y_1, \ldots, y_u) = \alpha \left( \sum_{i=1}^{s} x_i \right) \left( \sum_{j=1}^{u} y_j \right)$$

for some $\alpha > 0$. For any increment $\Gamma_{S_1}$ supported on $S_1$ only (i.e., $\gamma_j = 0$ for all $j \notin S_1$) the function $f$ is affine in the $x$-variables with coefficients given by $\alpha \sum_j y_j$. Therefore, for any $X \in \mathbb{R}^b$,

$$f(X + \Gamma_{S_1}) - f(X) - \langle \nabla f(X), \Gamma_{S_1} \rangle$$

$$= \alpha \left( \sum_{i=1}^{s} (x_i + \gamma_i) \right) \left( \sum_{j=1}^{u} y_j \right) - \alpha \left( \sum_{i=1}^{s} x_i \right) \left( \sum_{j=1}^{u} y_j \right) - \alpha \sum_{i=1}^{s} \left( \sum_{j=1}^{u} y_j \right) \gamma_i = 0,$$

and hence $L_{i,S_1}^0 = 0$ for every $i \in S_1$. For a joint perturbation $\Gamma_{S_2} = (\gamma_1, \ldots, \gamma_s, \eta_1, \ldots, \eta_u)$ supported on $S_2$, we have

$$f(X + \Gamma_{S_2}) - f(X) - \langle \nabla f(X), \Gamma_{S_2} \rangle$$

$$= \alpha \left( \sum_{i=1}^{s} (x_i + \gamma_i) \right) \left( \sum_{j=1}^{u} (y_j + \eta_j) \right) - \alpha \left( \sum_{i=1}^{s} x_i \right) \left( \sum_{j=1}^{u} y_j \right)$$

$$- \alpha \sum_{i=1}^{s} \left( \sum_{j=1}^{u} y_j \right) \gamma_i - \alpha \sum_{i=1}^{u} \left( \sum_{j=1}^{s} x_j \right) \eta_i$$

$$= \alpha \left( \sum_{i=1}^{s} \gamma_i \right) \left( \sum_{j=1}^{u} \eta_j \right).$$

In particular, fix any nonzero vectors $\tilde{\gamma} = (\tilde{\gamma}_1, \ldots, \tilde{\gamma}_s)$ and $\eta = (\eta_1, \ldots, \eta_u)$ and let the $x$-perturbation scale by a factor $\lambda$, i.e. $\gamma_i = \lambda \tilde{\gamma}_i$. Then

$$f(X + \Gamma) - f(X) - \langle \nabla f(X), \Gamma \rangle = \alpha \lambda \left( \sum_{i=1}^{s} \tilde{\gamma}_i \right) \left( \sum_{j=1}^{u} \eta_j \right),$$

which grows linearly in $\lambda$. Now, suppose, for contradiction that (15) holds with some finite $C(|S_1|, |S_2|)$. Since $L_{i,S_1}^0 = 0$ for every $i \in S_1$, the inequality forces $L_{i,S_2}^0 = 0$ for all $i \in S_1$. But then Assumption 4.2 would yield

$$\alpha \lambda \left( \sum_{i=1}^{s} \tilde{\gamma}_i \right) \left( \sum_{j=1}^{u} \eta_j \right) = \alpha \left( \sum_{i=1}^{t} \gamma_i \right) \left( \sum_{j=1}^{u} \eta_j \right)$$

$$\leq \sum_{i \in S_1} \frac{L_{i,S_2}^0}{2} \gamma_i^2 + \sum_{i \in S_2 \setminus S_1} \frac{L_{i,S_2}^0}{2} \eta_i^2$$

$$= \sum_{i \in S_2 \setminus S_1} \frac{L_{i,S_2}^0}{2} \eta_i^2,$$

where the right-hand side is independent of the scaling factor $\lambda$. Taking $\lambda \to \infty$ makes the left-hand side arbitrarily large, resulting in a contradiction. Hence no such $C$ can exist.

## C  ARBITRARY SAMPLING

One might wonder why the cost model in Section 3 distinguishes between two sets of constants $\{c_i\}_{i\in[b]}$ and $\{c_i^\sharp\}_{i\in[b]}$, given that in the RPT case, $c_i$ and $c_i^\sharp$ could be combined into a single constant. This is because Algorithms 1 and 2 do not restrict us to the layer sampling scheme studied in Section 4. One could argue that deviating from this progressive training framework is inefficient, as it discards gradients obtained "for free" during backpropagation: by design, at iteration $k$, the gradients $[\nabla_{s^k} f(X^k), \ldots, \nabla_b f(X^k)]$, where $s^k := \min S^k$, are necessarily computed, and thus available for the update (if, for some block $i \in [b]$, the cost $c_i^\sharp$ of applying the sharp operator is large, one can simply bypass this step and instead use the standard gradient update; this corresponds to setting $\|\cdot\|_{(i)} = \|\cdot\|_2$ in the algorithm). Nevertheless, since the general iteration complexity results in Theorems D.1, D.4 and F.1 hold for any distribution $\mathcal{D}$, the framework naturally allows experimentation with alternative sampling schemes, even if only out of theoretical curiosity.

With this in mind, we generalize the results from Section 4 to *arbitrary* layer samplings. Formally, at each iteration $k$, Algorithms 1 and 2 sample a random subset of layers $S^k \subseteq [b]$ from a distribution

$$\mathcal{D} : \mathfrak{P}([b]) \to [0,1], \qquad \sum_{S \subseteq [b]} \mathcal{D}(S) = 1,$$

with support $\mathrm{supp}(\mathcal{D}) := \{S \in \mathfrak{P}([b]) : \mathcal{D}(S) > 0\}$, where $\mathfrak{P}([b])$ denotes the power set of $[b]$. Only the layers in $S^k$ are updated, while the rest remain fixed. We write $\hat{S} \sim \mathcal{D}$ for a set-valued random variable with the same distribution as $S^k$. This formulation defines a broad family of algorithms, parameterized jointly by the sampling distribution $\mathcal{D}$ and the choice of layer-wise norms $\|\cdot\|_{(i)}$.

Mirroring the earlier discussion, we organize the analysis around two smoothness regimes–layer-wise smooth (Assumption 4.2) and generalized layer-wise smooth (Assumptions 4.3 and 4.4).

In what follows, we present convergence results for Algorithms 1 (Theorem F.1) and 2 (Theorems D.1 and D.4). When specialized to the RPT setting, these general results recover the guarantees stated in Theorems 4.1, 4.2, and 4.4, as detailed in Remarks D.3, D.6, and F.3.

## D    CONVERGENCE RESULTS – DETERMINISTIC GRADIENT CASE

### D.1    LAYER-WISE SMOOTH CASE

**Theorem D.1.** Let Assumptions 4.1 and 4.2 hold, and let $\{X^k\}_{k=0}^{K-1}$ be the iterates of Algorithm 2 run with stepsizes $\gamma_i^k = 1/L_{i,S^k}^0$. Then

$$\frac{1}{K} \sum_{k=0}^{K-1} \sum_{i=1}^{b} \frac{w_i}{\frac{1}{b}\sum_{j=1}^{b} w_j} \mathbb{E}\left[\left\|\nabla_i f(X^k)\right\|_{(i)\star}^2\right] \leq \frac{f(X^0)-f^\star}{K\left(\frac{1}{b}\sum_{j=1}^{b} w_j\right)},$$

where $w_i := \mathbb{E}\left[\frac{\mathbb{I}(i\in\hat{S})}{2L_{i,\hat{S}}^0}\right]$.

**Remark D.2.** For Theorem D.1 to be meaningful, the sampling must ensure that $w_i > 0$ for every $i \in [b]$, i.e.,

$$\mathbb{E}\left[\frac{\mathbb{I}\left(i \in \hat{S}\right)}{2L_{i,\hat{S}}^0}\right] > 0.$$

Otherwise, some layers would receive zero weight and the bound would provide no control over them. This is a natural condition, requiring that every layer is sampled with positive probability. Indeed, if $\mathbb{P}\left(i \in \hat{S}\right) = 0$, then the expectation vanishes, so $w_i > 0$ necessarily implies $\mathbb{P}\left(i \in \hat{S}\right) > 0$ for all $i \in [b]$.

**Remark D.3.** In the case of RPT (see Section 4), the weights from Theorem D.1 become

$$\mathbb{E}\left[\frac{\mathbb{I}\left(i \in \hat{S}\right)}{2L_{i,\hat{S}}^0}\right] = \sum_{s=1}^{b} \frac{\mathbb{I}(i \in \{s,\ldots,b\})}{2L_{i,\{s,\ldots,b\}}^0} p_s = \sum_{s=1}^{i} \frac{p_s}{2L_{i,\{s,\ldots,b\}}^0}.$$

Substituting this expression into the rate yields the result in Theorem 4.1.

*Proof of Theorem D.1.* Using Assumption 4.2, we get

$$f(X^{k+1})$$

$$\leq \quad f(X^k) + \left\langle \nabla f(X^k), X^{k+1} - X^k \right\rangle + \sum_{i\in S^k} \frac{L_{i,S^k}^0}{2}\left\|X_i^k - X_i^{k+1}\right\|_{(i)}^2$$

$$= \quad f(X^k) + \sum_{i\in S^k}\left(\left\langle \nabla_i f(X^k), X_i^{k+1} - X_i^k \right\rangle_{(i)} + \frac{L_{i,S^k}^0}{2}\left\|X_i^k - X_i^{k+1}\right\|_{(i)}^2\right)$$

$$= \quad f(X^k) + \sum_{i\in S^k}\left(-\gamma_i^k\left\langle \nabla_i f(X^k), (\nabla_i f(X^k))^\sharp \right\rangle_{(i)} + \frac{L_{i,S^k}^0(\gamma_i^k)^2}{2}\left\|(\nabla_i f(X^k))^\sharp\right\|_{(i)}^2\right)$$

$$\overset{(50),(51)}{=} \quad f(X^k) + \sum_{i\in S^k}\left(-\gamma_i^k\left\|\nabla_i f(X^k)\right\|_{(i)\star}^2 + \frac{L_{i,S^k}^0(\gamma_i^k)^2}{2}\left\|\nabla_i f(X^k)\right\|_{(i)\star}^2\right).$$

Choosing $\gamma_i^k = 1/L_{i,S^k}^0$ and rearranging, we get

$$\sum_{i\in S^k}\frac{\left\|\nabla_i f(X^k)\right\|_{(i)\star}^2}{2L_{i,S^k}^0} \leq f(X^k) - f(X^{k+1}).$$

Taking expectation conditional on $X^k$ (denoted as $\mathbb{E}_k\left[\cdot\right]$) gives

$$f(X^k) - \mathbb{E}_k\left[f(X^{k+1})\right] \quad \geq \quad \mathbb{E}_k\left[\sum_{i\in S^k}\frac{\left\|\nabla_i f(X^k)\right\|_{(i)\star}^2}{2L_{i,S^k}^0}\right]$$

$$= \mathbb{E}_k \left[ \sum_{i=1}^{b} \mathbb{I} \left( i \in S^k \right) \frac{\left\| \nabla_i f(X^k) \right\|_{(i)\star}^2}{2 L_{i,S^k}^0} \right]$$

$$= \sum_{i=1}^{b} \mathbb{E} \left[ \frac{\mathbb{I} \left( i \in \hat{S} \right)}{2 L_{i,\hat{S}}^0} \right] \left\| \nabla_i f(X^k) \right\|_{(i)\star}^2$$

$$= \sum_{i=1}^{b} w_i \left\| \nabla_i f(X^k) \right\|_{(i)\star}^2,$$

where we denoted $w_i := \mathbb{E} \left[ \frac{\mathbb{I}(i \in \hat{S})}{2 L_{i,\hat{S}}^0} \right]$ and $\mathbb{I} (\cdot)$ is the indicator function (i.e., for any event $A$, $\mathbb{I}(A) = 1$ if $A$ and $\mathbb{I}(A) = 0$ otherwise). Taking full expectation, summing over the first $K$ iterations and dividing by $\frac{K}{b} \sum_{j=1}^{b} w_j$, we get

$$\frac{1}{K} \sum_{k=0}^{K-1} \sum_{i=1}^{b} \frac{w_i}{\frac{1}{b} \sum_{j=1}^{b} w_j} \mathbb{E} \left[ \left\| \nabla_i f(X^k) \right\|_{(i)\star}^2 \right] \leq \frac{1}{\frac{K}{b} \sum_{j=1}^{b} w_j} \sum_{k=0}^{K-1} \left( \mathbb{E} \left[ f(X^k) \right] - \mathbb{E} \left[ f(X^{k+1}) \right] \right)$$

$$\leq \frac{f(X^0) - f^\star}{K \left( \frac{1}{b} \sum_{j=1}^{b} w_j \right)}.$$

$\square$

## D.2 Layer-wise generalized smooth case

**Theorem D.4.** Let Assumptions 4.1 and 4.3 hold, fix $\varepsilon > 0$, and let $\{X^k\}_{k=0}^{K-1}$ be the iterates of Algorithm 2 run with stepsizes $\gamma_i^k = \left( L_{i,S^k}^0 + L_{i,S^k}^1 \left\| \nabla_i f(X^k) \right\|_{(i)\star} \right)^{-1}$. Then, to guarantee that

$$\min_{k=0,\ldots,K-1} \sum_{i=1}^{b} \left[ \frac{w_i}{\frac{1}{b} \sum_{l=1}^{b} w_l} \mathbb{E} \left[ \left\| \nabla_i f(X^k) \right\|_{(i)\star} \right] \right] \leq \varepsilon,$$

it suffices to run the algorithm for

$$K = \left\lceil \frac{2\delta^0 \sum_{i=1}^{b} \frac{\mathbb{P}(i \in \hat{S}) \mathbb{E} \left[ L_{i,\hat{S}}^0 \middle| i \in \hat{S} \right]}{\left( \mathbb{E} \left[ L_{i,\hat{S}}^1 \middle| i \in \hat{S} \right] \right)^2}}{\varepsilon^2 \left( \frac{1}{b} \sum_{l=1}^{b} w_l \right)^2} + \frac{2\delta^0}{\varepsilon \left( \frac{1}{b} \sum_{l=1}^{b} w_l \right)} \right\rceil$$

iterations, where $\delta^0 := f(X^0) - f^\star$ and $w_i := \frac{\mathbb{P}(i \in \hat{S})}{\mathbb{E} \left[ L_{i,\hat{S}}^1 \middle| i \in \hat{S} \right]}$.

**Remark D.5.** The guarantee in Theorem D.4 is meaningful only if $w_i > 0$ for all $i \in [b]$, which is equivalent to requiring that $\mathbb{P} \left( i \in \hat{S} \right) > 0$ for all $i \in [b]$.

**Remark D.6.** For the RPT case (see Section 4), we have

$$\mathbb{P} \left( i \in \hat{S} \right) = \mathbb{P} \left( s \leq i \right) = \sum_{s=1}^{i} p_s$$

and

$$\mathbb{E} \left[ L_{i,\hat{S}}^\alpha \middle| i \in \hat{S} \right] = \frac{\mathbb{E} \left[ L_{i,\hat{S}}^\alpha \mathbb{I} \left( i \in \hat{S} \right) \right]}{\mathbb{P} \left( i \in \hat{S} \right)} = \frac{\sum_{s=1}^{i} p_s L_{i,\{s,\ldots,b\}}^\alpha}{\sum_{s=1}^{i} p_s}$$

for $\alpha \in \{0, 1\}$. Hence the weights from Theorem D.4 become

$$\frac{\mathbb{P} \left( i \in \hat{S} \right)}{\mathbb{E} \left[ L_{i,\hat{S}}^1 \middle| i \in \hat{S} \right]} = \frac{\left( \sum_{s=1}^{i} p_s \right)^2}{\sum_{s=1}^{i} p_s L_{i,\{s,\ldots,b\}}^\alpha}.$$

Substituting this expression into the rate yields the result in Theorem 4.2.

**Remark D.7.** When $\hat{S} = [b]$ with probability 1, the weights become $w_i = 1/L^1_{i,[b]}$, and hence Theorem D.4 guarantees that

$$\min_{k=0,\ldots,K-1} \sum_{i=1}^{b} \left[ \frac{1/L^1_{i,[b]}}{\frac{1}{b}\sum_{l=1}^{b} 1/L^1_{l,[b]}} \left\| \nabla_i f(X^k) \right\|_{(i)\star} \right] \leq \varepsilon,$$

after

$$K = \left\lceil \frac{2\delta^0 \sum_{i=1}^{b} L^0_{i,[b]}/\left(L^1_{i,[b]}\right)^2}{\varepsilon^2 \left(\frac{1}{b}\sum_{l=1}^{b} 1/L^1_{i,[b]}\right)^2} + \frac{2\delta^0}{\varepsilon \left(\frac{1}{b}\sum_{l=1}^{b} 1/L^1_{i,[b]}\right)} \right\rceil$$

iterations, recovering the rate of Gluon (Riabinin et al., 2025b).

*Proof of Theorem D.4.* Starting with Assumption 4.3, we have

$$f(X^{k+1})$$

$$\leq f(X^k) + \sum_{i \in S^k} \left( \left\langle \nabla_i f(X^k), X_i^{k+1} - X_i^k \right\rangle_{(i)} \right.$$

$$\left. + \frac{L^0_{i,S^k} + L^1_{i,S^k} \left\| \nabla_i f(X^k) \right\|_{(i)\star}}{2} \left\| X_i^{k+1} - X_i^k \right\|_{(i)}^2 \right)$$

$$= f(X^k) + \sum_{i \in S^k} \left( -\gamma_i^k \left\langle \nabla_i f(X^k), \left(\nabla_i f(X^k)\right)^\sharp \right\rangle_{(i)} \right.$$

$$\left. + \frac{L^0_{i,S^k} + L^1_{i,S^k} \left\| \nabla_i f(X^k) \right\|_{(i)\star}}{2} (\gamma_i^k)^2 \left\| \left(\nabla_i f(X^k)\right)^\sharp \right\|_{(i)}^2 \right)$$

$$\overset{(50),(51)}{=} f(X^k) + \sum_{i \in S^k} \left( -\gamma_i^k \left\| \nabla_i f(X^k) \right\|_{(i)\star}^2 \right.$$

$$\left. + \frac{L^0_{i,S^k} + L^1_{i,S^k} \left\| \nabla_i f(X^k) \right\|_{(i)\star}}{2} (\gamma_i^k)^2 \left\| \nabla_i f(X^k) \right\|_{(i)\star}^2 \right).$$

Taking $\gamma_i^k = \frac{1}{L^0_{i,S^k} + L^1_{i,S^k} \left\| \nabla_i f(X^k) \right\|_{(i)\star}}$ gives

$$f(X^{k+1}) \leq f(X^k) - \sum_{i \in S^k} \frac{\left\| \nabla_i f(X^k) \right\|_{(i)\star}^2}{2 \left( L^0_{i,S^k} + L^1_{i,S^k} \left\| \nabla_i f(X^k) \right\|_{(i)\star} \right)},$$

and hence

$$\sum_{k=0}^{K-1} \sum_{i \in S^k} \frac{\left\| \nabla_i f(X^k) \right\|_{(i)\star}^2}{2 \left( L^0_{i,S^k} + L^1_{i,S^k} \left\| \nabla_i f(X^k) \right\|_{(i)\star} \right)} \leq \sum_{k=0}^{K-1} \left( f(X^k) - f(X^{k+1}) \right)$$

$$\leq f(X^0) - f^\star := \delta^0.$$

Taking expectation

$$\delta^0 \geq \sum_{k=0}^{K-1} \mathbb{E} \left[ \sum_{i \in S^k} \frac{\left\| \nabla_i f(X^k) \right\|_{(i)\star}^2}{2 \left( L^0_{i,S^k} + L^1_{i,S^k} \left\| \nabla_i f(X^k) \right\|_{(i)\star} \right)} \right]$$

$$= \sum_{k=0}^{K-1} \mathbb{E} \left[ \mathbb{E} \left[ \sum_{i=1}^{b} \mathbb{I} \left( i \in S^k \right) \frac{\left\| \nabla_i f(X^k) \right\|_{(i)\star}^2}{2 \left( L^0_{i,S^k} + L^1_{i,S^k} \left\| \nabla_i f(X^k) \right\|_{(i)\star} \right)} \,\middle|\, X^k \right] \right]$$

$$= \sum_{k=0}^{K-1} \sum_{i=1}^{b} \mathbb{E}\left[ \mathbb{P}\left(i \in S^k\right) \mathbb{E}\left[ \left. \frac{\left\| \nabla_i f(X^k) \right\|_{(i)\star}^2}{2\left( L_{i,S^k}^0 + L_{i,S^k}^1 \left\| \nabla_i f(X^k) \right\|_{(i)\star} \right)} \right| X^k, \{i \in S^k\} \right] \right]$$

$$\stackrel{(i)}{\geq} \sum_{k=0}^{K-1} \sum_{i=1}^{b} \mathbb{E}\left[ \frac{\left\| \nabla_i f(X^k) \right\|_{(i)\star}^2 \mathbb{P}\left(i \in S^k\right)}{2\left( \mathbb{E}\left[ L_{i,S^k}^0 \,\middle|\, X^k, \{i \in S^k\} \right] + \mathbb{E}\left[ L_{i,S^k}^1 \,\middle|\, X^k, \{i \in S^k\} \right] \left\| \nabla_i f(X^k) \right\|_{(i)\star} \right)} \right]$$

$$\stackrel{(ii)}{=} \frac{1}{2} \sum_{k=0}^{K-1} \sum_{i=1}^{b} \mathbb{E}\left[ \frac{\left\| \nabla_i f(X^k) \right\|_{(i)\star}^2 \mathbb{P}\left(i \in \hat{S}\right)}{\mathbb{E}\left[ L_{i,\hat{S}}^0 \,\middle|\, i \in \hat{S} \right] + \mathbb{E}\left[ L_{i,\hat{S}}^1 \,\middle|\, i \in \hat{S} \right] \left\| \nabla_i f(X^k) \right\|_{(i)\star}} \right]$$

$$\stackrel{(iii)}{\geq} \frac{1}{2} \sum_{k=0}^{K-1} \sum_{i=1}^{b} \frac{\mathbb{E}\left[ \left\| \nabla_i f(X^k) \right\|_{(i)\star} \right]^2 \mathbb{P}\left(i \in \hat{S}\right)}{\mathbb{E}\left[ L_{i,\hat{S}}^0 \,\middle|\, i \in \hat{S} \right] + \mathbb{E}\left[ L_{i,\hat{S}}^1 \,\middle|\, i \in \hat{S} \right] \mathbb{E}\left[ \left\| \nabla_i f(X^k) \right\|_{(i)\star} \right]}, \tag{16}$$

where in $(i)$ we used Jensen's inequality and convexity of the function $t \mapsto \frac{1}{t}$ for $t > 0$, $(ii)$ follows from independence of $\hat{S}$ and $X^k$, and $(iii)$ is a consequence of convexity of the function $t \mapsto \frac{t^2 \mathbb{P}(i \in S^k)}{\mathbb{E}\left[ L_{i,\hat{S}}^0 \middle| i \in \hat{S} \right] + \mathbb{E}\left[ L_{i,\hat{S}}^1 \middle| i \in \hat{S} \right] t}$ and Jensen's inequality.

Now, using Lemma H.2 with $x_i = \sqrt{\mathbb{P}(i \in \hat{S})}/\mathbb{E}\left[ L_{i,\hat{S}}^1 \middle| i \in \hat{S} \right]$, $y_i = \sqrt{\mathbb{P}\left( i \in \hat{S} \right)}\,\mathbb{E}\left[ \left\| \nabla_i f(X^k) \right\|_{(i)\star} \right]$ and $z_i = \mathbb{E}\left[ L_{i,\hat{S}}^0 \,\middle|\, i \in \hat{S} \right] + \mathbb{E}\left[ L_{i,\hat{S}}^1 \,\middle|\, i \in \hat{S} \right] \mathbb{E}\left[ \left\| \nabla_i f(X^k) \right\|_{(i)\star} \right]$, we obtain

$$\sum_{i=1}^{b} \frac{\mathbb{P}\left(i \in \hat{S}\right) \mathbb{E}\left[ \left\| \nabla_i f(X^k) \right\|_{(i)\star} \right]^2}{\mathbb{E}\left[ L_{i,\hat{S}}^0 \,\middle|\, i \in \hat{S} \right] + \mathbb{E}\left[ L_{i,\hat{S}}^1 \,\middle|\, i \in \hat{S} \right] \mathbb{E}\left[ \left\| \nabla_i f(X^k) \right\|_{(i)\star} \right]}$$

$$\geq \frac{\left( \sum_{i=1}^{b} \frac{\mathbb{P}(i \in \hat{S})}{\mathbb{E}\left[ L_{i,\hat{S}}^1 \middle| i \in \hat{S} \right]} \mathbb{E}\left[ \left\| \nabla_i f(X^k) \right\|_{(i)\star} \right] \right)^2}{\sum_{i=1}^{b} \left( \frac{\mathbb{P}(i \in \hat{S})}{\left( \mathbb{E}\left[ L_{i,\hat{S}}^1 \middle| i \in \hat{S} \right] \right)^2} \mathbb{E}\left[ L_{i,\hat{S}}^0 \,\middle|\, i \in \hat{S} \right] + \frac{\mathbb{P}(i \in \hat{S})}{\mathbb{E}\left[ L_{i,\hat{S}}^1 \middle| i \in \hat{S} \right]} \mathbb{E}\left[ \left\| \nabla_i f(X^k) \right\|_{(i)\star} \right] \right)}.$$

Applying this in (16), we get

$$\delta^0 \geq \frac{1}{2} \sum_{k=0}^{K-1} \sum_{i=1}^{b} \frac{\mathbb{E}\left[ \left\| \nabla_i f(X^k) \right\|_{(i)\star} \right]^2 \mathbb{P}\left(i \in \hat{S}\right)}{\mathbb{E}\left[ L_{i,\hat{S}}^0 \,\middle|\, i \in \hat{S} \right] + \mathbb{E}\left[ L_{i,\hat{S}}^1 \,\middle|\, i \in \hat{S} \right] \mathbb{E}\left[ \left\| \nabla_i f(X^k) \right\|_{(i)\star} \right]}$$

$$\geq \frac{1}{2} \sum_{k=0}^{K-1} \frac{\left( \sum_{i=1}^{b} \frac{\mathbb{P}(i \in \hat{S})}{\mathbb{E}\left[ L_{i,\hat{S}}^1 \middle| i \in \hat{S} \right]} \mathbb{E}\left[ \left\| \nabla_i f(X^k) \right\|_{(i)\star} \right] \right)^2}{\sum_{i=1}^{b} \left( \frac{\mathbb{P}(i \in \hat{S})}{\left( \mathbb{E}\left[ L_{i,\hat{S}}^1 \middle| i \in \hat{S} \right] \right)^2} \mathbb{E}\left[ L_{i,\hat{S}}^0 \,\middle|\, i \in \hat{S} \right] + \frac{\mathbb{P}(i \in \hat{S})}{\mathbb{E}\left[ L_{i,\hat{S}}^1 \middle| i \in \hat{S} \right]} \mathbb{E}\left[ \left\| \nabla_i f(X^k) \right\|_{(i)\star} \right] \right)}$$

$$= \frac{1}{2} \sum_{k=0}^{K-1} \psi \left( \sum_{i=1}^{b} \frac{\mathbb{P}\left(i \in \hat{S}\right)}{\mathbb{E}\left[ L_{i,\hat{S}}^1 \,\middle|\, i \in \hat{S} \right]} \mathbb{E}\left[ \left\| \nabla_i f(X^k) \right\|_{(i)\star} \right] \right),$$

where $\psi(t) := \dfrac{t^2}{\sum_{i=1}^{b} \frac{\mathbb{P}(i \in \hat{S})}{\left( \mathbb{E}\left[ L_{i,\hat{S}}^1 \middle| i \in \hat{S} \right] \right)^2} \mathbb{E}\left[ L_{i,\hat{S}}^0 \middle| i \in \hat{S} \right] + t}$. Since $\psi$ is increasing for $t > 0$, we have

$$\delta^0 \geq \frac{1}{2} \sum_{k=0}^{K-1} \psi \left( \sum_{i=1}^{b} \frac{\mathbb{P}\left(i \in \hat{S}\right)}{\mathbb{E}\left[ L_{i,\hat{S}}^1 \,\middle|\, i \in \hat{S} \right]} \mathbb{E}\left[ \left\| \nabla_i f(X^k) \right\|_{(i)\star} \right] \right)$$

$$\geq \quad \frac{K}{2}\psi\left(\min_{k=0,\dots,K-1}\sum_{i=1}^{b}\frac{\mathbb{P}\left(i\in\hat{S}\right)}{\mathbb{E}\left[L_{i,\hat{S}}^{1}\,\middle|\,i\in\hat{S}\right]}\mathbb{E}\left[\left\|\nabla_{i}f(X^k)\right\|_{(i)\star}\right]\right).$$

Moreover, since $\psi$ is monotonic, it has an inverse $\psi^{-1}$. Thus

$$\psi^{-1}\left(\frac{2\delta^0}{K}\right) \quad\geq\quad \min_{k=0,\dots,K-1}\sum_{i=1}^{b}\frac{\mathbb{P}\left(i\in\hat{S}\right)}{\mathbb{E}\left[L_{i,\hat{S}}^{1}\,\middle|\,i\in\hat{S}\right]}\mathbb{E}\left[\left\|\nabla_{i}f(X^k)\right\|_{(i)\star}\right]$$

$$=\quad \min_{k=0,\dots,K-1}\sum_{i=1}^{b}w_{i}\mathbb{E}\left[\left\|\nabla_{i}f(X^k)\right\|_{(i)\star}\right],$$

where $w_i := \frac{\mathbb{P}\left(i\in\hat{S}\right)}{\mathbb{E}\left[L_{i,\hat{S}}^{1}\,\middle|\,i\in\hat{S}\right]}$. This in turn means that to reach the precision

$$\min_{k=0,\dots,K-1}\sum_{i=1}^{b}\left[\frac{w_i}{\frac{1}{b}\sum_{l=1}^{b}w_l}\mathbb{E}\left[\left\|\nabla_{i}f(X^k)\right\|_{(i)\star}\right]\right] \leq \varepsilon,$$

it suffices to run the algorithm for

$$K \quad=\quad \left\lceil\frac{2\delta^0}{\psi\left(\varepsilon\left(\frac{1}{b}\sum_{l=1}^{b}w_l\right)\right)}\right\rceil = \left\lceil\frac{2\delta^0\sum_{i=1}^{b}\frac{\mathbb{P}(i\in\hat{S})\mathbb{E}\left[L_{i,\hat{S}}^{0}\,\middle|\,i\in\hat{S}\right]}{\left(\mathbb{E}\left[L_{i,\hat{S}}^{1}\,\middle|\,i\in\hat{S}\right]\right)^2} + 2\delta^0\left(\varepsilon\left(\frac{1}{b}\sum_{l=1}^{b}w_l\right)\right)}{\left(\varepsilon\left(\frac{1}{b}\sum_{l=1}^{b}w_l\right)\right)^2}\right\rceil$$

$$=\quad \left\lceil\frac{2\delta^0\sum_{i=1}^{b}\frac{\mathbb{P}(i\in\hat{S})\mathbb{E}\left[L_{i,\hat{S}}^{0}\,\middle|\,i\in\hat{S}\right]}{\left(\mathbb{E}\left[L_{i,\hat{S}}^{1}\,\middle|\,i\in\hat{S}\right]\right)^2}}{\varepsilon^2\left(\frac{1}{b}\sum_{l=1}^{b}\frac{\mathbb{P}(l\in\hat{S})}{\mathbb{E}\left[L_{l,\hat{S}}^{1}\,\middle|\,l\in\hat{S}\right]}\right)^2} + \frac{2\delta^0}{\varepsilon\left(\frac{1}{b}\sum_{l=1}^{b}\frac{\mathbb{P}(l\in\hat{S})}{\mathbb{E}\left[L_{l,\hat{S}}^{1}\,\middle|\,l\in\hat{S}\right]}\right)}\right\rceil$$

iterations. $\qquad\square$

## E  OPTIMIZING THE COST − DETERMINISTIC GRADIENT CASE

Let $S^k \subseteq \{1, \ldots, b\}$ be the random subset sampled at iteration $k$, and $s^k := \min S^k$ its smallest index. Recall the per-iteration cost

$$\text{cost}(S^k) = c_{\text{ov}} + \sum_{i=s^k}^{b} c_i + \sum_{i \in S^k} c_i^\sharp.$$

Define the two marginal probabilities

$$F_i := \mathbb{P}\left(\hat{s} \leq i\right), \qquad Q_i := \mathbb{P}\left(i \in \hat{S}\right),$$

where $\hat{s}$ is a random variable following the same distribution as $s^k$ (since $S^k \sim \mathcal{D}$, $k \geq 0$, are i.i.d., the same holds for $s^k$). Since

$$\mathbb{E}\left[\sum_{i=s^k}^{b} c_i\right] = \sum_{j=1}^{b} \mathbb{P}\left(s^k = j\right) \sum_{i=j}^{b} c_i = \sum_{i=1}^{b} c_i \sum_{j=1}^{i} \mathbb{P}\left(s^k = j\right) = \sum_{i=1}^{b} c_i \mathbb{P}\left(s^k \leq i\right)$$

and

$$\mathbb{E}\left[\sum_{i \in S^k} c_i^\sharp\right] = \mathbb{E}\left[\sum_{i=1}^{b} \mathbb{I}\left(i \in S^k\right) c_i^\sharp\right] = \sum_{i=1}^{b} c_i^\sharp \mathbb{P}\left(i \in S^k\right),$$

the expected cost of one iteration is

$$\mathbb{E}\left[\text{cost}(\hat{S})\right] = c_{\text{ov}} + \sum_{i=1}^{b} c_i F_i + \sum_{i=1}^{b} c_i^\sharp Q_i. \tag{17}$$

Hence, to evaluate the expected cost, it suffices to compute the two marginals $F_i$ and $Q_i$ under the chosen sampling scheme.

In the sequel, we describe a few example sampling strategies considered in this work and analyze their costs under the layer-wise smooth setting (see Section E.1) and the generalized layer-wise smooth setting (see Section E.2).

1. **RPT.** Sample $\hat{s} \in \{1, \ldots, b\}$, where $p_i = \mathbb{P}\left(\hat{s} = i\right)$, and set $\hat{S} = \{\hat{s}, \ldots, b\}$. Then

$$F_i = \mathbb{P}\left(\hat{s} \leq i\right) = \sum_{j=1}^{i} p_j, \qquad Q_i = \mathbb{P}\left(i \in \hat{S}\right) = \mathbb{P}\left(\hat{s} \leq i\right) = F_i.$$

Therefore,

$$\mathbb{E}\left[\text{cost}(\hat{S})\right] = c_{\text{ov}} + \sum_{i=1}^{b} (c_i + c_i^\sharp) F_i = c_{\text{ov}} + \sum_{i=1}^{b} (c_i + c_i^\sharp) \sum_{j=1}^{i} p_j.$$

2. **$\tau$-nice sampling.** Choose $\hat{S}$ uniformly from all subsets of $[b]$ of size $\tau$. Then

$$F_i = 1 - \mathbb{P}\left(\hat{s} > i\right) = 1 - \frac{\binom{b-i}{\tau}}{\binom{b}{\tau}},$$

$$Q_i = \frac{\binom{b-1}{\tau-1}}{\binom{b}{\tau}} = \frac{\tau}{b},$$

where we use the convention $\binom{n}{k} = 0$ for $n < k$ (so the formula for $F_i$ automatically gives $F_i = 1$ when $b - i < \tau$). Hence

$$\mathbb{E}\left[\text{cost}(\hat{S})\right] = c_{\text{ov}} + \sum_{i=1}^{b} c_i \left(1 - \frac{\binom{b-i}{\tau}}{\binom{b}{\tau}}\right) + \frac{\tau}{b} \sum_{i=1}^{b} c_i^\sharp.$$

3. $\tau$-**submodel sampling.** Sample a starting index $\hat{s} \in \{1, \ldots, b - \tau + 1\}$ with probability $p_i = \mathbb{P}(\hat{s} = i)$ (where $\tau \in [b]$ is fixed), and set $\hat{S} = \{\hat{s}, \ldots, \hat{s} + \tau - 1\}$ (i.e., a block of $\tau$ consecutive layers). Then the marginals are

$$F_i = \mathbb{P}(\hat{s} \le i) = \sum_{j=1}^{\min\{i, b-\tau+1\}} p_j,$$

$$Q_i = \mathbb{P}(i \in \hat{S}) = \sum_{j=\max\{1, i-\tau+1\}}^{\min\{i, b-\tau+1\}} p_j.$$

The expected per-iteration cost is therefore

$$\mathbb{E}\left[\text{cost}(\hat{S})\right] = c_{\text{ov}} + \sum_{i=1}^{b} c_i \left(\sum_{j=1}^{\min\{i, b-\tau+1\}} p_j\right) + \sum_{i=1}^{b} c_i^{\sharp} \left(\sum_{j=\max\{1, i-\tau+1\}}^{\min\{i, b-\tau+1\}} p_j\right).$$

4. **Arbitrary submodel sampling.** Let $\{B_1, \ldots, B_m\}$ be a partition of $[b]$ into disjoint blocks of arbitrary indices, i.e.,

$$B_1 \cup \cdots \cup B_m = [b], \quad B_k \cap B_l = \emptyset \quad \text{for } k \ne l.$$

At each iteration, pick block $B_i$ with probability $p_i$ (where $\sum_{i=1}^{m} p_i = 1$) and set $\hat{S} = B_i$. For $i \in [b]$, let $k(i)$ denote the unique block with $i \in B_{k(i)}$, and let $\underline{b}_k := \min B_k$. Then

$$F_i = \mathbb{P}(\hat{s} \le i) = \sum_{j: \underline{b}_j \le i} p_j,$$

$$Q_i = \mathbb{P}(i \in \hat{S}) = p_{k(i)}.$$

The expected cost per iteration is

$$\mathbb{E}\left[\text{cost}(\hat{S})\right] = c_{\text{ov}} + \sum_{i=1}^{b} c_i \left(\sum_{k: \underline{b}_k \le i} p_k\right) + \sum_{i=1}^{b} c_i^{\sharp} p_{k(i)}.$$

We now consider the algorithm's performance under the two smoothness regimes.

### E.1 SMOOTH CASE

According to Theorem D.1, under Assumption 4.2, Algorithm 2 run with stepsizes $\gamma_i^k = 1/L_{i, S^k}^0$ guarantees that

$$\left(\min_{i \in [b]} w_i\right) \frac{1}{K} \sum_{k=0}^{K-1} \sum_{i=1}^{b} \mathbb{E}\left[\left\|\nabla_i f(X^k)\right\|_{(i)\star}^2\right] \le \frac{1}{K} \sum_{k=0}^{K-1} \sum_{i=1}^{b} w_i \mathbb{E}\left[\left\|\nabla_i f(X^k)\right\|_{(i)\star}^2\right]$$

$$\le \frac{f(X^0) - f^\star}{K},$$

where $w_i := \mathbb{E}\left[\frac{\mathbb{I}(i \in \hat{S})}{2 L_{i, \hat{S}}^0}\right]$. Thus, to ensure that $\frac{1}{K} \sum_{k=0}^{K-1} \sum_{i=1}^{b} \mathbb{E}\left[\left\|\nabla_i f(X^k)\right\|_{(i)\star}^2\right] \le \varepsilon$, it suffices to run it for

$$K = \left\lceil \frac{f(X^0) - f^\star}{\varepsilon \left(\min_{i \in [b]} w_i\right)} \right\rceil$$

iterations. Now, recall from (17) that the expected cost of a single iteration is

$$\mathbb{E}\left[\text{cost}(S^k)\right] = c_{\text{ov}} + \sum_{i=1}^{b} c_i \mathbb{P}\left(\min S^k \le i\right) + \sum_{i=1}^{b} c_i^{\sharp} \mathbb{P}\left(i \in S^k\right).$$

Hence, the expected cost of the entire optimization procedure can be written as

$$
\begin{aligned}
\text{cost}_\varepsilon(\mathcal{D}) \quad &= \quad K \times \mathbb{E}\left[\text{cost}(\hat{S})\right] \\[2mm]
&= \quad K \times \left( c_{\text{ov}} + \sum_{i=1}^{b} c_i \mathbb{P}\left(\min \hat{S} \le i\right) + \sum_{i=1}^{b} c_i^{\sharp} \mathbb{P}\left(i \in \hat{S}\right) \right) \\[2mm]
&\propto \quad \frac{c_{\text{ov}} + \sum_{i=1}^{b} c_i \mathbb{P}\left(\min \hat{S} \le i\right) + \sum_{i=1}^{b} c_i^{\sharp} \mathbb{P}\left(i \in \hat{S}\right)}{\min_{i \in [b]} \mathbb{E}\left[\frac{\mathbb{I}\left(i \in \hat{S}\right)}{2L_{i,\hat{S}}^0}\right]},
\end{aligned}
$$

and the cost minimization problem to be solved is

$$
\min_{\mathcal{D}: \mathfrak{P}([b]) \to [0,1], \sum_{S \subseteq [b]} \mathcal{D}(S) = 1} \frac{c_{\text{ov}} + \sum_{i=1}^{b} c_i \mathbb{P}\left(\min \hat{S} \le i\right) + \sum_{i=1}^{b} c_i^{\sharp} \mathbb{P}\left(i \in \hat{S}\right)}{\min_{i \in [b]} \mathbb{E}\left[\frac{\mathbb{I}\left(i \in \hat{S}\right)}{2L_{i,\hat{S}}^0}\right]}. \tag{18}
$$

The task above is an optimization over probability distributions on the power set $\mathfrak{P}([b])$, which has dimension $2^b$, making a direct solution intractable for large $b$. Instead of tackling it in full generality, we can restrict $\mathcal{D}$ to some parametric family. For certain such families, the ratio objective simplifies to a linear–fractional program in the cumulative marginals, which has a closed form solution or can be solved efficiently (e.g., via the Dinkelbach algorithm (Dinkelbach, 1967)).

Let us now consider some specific examples, starting with the procedure considered in the main part of this paper.

### E.1.1 RANDOMIZED PROGRESSIVE TRAINING

Sample $\hat{s} \in \{1, \dots, b\}$, where $p_i = \mathbb{P}(\hat{s} = i)$, and set $\hat{S} = \{\hat{s}, \dots, b\}$.

We first consider the randomized progressive training setting introduced in Section 4. Under this sampling strategy, we have

$$
F_i = \mathbb{P}(\hat{s} \le i) = \sum_{j=1}^{i} p_j, \qquad Q_i = \mathbb{P}\left(i \in \hat{S}\right) = \mathbb{P}(\hat{s} \le i) = F_i,
$$

and hence

$$
\mathbb{E}\left[\text{cost}(\hat{S})\right] = c_{\text{ov}} + \sum_{i=1}^{b} (c_i + c_i^{\sharp}) F_i = c_{\text{ov}} + \sum_{i=1}^{b} (c_i + c_i^{\sharp}) \sum_{j=1}^{i} p_j.
$$

Combining it with the fact that

$$
\mathbb{E}\left[\frac{\mathbb{I}\left(i \in \hat{S}\right)}{2L_{i,\hat{S}}^0}\right] = \sum_{s=1}^{b} \frac{\mathbb{I}(i \in \{s, \dots, b\})}{2L_{i,\{s,\dots,b\}}^0} p_s = \sum_{s=1}^{i} \frac{p_s}{2L_{i,\{s,\dots,b\}}^0},
$$

we get

$$
\text{cost}_\varepsilon(\mathcal{D}) \propto \frac{\mathbb{E}\left[\text{cost}(\hat{S})\right]}{\min_{i \in [b]} \mathbb{E}\left[\frac{\mathbb{I}\left(i \in \hat{S}\right)}{2L_{i,\hat{S}}^0}\right]} = \frac{c_{\text{ov}} + \sum_{i=1}^{b} (c_i + c_i^{\sharp}) \sum_{j=1}^{i} p_j}{\min_{i \in [b]} \left\{ \sum_{s=1}^{i} \frac{p_s}{2L_{i,\{s,\dots,b\}}^0} \right\}}. \tag{19}
$$

**Optimal probabilities.** First, note that the numerator can be rewritten as

$$
c_{\text{ov}} + \sum_{i=1}^{b} (c_i + c_i^{\sharp}) \sum_{j=1}^{i} p_j = c_{\text{ov}} + \sum_{j=1}^{b} \left[ \sum_{i \ge j} (c_i + c_i^{\sharp}) \right] p_j = \sum_{j=1}^{b} \left[ c_{\text{ov}} + \sum_{i \ge j} (c_i + c_i^{\sharp}) \right] p_j,
$$

where the second equality follows from $\sum_{j=1}^{b} p_j = 1$. Denote

$$d_j := c_{\mathrm{ov}} + \sum_{i \geq j} (c_i + c_i^{\sharp})$$

for $j \in [b]$, and let

$$\delta_{i,s} := \frac{1}{2L_{i,\{s,\dots,b\}}}$$

for $i \in [b]$ and $s \leq i$. Clearly

$$d_1 > d_2 > \dots > d_b \tag{20}$$

and

$$\delta_{i,1} \leq \delta_{i,2} \leq \dots \leq \delta_{i,i} \qquad \forall i \in [b]. \tag{21}$$

Based on (19), the search for optimal probabilities reduces to solving the following linear fractional program:

$$
\begin{aligned}
\min_{p,t} \quad & \frac{d_1 p_1 + \dots + d_b p_b}{t} \\
\text{s.t.} \quad & p_1, \dots, p_b \geq 0 \\
& p_1 + \dots + p_b = 1 \\
& t \geq 0 \\
& t \leq \delta_{1,1} p_1 \\
& \quad \vdots \\
& t \leq \delta_{i,1} p_1 + \dots + \delta_{i,i} p_i \\
& \quad \vdots \\
& t \leq \delta_{b,1} p_1 + \dots + \delta_{b,b} p_b.
\end{aligned}
\tag{22}
$$

This program can be written equivalently as

$$
\begin{aligned}
\min_{q} \quad & d_1 q_1 + \dots + d_b q_b \\
\text{s.t.} \quad & q_1, \dots, q_b \geq 0 \\
& \delta_{1,1} q_1 \geq 1 \\
& \quad \vdots \\
& \delta_{i,1} q_1 + \dots + \delta_{i,i} q_i \geq 1 \\
& \quad \vdots \\
& \delta_{b,1} q_1 + \dots + \delta_{b,b} q_b \geq 1.
\end{aligned}
\tag{23}
$$

Based on that, we can derive a recursive prescription for the optimal probabilities.

First, we show that if $\sum_{s=1}^{i} \delta_{i,s} q_s > 1$ and $q_i > 0$ for some $i$, then one can shift a small amount of mass from $q_i$ to $q_{i+1}$ and strictly reduce the objective, without violating any constraints.

**Lemma E.1.** Let $q$ be an optimal point of (23) and fix $i \in [b-1]$. If $\sum_{s=1}^{i} \delta_{i,s} q_s > 1$, then $q_i = 0$. Equivalently,

$$q_i > 0 \quad \implies \quad \sum_{s=1}^{i} \delta_{i,s} q_s = 1. \tag{24}$$

*Proof.* Suppose that $i$ is such that $\sum_{s=1}^{i} \delta_{i,s} q_s > 1$, but $q_i > 0$. Then there exists $\varepsilon > 0$ such that $q_i \geq \varepsilon$ and

$$0 < \varepsilon \leq \frac{\sum_{s=1}^{i} \delta_{i,s} q_s - 1}{\delta_{i,i}}.$$

Define

$$\tilde{q}_i = q_i - \varepsilon, \quad \tilde{q}_{i+1} = q_{i+1} + \varepsilon, \quad \tilde{q}_s = q_s \text{ for } s \notin \{i, i+1\}.$$

Then $\{\tilde{q}_i\}_{i \in [b]}$ satisfy all constraints:

- Constraint $k < i$: $\sum_{s=1}^{k} \delta_{k,s} \tilde{q}_s = \sum_{s=1}^{k} \delta_{k,s} q_s \geq 1$,

- Constraint $i$: $\sum_{s=1}^{i} \delta_{i,s} \tilde{q}_s = \sum_{s=1}^{i} \delta_{i,s} q_s - \delta_{i,i} \varepsilon \geq 1$ by the choice of $\varepsilon$,

- Constraint $k > i$: $\sum_{s=1}^{k} \delta_{k,s} \tilde{q}_s = \sum_{s=1}^{k} \delta_{k,s} q_s + \varepsilon(\delta_{k,i+1} - \delta_{k,i}) \geq \sum_{s=1}^{k} \delta_{k,s} q_s \geq 1$
  using the monotonicity $\delta_{k,i} \leq \delta_{k,i+1}$.

At the same time, the value of the objective decreases, as $\sum_j d_j \tilde{q}_j - \sum_j d_j q_j = (d_{i+1} - d_i)\varepsilon < 0$, contradicting optimality of $q$. Therefore, we must have $q_i = 0$. $\qquad \square$

Let us now use Lemma E.1 to derive a recursive construction of the optimal probabilities. Let $q^\star = (q_1^\star, \ldots, q_b^\star)$ be a solution of (23). As established in (24), any positive coordinate forces its constraint to be tight. Constraint 1 is $\delta_{1,1} q_1^\star \geq 1$, meaning that $q_1^\star > 0$, and so $\delta_{1,1} q_1^\star = 1$. Thus

$$q_1^\star = \frac{1}{\delta_{1,1}} = 2L_{1,\{1,\ldots,b\}}.$$

Now, suppose that $q_1^\star, \ldots, q_{i-1}^\star$ have already been determined. Then, constraint $i$ reads

$$\sum_{s=1}^{i-1} \delta_{i,s} q_s^\star + \delta_{i,i} q_i^\star \geq 1.$$

Define the residual

$$r_i^\star := 1 - \sum_{s=1}^{i-1} \delta_{i,s} q_s^\star = 1 - \sum_{s=1}^{i-1} \frac{q_s^\star}{2L_{i,\{s,\ldots,b\}}}.$$

If $r_i^\star \leq 0$, then $(q_1^\star, \ldots, q_{i-1}^\star)$ already satisfy the constraint, so by (24), we must have $q_i^\star = 0$. If $r_i^\star > 0$, then for $(q_1^\star, \ldots, q_i^\star)$ to satisfy the constraint, we need $q_i^\star > 0$, and hence by Lemma E.1, we have $\sum_{s=1}^{i} \delta_{i,s} q_s^\star = 1$, meaning that

$$q_i^\star = \frac{r_i^\star}{\delta_{i,i}} = 2r_i^\star L_{i,\{i,\ldots,b\}}.$$

Combining the above yields the recursion

$$q_1^\star = 2L_{1,\{1,\ldots,b\}},$$

$$q_i^\star = 2[r_i^\star]_+ L_{i,\{i,\ldots,b\}}, \qquad r_i^\star = 1 - \sum_{s=1}^{i-1} \frac{q_s^\star}{2L_{i,\{s,\ldots,b\}}}, \quad i \in \{2, \ldots, b\},$$

where $[x]_+ := \max\{x, 0\}$. The optimal probabilities are finally recovered by normalization

$$p_i^\star = \frac{q_i^\star}{\sum_{j=1}^{b} q_j^\star}.$$

**Remark E.1.** Note that $(p_1^\star, p_2^\star, \ldots, p_b^\star) = (1, 0, \ldots, 0)$ if and only if $q_1^\star = \sum_{j=1}^{b} q_j^\star$, i.e., $q_2^\star = \ldots = q_b^\star = 0$. But for that to be the case, we need $r_i^\star \leq 0$ for all $i \in \{2, \ldots, b\}$, so

$$1 \leq \sum_{s=1}^{i-1} \delta_{i,s} q_s^\star = \delta_{i,1} q_1^\star = \frac{\delta_{i,1}}{\delta_{1,1}} = \frac{L_{1,\{1,\ldots,b\}}}{L_{i,\{1,\ldots,b\}}} \qquad \forall i \in \{2, \ldots, b\}.$$

Therefore, choosing $p_1 = 1$ is optimal if and only if

$$L_{1,\{1,\ldots,b\}} = \max_{i \in [b]} L_{i,\{1,\ldots,b\}},$$

which proves Theorem 4.3.

### E.1.2 $\tau$-NICE SAMPLING

Choose $\hat{S}$ uniformly from all subsets of $[b]$ of size $\tau$.

Every $\tau$-subset has probability $\binom{b}{\tau}^{-1}$. Thus

$$
\mathbb{E}\left[ \frac{\mathbb{I}\left( i \in \hat{S} \right)}{2L^0_{i,\hat{S}}} \right] = \frac{1}{\binom{b}{\tau}} \sum_{\substack{S \subseteq [b] \\ |S| = \tau}} \frac{\mathbb{I}\left( i \in S \right)}{2L^0_{i,S}},
$$

and hence

$$
\mathrm{cost}_\varepsilon(\mathcal{D}) \quad \propto \quad \frac{\mathbb{E}\left[ \mathrm{cost}(\hat{S}) \right]}{\min_{i \in [b]} \mathbb{E}\left[ \frac{\mathbb{I}(i \in \hat{S})}{2L^0_{i,\hat{S}}} \right]} = \frac{c_{\mathrm{ov}} + \sum_{j=1}^{b} c_j \left( 1 - \frac{\binom{b-j}{\tau}}{\binom{b}{\tau}} \right) + \frac{\tau}{b} \sum_{j=1}^{b} c_j^\sharp}{\min_{i \in [b]} \frac{1}{\binom{b}{\tau}} \sum_{\substack{S \subseteq [b] \\ |S| = \tau}} \frac{\mathbb{I}(i \in S)}{2L^0_{i,S}}}
$$

$$
= \frac{c_{\mathrm{ov}} \binom{b}{\tau} + \sum_{j=1}^{b} c_j \left( \binom{b}{\tau} - \binom{b-j}{\tau} \right) + \binom{b}{\tau} \frac{\tau}{b} \sum_{j=1}^{b} c_j^\sharp}{\min_{i \in [b]} \left\{ \sum_{\substack{S \subseteq [b] \\ |S| = \tau}} \frac{\mathbb{I}(i \in S)}{2L^0_{i,S}} \right\}}.
$$

In general there is no closed-form expression for $\tau$ minimizing this cost because the denominator depends on $\{L^0_{i,S}\}_{S \subseteq [b]}$ in a highly problem-specific way. That said, it can be shown that $\tau = b$ need not be optimal (indeed, it can be very sub-optimal). For $\tau = b$, we always have $S = [b]$, and the cost becomes

$$
\mathrm{cost}_\varepsilon(\mathcal{D}) \quad \propto \quad \frac{c_{\mathrm{ov}} + \sum_{j=1}^{b} c_j \left( 1 - \binom{b-j}{b} \right) + \sum_{j=1}^{b} c_j^\sharp}{\min_{i \in [b]} \frac{1}{2L^0_{i,[b]}}}
$$

$$
= \quad 2 \max_{i \in [b]} L^0_{i,[b]} \left( c_{\mathrm{ov}} + \sum_{j=1}^{b} \left( c_j + c_j^\sharp \right) \right)
$$

(recall that we use the convention $\binom{n}{k} = 0$ for $n < k$).

Now, let us make a simplifying assumption that $L^0_{i,S}$ depends only on $i$ and $|S|$ and define $L^0_{i,\tau} := L^0_{i,S}$ for $|S| = \tau$. Then

$$
\mathrm{cost}_\varepsilon(\mathcal{D}) \quad \propto \quad \frac{c_{\mathrm{ov}} \binom{b}{\tau} + \sum_{j=1}^{b} c_j \left( \binom{b}{\tau} - \binom{b-j}{\tau} \right) + \binom{b}{\tau} \frac{\tau}{b} \sum_{j=1}^{b} c_j^\sharp}{\min_{i \in [b]} \left\{ \sum_{\substack{S \subseteq [b] \\ |S| = \tau}} \frac{\mathbb{I}(i \in S)}{2L^0_{i,\tau}} \right\}}
$$

$$
= \quad \frac{c_{\mathrm{ov}} \binom{b}{\tau} + \sum_{j=1}^{b} c_j \left( \binom{b}{\tau} - \binom{b-j}{\tau} \right) + \binom{b}{\tau} \frac{\tau}{b} \sum_{j=1}^{b} c_j^\sharp}{\min_{i \in [b]} \left\{ \frac{1}{2L^0_{i,\tau}} \binom{b-1}{\tau-1} \right\}}
$$

$$
= \quad 2 \max_{i \in [b]} L^0_{i,\tau} \left( \frac{b}{\tau} c_{\mathrm{ov}} + \sum_{j=1}^{b} c_j \left( \frac{b}{\tau} - \frac{\binom{b-j}{\tau}}{\binom{b-1}{\tau-1}} \right) + \sum_{j=1}^{b} c_j^\sharp \right).
$$

Define

$$
A(\tau) := \max_{i \in [b]} L^0_{i,\tau}, \qquad B(\tau) := \frac{b}{\tau} c_{\mathrm{ov}} + \sum_{j=1}^{b} c_j \left( \frac{b}{\tau} - \frac{\binom{b-j}{\tau}}{\binom{b-1}{\tau-1}} \right) + \sum_{j=1}^{b} c_j^\sharp.
$$

Then, the objective function to be minimized is

$$
f(\tau) := A(\tau) B(\tau).
$$

By definition, $\tau = b$ is optimal if and only if $f(b) \leq f(\tau)$ for all $\tau \in \{1, \ldots, b-1\}$.

First, we show that $B$ is decreasing in $\tau$. To this end, let $1 \leq \tau_1 < \tau_2 \leq b$ and consider the difference

$$
B(\tau_1) - B(\tau_2) = \left(\frac{b}{\tau_1} - \frac{b}{\tau_2}\right) c_{\text{ov}} + \sum_{j=1}^{b} c_j \left(\frac{b}{\tau_1} - \frac{\binom{b-j}{\tau_1}}{\binom{b-1}{\tau_1-1}} - \frac{b}{\tau_2} + \frac{\binom{b-j}{\tau_2}}{\binom{b-1}{\tau_2-1}}\right)
$$

$$
= \left(\frac{b}{\tau_1} - \frac{b}{\tau_2}\right) c_{\text{ov}} + \sum_{j=1}^{b} c_j \left(h_j(\tau_1) - h_j(\tau_2)\right),
$$

where $h_j(\tau) := \frac{b}{\tau} - \frac{\binom{b-j}{\tau}}{\binom{b-1}{\tau-1}}$. Clearly, the first term is positive. Let us focus on the second term. Using Pascal's identity, we have

$$
h_{j+1}(\tau) - h_j(\tau) = \frac{\binom{b-j}{\tau} - \binom{b-j-1}{\tau}}{\binom{b-1}{\tau-1}} = \frac{\binom{b-j-1}{\tau-1}}{\binom{b-1}{\tau-1}}.
$$

Therefore,

$$
h_j(\tau) = h_j(\tau) - h_0(\tau) = \sum_{m=0}^{j-1} \left(h_{m+1}(\tau) - h_m(\tau)\right) = \sum_{m=0}^{j-1} \frac{\binom{b-m-1}{\tau-1}}{\binom{b-1}{\tau-1}},
$$

where

$$
\frac{\binom{b-m-1}{\tau-1}}{\binom{b-1}{\tau-1}} = \frac{b-\tau}{b-m-\tau} \frac{\binom{b-m-1}{\tau}}{\binom{b-1}{\tau}} \geq \frac{\binom{b-m-1}{\tau}}{\binom{b-1}{\tau}},
$$

and the inequality is strict for $m > 0$. It follows that for any $j \in [b]$

$$
h_j(\tau) = \sum_{m=0}^{j-1} \frac{\binom{b-m-1}{\tau-1}}{\binom{b-1}{\tau-1}} \geq \sum_{m=0}^{j-1} \frac{\binom{b-m-1}{\tau}}{\binom{b-1}{\tau}} = h_j(\tau+1).
$$

Thus, except for the trivial case when $j = 1$ (where $h_1(\tau) \equiv 1$), the function $h_j(\tau)$ is strictly decreasing in $\tau$, implying that $h_j(\tau_1) - h_j(\tau_2) > 0$, which proves that $B(\tau_1) > B(\tau_2)$. Thus

$$
B(\tau) \geq B(b) \quad \forall \tau \leq b,
$$

with strict inequality if $c_{\text{ov}} > 0$ and $\tau < b$.

Now, note that in general larger $\tau$ leads to a higher Lipschitz constant, and hence one may expect $A$ to be an increasing function of $\tau$. If $A(b)$ is much larger than $A(\tau)$, it may compensate for the decrease in $B(\tau)$, resulting in $f(\tau) = A(\tau)B(\tau) < A(b)B(b) = f(b)$, and so $\tau = b$ may not be optimal.

As an example, suppose that $L_{i,\tau}^0$ scales linearly with $\tau$, i.e, $A(\tau) = \max_{i \in [b]} \tau L_i^0$ for some $L_i^0 \geq 0$. Then

$$
\text{cost}_\varepsilon(\mathcal{D}) \propto \max_{i \in [b]} L_i^0 \left(bc_{\text{ov}} + \sum_{j=1}^{b} c_j \left(b - \tau \frac{\binom{b-j}{\tau}}{\binom{b-1}{\tau-1}}\right) + \tau \sum_{j=1}^{b} c_j^\sharp\right)
$$

$$
= \max_{i \in [b]} L_i^0 \left(bc_{\text{ov}} + b\sum_{j=1}^{b} c_j + \tau \sum_{j=1}^{b} \left(c_j^\sharp - c_j \frac{\binom{b-j}{\tau}}{\binom{b-1}{\tau-1}}\right)\right).
$$

Now, consider the function

$$
\Phi(\tau) := \tau \sum_{j=1}^{b} \left(c_j^\sharp - c_j \frac{\binom{b-j}{\tau}}{\binom{b-1}{\tau-1}}\right).
$$

Note that

$$
\begin{aligned}
\Phi(\tau+1) - \Phi(\tau) &= (\tau+1)\sum_{j=1}^{b}\left(c_j^{\sharp} - c_j\frac{\binom{b-j}{\tau+1}}{\binom{b-1}{\tau}}\right) - \tau\sum_{j=1}^{b}\left(c_j^{\sharp} - c_j\frac{\binom{b-j}{\tau}}{\binom{b-1}{\tau-1}}\right) \\
&= \sum_{j=1}^{b}c_j\left(\tau\frac{\binom{b-j}{\tau}}{\binom{b-1}{\tau-1}} - (\tau+1)\frac{\binom{b-j}{\tau+1}}{\binom{b-1}{\tau}}\right) + \sum_{j=1}^{b}c_j^{\sharp} \\
&= \sum_{j=1}^{b}c_j\frac{j(b-j)!(b-\tau-1)!}{(b-1)!(b-j-\tau)!} + \sum_{j=1}^{b}c_j^{\sharp} \geq 0
\end{aligned}
$$

(with the convention that the right-hand side is 0 when $b - j - \tau < 0$). Moreover, the increment is strictly positive if either $\sum_{j=1}^{b}c_j^{\sharp} > 0$ or there exists $j$ with $c_j > 0$ and $b - j - \tau \geq 0$. Hence $\Phi(\tau)$ is non-decreasing in $\tau$, and strictly increasing whenever one of these conditions holds, and the optimal choice is $\tau^{\star} = 1$.

### E.1.3 $\tau$-SUBMODEL SAMPLING

Sample a starting index $\hat{s} \in \{1,\ldots,b-\tau+1\}$ with probability $p_i = \mathbb{P}(\hat{s} = i)$ and set $\hat{S} = \{\hat{s},\ldots,\hat{s}+\tau-1\}$.

The denominator is

$$
\mathbb{E}\left[\frac{\mathbb{I}\left(i\in\hat{S}\right)}{2L_{i,\hat{S}}^{0}}\right] = \sum_{j=1}^{b-\tau+1}p_j\frac{\mathbb{I}\left(i\in\{j,\ldots,j+\tau-1\}\right)}{2L_{i,\{j,\ldots,j+\tau-1\}}^{0}} = \sum_{j=\max\{1,i-\tau+1\}}^{\min\{i,b-\tau+1\}}\frac{p_j}{2L_{i,\{j,\ldots,j+\tau-1\}}^{0}}.
$$

Hence the total expected cost is proportional to

$$
\begin{aligned}
\mathrm{cost}_{\varepsilon}(\mathcal{D}) &\propto \frac{\mathbb{E}\left[\mathrm{cost}(\hat{S})\right]}{\min_{i\in[b]}\mathbb{E}\left[\frac{\mathbb{I}(i\in\hat{S})}{2L_{i,\hat{S}}^{0}}\right]} \\
&= \frac{c_{\mathrm{ov}} + \sum_{j=1}^{b}c_j\left(\sum_{i=1}^{\min\{j,b-\tau+1\}}p_i\right) + \sum_{j=1}^{b}c_j^{\sharp}\left(\sum_{i=\max\{1,j-\tau+1\}}^{\min\{j,b-\tau+1\}}p_i\right)}{\min_{i\in[b]}\left\{\sum_{j=\max\{1,i-\tau+1\}}^{\min\{i,b-\tau+1\}}\frac{p_j}{2L_{i,\{j,\ldots,j+\tau-1\}}^{0}}\right\}}.
\end{aligned}
$$

**Partitioned $\tau$-submodel sampling.** Let us consider a special case of the above sampling scheme where the submodels assigned non-zero probability *partition* $[b]$. For simplicity, suppose that $b$ is divisible by $\tau$, let $m = b/\tau$, and define the block start indices via

$$
s_k = (k-1)\tau+1, \qquad k = 1,\ldots,m.
$$

The algorithm then picks block $B_k := \{s_k,\ldots,s_k+\tau-1\}$ with probability $p_{s_k} > 0$ (where $\sum_{k=1}^{m}p_{s_k} = 1$). This is equivalent to the submodel sampling with starting-index distribution satisfying

$$
p_{s_k} = p_{(k-1)\tau+1} > 0 \quad (k = 1,\ldots,m), \qquad p_j = 0 \text{ otherwise.}
$$

Plugging this choice into the general submodel expressions immediately gives

$$
F_i = \sum_{j=1}^{\min\{i,b-\tau+1\}}p_j = \sum_{k=1}^{\min\{m,\lfloor(i-1)/\tau\rfloor+1\}}p_{s_k},
$$

$$
Q_i = \sum_{j=\max\{1,i-\tau+1\}}^{\min\{i,b-\tau+1\}}p_j = p_{s_{\lceil i/\tau\rceil}},
$$

$$
\mathbb{E}\left[\frac{\mathbb{I}\left(i\in\hat{S}\right)}{2L_{i,\hat{S}}^{0}}\right] = \sum_{j=\max\{1,i-\tau+1\}}^{\min\{i,b-\tau+1\}}\frac{p_j}{2L_{i,\{j,\ldots,j+\tau-1\}}^{0}} = \frac{p_{s_{\lceil i/\tau\rceil}}}{2L_{i,B_{\lceil i/\tau\rceil}}^{0}}.
$$

Therefore the full cost reduces to

$$
\text{cost}_\varepsilon(\mathcal{D}) \quad \propto \quad \frac{c_{\text{ov}} + \sum_{j=1}^{b} c_j \left( \sum_{k=1}^{\min\{m, \lfloor (j-1)/\tau \rfloor + 1\}} p_{s_k} \right) + \sum_{j=1}^{b} c_j^\sharp p_{s_{\lceil j/\tau \rceil}}}{\min_{i \in [b]} \left\{ \frac{p_{s_{\lceil i/\tau \rceil}}}{2 L_{i, B_{\lceil i/\tau \rceil}}^0} \right\}}
$$

$$
= \quad \frac{c_{\text{ov}} + \sum_{j=1}^{b} c_j \left( \sum_{k=1}^{\lceil j/\tau \rceil} p_{s_k} \right) + \sum_{j=1}^{b} c_j^\sharp p_{s_{\lceil j/\tau \rceil}}}{\min_{i \in [b]} \left\{ \frac{p_{s_{\lceil i/\tau \rceil}}}{2 L_{i, B_{\lceil i/\tau \rceil}}^0} \right\}}. \tag{25}
$$

Now, note that

$$
\sum_{j=1}^{b} c_j \left( \sum_{k=1}^{\lceil j/\tau \rceil} p_{s_k} \right) = \sum_{j=1}^{\tau} c_j p_{s_1} + \sum_{j=\tau+1}^{2\tau} c_j \left( p_{s_1} + p_{s_2} \right) + \ldots + \sum_{j=(m-1)\tau+1}^{m\tau} c_j \left( p_{s_1} + \ldots + p_{s_m} \right)
$$

$$
= p_{s_1} \sum_{j=1}^{\tau} c_j + \left( p_{s_1} + p_{s_2} \right) \sum_{j=\tau+1}^{2\tau} c_j + \ldots + \left( p_{s_1} + \ldots + p_{s_m} \right) \sum_{j=(m-1)\tau+1}^{m\tau} c_j
$$

$$
= p_{s_1} \sum_{j=1}^{m\tau} c_j + p_{s_2} \sum_{j=\tau+1}^{m\tau} c_j + \ldots + p_{s_m} \sum_{j=(m-1)\tau+1}^{m\tau} c_j
$$

and

$$
\sum_{j=1}^{b} c_j^\sharp p_{s_{\lceil j/\tau \rceil}} = \sum_{j=1}^{\tau} c_j^\sharp p_{s_1} + \sum_{j=\tau+1}^{2\tau} c_j^\sharp p_{s_2} + \ldots + \sum_{j=(m-1)\tau+1}^{m\tau} c_j^\sharp p_{s_m}.
$$

Hence

$$
\text{cost}_\varepsilon(\mathcal{D})
$$

$$
\propto \frac{1}{\min_{i \in [b]} \left\{ \frac{p_{s_{\lceil i/\tau \rceil}}}{2 L_{i, B_{\lceil i/\tau \rceil}}^0} \right\}} \left[ c_{\text{ov}} + p_{s_1} \left( \sum_{j=1}^{m\tau} c_j + \sum_{j=1}^{\tau} c_j^\sharp \right) + p_{s_2} \left( \sum_{j=\tau+1}^{m\tau} c_j + \sum_{j=\tau+1}^{2\tau} c_j^\sharp \right) + \ldots \right.
$$

$$
\left. + p_{s_m} \left( \sum_{j=(m-1)\tau+1}^{m\tau} c_j + \sum_{j=(m-1)\tau+1}^{m\tau} c_j^\sharp \right) \right]
$$

$$
= \frac{p_{s_1} d_1 + p_{s_2} d_2 + \ldots + p_{s_m} d_m}{\min_{i \in [b]} \left\{ \frac{p_{s_{\lceil i/\tau \rceil}}}{2 L_{i, B_{\lceil i/\tau \rceil}}^0} \right\}},
$$

where we used the fact that $\sum_{j=1}^{m} p_{s_j} = 1$ and denoted $d_i := c_{\text{ov}} + \sum_{j=(i-1)\tau+1}^{m\tau} c_j + \sum_{j=(i-1)\tau+1}^{i\tau} c_j^\sharp$.

**Optimal probabilities for fixed $\tau$.** We follow an approach similar to that in Section E.1.1 to find the optimal probabilities for a fixed choice of $\tau$. For $i \in B_k$, define

$$
\delta_{i,k} := \frac{1}{2 L_{i, B_k}^0}
$$

and set

$$
\delta_k^{\min} := \min_{i \in B_k} \delta_{i,k} = \frac{1}{2 \max_{i \in B_k} L_{i, B_k}^0}.
$$

Then the expected cost can be represented as

$$\text{cost}_\varepsilon(\mathcal{D}) \propto \frac{d_1 p_{s_1} + \cdots + d_m p_{s_m}}{\min_{i \in [b]} \left\{ \frac{p_{s_{\lceil i/\tau \rceil}}}{2 L^0_{i,B_{\lceil i/\tau \rceil}}} \right\}} = \frac{d_1 p_{s_1} + \cdots + d_m p_{s_m}}{\min_{k \in [m]} \min_{i \in B_k} \{ \delta_{i,k} p_{s_k} \}}.$$

This corresponds to the linear fractional program

$$
\begin{aligned}
\min_{p,t} \quad & \frac{d_1 p_{s_1} + \cdots + d_m p_{s_m}}{t} \\
\text{s.t.} \quad & p_{s_1}, \ldots, p_{s_m} \geq 0, \\
& p_{s_1} + \cdots + p_{s_m} = 1, \\
& t \geq 0, \\
& t \leq \delta_{i,k} p_{s_k}, \qquad k \in [m], i \in B_k.
\end{aligned}
\tag{26}
$$

The standard change of variables $q_k = p_{s_k}/t$ yields the equivalent linear program

$$
\begin{aligned}
\min_{q} \quad & d_1 q_1 + \cdots + d_m q_m \\
\text{s.t.} \quad & q_1, \ldots, q_m \geq 0, \\
& \delta_{i,k} q_k \geq 1 \qquad k \in [m], i \in B_k.
\end{aligned}
\tag{27}
$$

Because each block $k$ only appears in constraints of the form $\delta_{i,k} q_k \geq 1$ (for $i \in B_k$), the constraints for block $k$ reduce to the single constraint

$$\delta_k^{\min} q_k \geq 1 \quad \Longleftrightarrow \quad q_k \geq \frac{1}{\delta_k^{\min}}.$$

Hence (27) is separable and its optimal solution is

$$q_k^\star = \frac{1}{\delta_k^{\min}}, \qquad k \in [m],$$

with optimal objective value

$$\sum_{k=1}^m d_k q_k^\star = \sum_{k=1}^m \frac{d_k}{\delta_k^{\min}}.$$

Now, let us introduce dual variables $\lambda_i \geq 0$ for the constraints $\delta_{i,k} q_k \geq 1$, $i \in B_k$. The dual of (27) is

$$
\begin{aligned}
\max_{\lambda} \quad & \lambda_1 + \ldots + \lambda_b \\
\text{s.t.} \quad & \lambda_1, \ldots, \lambda_b \geq 0, \\
& d_k \geq \sum_{i \in B_k} \lambda_i \delta_{i,k}, \qquad k \in [m].
\end{aligned}
$$

To certify optimality of $q^\star$, for each block $k$, choose an index $i_k \in B_k$ attaining the minimum $\delta_k^{\min}$ and set

$$\lambda_{i_k} = \frac{d_k}{\delta_{i_k,k}}, \qquad \lambda_i = 0 \text{ for } i \notin \{i_1, \ldots, i_m\}.$$

Then for each block $k$,

$$\sum_{i \in B_k} \delta_{i,k} \lambda_i = \delta_{i_k,k} \lambda_{i_k} = d_k,$$

so the dual constraints hold with equality and the dual objective equals

$$\sum_{i=1}^b \lambda_i = \sum_{k=1}^m \frac{d_k}{\delta_k^{\min}},$$

matching the primal objective. By strong duality $q^\star$ is optimal.

From $q_k^\star = 1/\delta_k^{\min}$ and $p_{s_k} = tq_k = q_k/\sum_{l=1}^m q_l$ we obtain the optimal block probabilities

$$p_{s_k}^\star = \frac{1/\delta_k^{\min}}{\sum_{l=1}^m 1/\delta_l^{\min}} = \frac{\max_{i \in B_k} L_{i,B_k}^0}{\sum_{l=1}^m \max_{i \in B_l} L_{i,B_l}^0}.$$

so that each block's probability is proportional to the worst-case local smoothness constant inside that block. For this choice, the minimal expected cost is

$$\text{cost}_\varepsilon(\mathcal{D}) \propto \sum_{k=1}^m \frac{d_k}{\delta_k^{\min}} = 2\sum_{k=1}^m d_k \max_{i \in B_k} L_{i,B_k}^0.$$

**Choosing $\tau$.** We now show that the cost is not necessarily minimized by choosing $\tau = b$. To this end, we want to minimize the function

$$\Phi(\tau) := 2\sum_{k=1}^m d_k(\tau) \max_{i \in B_k} L_{i,B_k}^0$$

(where we explicitly emphasize the dependence of $d_k$ on $\tau$).

To gain intuition about which extreme ($\tau = 1$ or $\tau = b$) may be preferable, assume that the costs are constant, i.e.,

$$c_j \equiv c, \qquad c_j^\sharp \equiv c^\sharp,$$

in which case

$$\sum_{k=1}^m d_k(\tau) = \sum_{k=1}^m \left( c_{\text{ov}} + \sum_{j=(k-1)\tau+1}^{m\tau} c + \sum_{j=(k-1)\tau+1}^{k\tau} c^\sharp \right)$$

$$= \sum_{k=1}^m \left( c_{\text{ov}} + c\left( m\tau - (k-1)\tau \right) + c^\sharp \left( k\tau - (k-1)\tau \right) \right)$$

$$= mc_{\text{ov}} + c\tau \frac{m(m+1)}{2} + m\tau c^\sharp.$$

Suppose in addition that the worst-case local smoothness per block does not depend on the block index, i.e.,

$$\max_{i \in B_k} L_{i,B_k}^0 \equiv L^0(\tau) \quad \forall k \in [m]$$

for some non-decreasing function $L$. Then

$$\Phi(\tau) = 2L^0(\tau) \sum_{k=1}^m d_k(\tau) = 2L^0(\tau) \left( \frac{b}{\tau} \left( c_{\text{ov}} + \frac{bc}{2} \right) + \frac{bc}{2} + bc^\sharp \right).$$

Thus, the $\tau$-dependence of $\Phi$ is the product of a non-decreasing factor $L_0(\tau)$ and a factor that decreases like $1/\tau$ plus additive constants. Consequently:

- If $L^0(\tau)$ is constant in $\tau$ (no worsening with larger blocks), then $\Phi(\tau)$ is strictly decreasing in $\tau$ and the minimizer is $\tau^\star = b$.
- If $L^0(\tau)$ grows sublinearly, then $L^0(\tau)\frac{b}{\tau}\left( c_{\text{ov}} + \frac{bc}{2} \right)$ is decreasing in $\tau$, while $L^0(\tau)\left( \frac{cb}{2} + bc^\sharp \right)$ is increasing. Hence, the optimal $\tau$ may lie strictly between 1 and $b$, depending on the relative magnitudes of the costs.
- If $L^0(\tau)$ increases at least linearly in $\tau$, then $\Phi(\tau)$ is increasing in $\tau$, and hence $\tau^\star = 1$.

### E.1.4 ARBITRARY SUBMODEL SAMPLING

Let $\{B_1, \ldots, B_m\}$ be a partition of $[b]$. Set $\hat{S} = B_i$ with probability $p_i$ (where $\sum_{i=1}^{m} p_i = 1$). For $j \in [b]$, $k(j)$ denotes the unique block with $j \in B_{k(j)}$ and $\underline{b}_k := \min B_k$.

First, note that the cost can be expressed block-wise as

$$
\mathbb{E}\left[\text{cost}(\hat{S})\right] = c_{\text{ov}} + \sum_{i=1}^{b} c_i \left(\sum_{k:\underline{b}_k \leq i} p_k\right) + \sum_{i=1}^{b} c_i^{\sharp} p_{k(i)}
$$

$$
= c_{\text{ov}} + \sum_{k=1}^{m} \sum_{i \geq \underline{b}_k} c_i p_k + \sum_{k=1}^{m} \sum_{i \in B_k} c_i^{\sharp} p_k
$$

$$
= \sum_{k=1}^{m} d_k p_k,
$$

where $d_k := c_{\text{ov}} + \sum_{j \geq \underline{b}_k} c_j + \sum_{j \in B_k} c_j^{\sharp}$. We also have

$$
\mathbb{E}\left[\frac{\mathbb{I}\left(i \in \hat{S}\right)}{2L_{i,\hat{S}}^0}\right] = \frac{p_{k(i)}}{2L_{i,B_{k(i)}}^0},
$$

and hence

$$
\text{cost}_\varepsilon(\mathcal{D}) \propto \frac{\mathbb{E}\left[\text{cost}(\hat{S})\right]}{\min_{i \in [b]} \mathbb{E}\left[\frac{\mathbb{I}\left(i \in \hat{S}\right)}{2L_{i,\hat{S}}^0}\right]} = \frac{\sum_{k=1}^{m} d_k p_k}{\min_{k \in [m]} \min_{i \in B_k} \left\{\frac{p_k}{2L_{i,B_k}^0}\right\}}.
$$

Now, define

$$
\delta_{i,k} := \frac{1}{2L_{i,B_k}^0}, \qquad \delta_k^{\min} := \min_{i \in B_k} \delta_{i,k}.
$$

By the same linear fractional reduction as in (26), the unique optimal solution is

$$
p_k^\star = \frac{1/\delta_k^{\min}}{\sum_{l=1}^{m} 1/\delta_l^{\min}} = \frac{\max_{i \in B_k} L_{i,B_k}^0}{\sum_{l=1}^{m} \max_{i \in B_l} L_{i,B_l}^0}.
$$

With this choice, the objective is

$$
\text{cost}_\varepsilon(\mathcal{D}) \quad \propto \quad 2 \sum_{k=1}^{m} d_k \max_{i \in B_k} L_{i,B_k}^0 = 2 \sum_{k=1}^{m} \left(c_{\text{ov}} + \sum_{j \geq \underline{b}_k} c_j + \sum_{j \in B_k} c_j^{\sharp}\right) \max_{i \in B_k} L_{i,B_k}^0. \quad (28)
$$

Let us compare it with the cost for full model training

$$
\mathbb{E}\left[\text{cost}_{\text{full}}(K)\right] \propto 2 \max_{i \in [b]} L_{i,[b]}^0 \left(c_{\text{ov}} + \sum_{j=1}^{b} \left(c_j + c_j^{\sharp}\right)\right).
$$

**Example 1** (Grouping layers by cost similarity). Suppose that the layers are partitioned into groups according to their similarity, so that within each block, all per-layer costs are (approximately) the same. Concretely, for every $k \in [m]$ and every $j \in B_k$ assume that

$$
c_j \equiv \underline{c}_k, \qquad c_j^{\sharp} \equiv \underline{c}_k^{\sharp}.
$$

Then

$$
\text{cost}_\varepsilon(\mathcal{D}) \quad \propto \quad 2 \sum_{k=1}^{m} \left(c_{\text{ov}} + \sum_{j \geq \underline{b}_k} c_j + \sum_{j \in B_k} c_j^{\sharp}\right) \max_{i \in B_k} L_{i,B_k}^0
$$

$$= 2 \sum_{k=1}^{m} \left( c_{\mathrm{ov}} + \sum_{j \geq \underline{b}_k} c_j + |B_k| \underline{c}_k^{\sharp} \right) \max_{i \in B_k} L_{i,B_k}^0$$

and

$$\mathbb{E}\left[ \mathrm{cost}_{\mathrm{full}}(K) \right] \propto 2 \max_{i \in [b]} L_{i,[b]}^0 \left( c_{\mathrm{ov}} + \sum_{l=1}^{m} |B_l| \left( \underline{c}_l + \underline{c}_l^{\sharp} \right) \right).$$

Therefore, partitioned arbitrary submodel sampling with the optimal probabilities is better than full model training if and only if

$$\sum_{k=1}^{m} \max_{i \in B_k} L_{i,B_k}^0 \left( c_{\mathrm{ov}} + \sum_{j \geq \underline{b}_k} c_j + |B_k| \underline{c}_k^{\sharp} \right) \leq \max_{i \in [b]} L_{i,[b]}^0 \left( c_{\mathrm{ov}} + \sum_{k=1}^{m} |B_k| \left( \underline{c}_k + \underline{c}_k^{\sharp} \right) \right). \quad (29)$$

If each block is much better conditioned than the full model, i.e. $\max_{i \in B_k} L_{i,B_k}^0 \ll \max_{i \in [b]} L_{i,[b]}^0$ for all $k$, then the left side of (29) can be much smaller than the right side and partitioned submodel sampling will be advantageous even if the block-wise tail sums are moderately large. Indeed, if $\max_{i \in B_k} L_{i,B_k}^0 \approx \frac{1}{m} \max_{i \in [b]} L_{i,[b]}^0$, then

$$\sum_{k=1}^{m} \max_{i \in B_k} L_{i,B_k}^0 \left( c_{\mathrm{ov}} + \sum_{j \geq \underline{b}_k} c_j + |B_k| \underline{c}_k^{\sharp} \right)$$

$$\approx \frac{1}{m} \max_{i \in [b]} L_{i,[b]}^0 \sum_{k=1}^{m} \left( c_{\mathrm{ov}} + \sum_{j \geq \underline{b}_k} c_j + |B_k| \underline{c}_k^{\sharp} \right)$$

$$= \max_{i \in [b]} L_{i,[b]}^0 \left( c_{\mathrm{ov}} + \frac{1}{m} \sum_{k=1}^{m} \sum_{j \geq \underline{b}_k} c_j + \frac{1}{m} \sum_{k=1}^{m} |B_k| \underline{c}_k^{\sharp} \right),$$

which improves upon full model training if $\frac{1}{m} \sum_{k=1}^{m} \sum_{j \geq \underline{b}_k} c_j + \frac{1}{m} \sum_{k=1}^{m} |B_k| \underline{c}_k^{\sharp} \leq \sum_{k=1}^{m} |B_k| \left( \underline{c}_k + \underline{c}_k^{\sharp} \right)$.

**Example 2** (Grouping layers across transformer blocks). In their general forms, (28) and (29) are hard to interpret. Let us consider a simplified language model motivating example behind considering this sampling strategy. Suppose the network is composed of $T$ identical transformer blocks, each containing $L$ layers, so that $b = TL$, with the natural block structure: layer index $j \in [b]$ corresponds to position $l \in \{1, \ldots, L\}$ inside transformer block $t \in [T]$ via $j = l + (t-1)L$.

Consider the partition that groups together *same-position* layers across transformer blocks:

$$B_l = \{l, l + L, l + 2L, \ldots, l + (T-1)L\}, \qquad \forall l \in [L],$$

so $m = L$ and $|B_l| = T$ for every $l$. For each block $B_l$ we have $\underline{b}_l = \min B_l = l$.

In this setting, for block $B_l$ the tail sum that appears in $d_l$ simplifies to

$$\sum_{j \geq \underline{b}_l} c_j = \sum_{j=l}^{b} c_j.$$

Note that we grouped the layers is such a way that the layers within each block are of the same type, and hence we may expect the costs to be roughly the same within each block. That is, we assume that

$$c_{l+tL} \equiv \underline{c}_l, \qquad c_{l+tL}^{\sharp} \equiv \underline{c}_l^{\sharp}, \qquad \forall l \in [L], \ t \in \{0, \ldots, T-1\},$$

in which case the tail sum further simplifies to

$$\sum_{j=l}^{b} c_j = \sum_{j=l}^{L} \underline{c}_j + (T-1) \sum_{j=1}^{L} \underline{c}_j.$$

Hence, under the periodic costs assumption the block-wise cost constants become

$$d_l = c_{\text{ov}} + \sum_{j \geq \underline{b}_l} c_j + \sum_{j \in B_l} c_j^{\sharp} = c_{\text{ov}} + \sum_{j=l}^{L} \underline{c}_j + (T-1) \sum_{j=1}^{L} \underline{c}_j + T \underline{c}_l^{\sharp}.$$

If the local smoothness within a block are also approximately homogeneous, i.e.,

$$L_{i,B_l}^0 \equiv L_l^0 \qquad \forall i \in B_l,$$

then

$$\max_{i \in B_l} L_{i,B_l}^0 = L_l^0,$$

and the previously derived expected cost (for the optimal block probabilities) reduces to

$$\text{cost}_{\varepsilon}(\mathcal{D}) \propto 2 \sum_{l=1}^{L} L_l^0 \left( c_{\text{ov}} + \sum_{j=l}^{L} \underline{c}_j + (T-1) \sum_{j=1}^{L} \underline{c}_j + T \underline{c}_l^{\sharp} \right).$$

Under the same periodicity assumptions, the full-model expected cost is

$$\mathbb{E}\left[\text{cost}_{\text{full}}(K)\right] \propto 2 \max_{i \in [b]} L_{i,[b]}^0 \left( c_{\text{ov}} + T \sum_{j=1}^{L} \underline{c}_j + T \sum_{j=1}^{L} \underline{c}_j^{\sharp} \right).$$

Comparing the two, partitioned sampling is better if

$$\sum_{l=1}^{L} L_l^0 \left( c_{\text{ov}} + \sum_{j=l}^{L} \underline{c}_j + (T-1) \sum_{j=1}^{L} \underline{c}_j + T \underline{c}_l^{\sharp} \right) \leq \max_{i \in [b]} L_{i,[b]}^0 \left( c_{\text{ov}} + T \sum_{j=1}^{L} \underline{c}_j + T \sum_{j=1}^{L} \underline{c}_j^{\sharp} \right). \tag{30}$$

Now, let us denote

$$\underline{C} := \sum_{j=1}^{L} \underline{c}_j, \qquad \underline{C}^{\sharp} := \sum_{j=1}^{L} \underline{c}_j^{\sharp}.$$

Then (30) is equivalent to

$$\sum_{l=1}^{L} L_l^0 \left( c_{\text{ov}} + \sum_{j=l}^{L} \underline{c}_j \right) + T \sum_{l=1}^{L} L_l^0 \left( \frac{T-1}{T} \underline{C} + \underline{c}_l^{\sharp} \right) \leq T \left( \underline{C} + \underline{C}^{\sharp} \right) \max_{i \in [b]} L_{i,[b]}^0 + c_{\text{ov}} \max_{i \in [b]} L_{i,[b]}^0. \tag{31}$$

For large $T$, the dominant terms are those multiplied by $T$. Then, the leading-order criterion becomes

$$\sum_{l=1}^{L} L_l^0 \left( \underline{C} + \underline{c}_l^{\sharp} \right) \leq \max_{i \in [b]} L_{i,[b]}^0 (\underline{C} + \underline{C}^{\sharp}), \tag{32}$$

which can hold when $L_l^0 \ll \max_{i \in [b]} L_{i,[b]}^0$.

### E.2 GENERALIZED SMOOTH CASE

According to Theorem D.4, under Assumption 4.3, Algorithm 2 run with stepsizes $\gamma_i^k = \left( L_{i,S^k}^0 + L_{i,S^k}^1 \left\| \nabla_i f(X^k) \right\|_{(i)\star} \right)^{-1}$ guarantees that

$$\min_{k=0,\dots,K-1} \sum_{i=1}^{b} \left[ \frac{w_i}{\frac{1}{b} \sum_{l=1}^{b} w_l} \mathbb{E}\left[ \left\| \nabla_i f(X^k) \right\|_{(i)\star} \right] \right] \leq \varepsilon, \tag{33}$$

after

$$K = \left\lceil \frac{2\delta^0 \sum_{i=1}^{b} \frac{\mathbb{P}(i\in\hat{S})\mathbb{E}\left[\left. L_{i,\hat{S}}^0 \right| i\in\hat{S}\right]}{\left(\mathbb{E}\left[\left. L_{i,\hat{S}}^1 \right| i\in\hat{S}\right]\right)^2}}{\varepsilon^2 \left(\frac{1}{b}\sum_{l=1}^{b} \frac{\mathbb{P}(l\in\hat{S})}{\mathbb{E}\left[\left. L_{l,\hat{S}}^1 \right| l\in\hat{S}\right]}\right)^2} + \frac{2\delta^0}{\varepsilon\left(\frac{1}{b}\sum_{l=1}^{b} \frac{\mathbb{P}(l\in\hat{S})}{\mathbb{E}\left[\left. L_{l,\hat{S}}^1 \right| l\in\hat{S}\right]}\right)} \right\rceil \tag{34}$$

iterations, where $\delta^0 := f(X^0) - f^\star$ and $w_i := \frac{\mathbb{P}(i\in\hat{S})}{\mathbb{E}\left[\left. L_{i,\hat{S}}^1 \right| i\in\hat{S}\right]}$. To obtain a guarantee on an unweighted gradient sum, note that

$$\begin{aligned}
\underline{\varepsilon} &\geq \min_{k=0,\ldots,K-1} \sum_{i=1}^{b} \left[\frac{w_i}{\frac{1}{b}\sum_{l=1}^{b} w_l} \mathbb{E}\left[\left\|\nabla_i f(X^k)\right\|_{(i)\star}\right]\right] \\
&\geq \frac{\min_{i\in[b]} w_i}{\frac{1}{b}\sum_{l=1}^{b} w_l} \min_{k=0,\ldots,K-1} \sum_{i=1}^{b} \mathbb{E}\left[\left\|\nabla_i f(X^k)\right\|_{(i)\star}\right],
\end{aligned}$$

and hence, substituting in (33) and (34), we have

$$\min_{k=0,\ldots,K-1} \sum_{i=1}^{b} \mathbb{E}\left[\left\|\nabla_i f(X^k)\right\|_{(i)\star}\right] \leq \frac{\underline{\varepsilon}\frac{1}{b}\sum_{l=1}^{b} w_l}{\min_{i\in[b]} w_i} := \varepsilon$$

after

$$K = \left\lceil \frac{2\delta^0 \sum_{i=1}^{b} \frac{\mathbb{P}(i\in\hat{S})\mathbb{E}\left[\left. L_{i,\hat{S}}^0 \right| i\in\hat{S}\right]}{\left(\mathbb{E}\left[\left. L_{i,\hat{S}}^1 \right| i\in\hat{S}\right]\right)^2}}{\varepsilon^2 \min_{i\in[b]}\left(\frac{\mathbb{P}(i\in\hat{S})}{\mathbb{E}\left[\left. L_{i,\hat{S}}^1 \right| i\in\hat{S}\right]}\right)^2} + \frac{2\delta^0}{\varepsilon \min_{i\in[b]} \frac{\mathbb{P}(i\in\hat{S})}{\mathbb{E}\left[\left. L_{i,\hat{S}}^1 \right| i\in\hat{S}\right]}} \right\rceil$$

iterations. Moreover, recall from (17) that the expected cost of a single iteration is

$$\mathbb{E}\left[\text{cost}(\hat{S})\right] = c_{\text{ov}} + \sum_{i=1}^{b} c_i F_i + \sum_{i=1}^{b} c_i^\sharp Q_i.$$

Hence, using the fact that

$$\mathbb{E}\left[\left. L_{i,\hat{S}}^\alpha \right| i \in \hat{S}\right] = \frac{\mathbb{E}\left[L_{i,\hat{S}}^\alpha \mathbb{I}\left(i \in \hat{S}\right)\right]}{\mathbb{P}\left(i \in \hat{S}\right)}$$

for $\alpha \in \{0, 1\}$, the expected cost of the entire optimization procedure is

$$\text{cost}_\varepsilon(\mathcal{D}) = K \times \mathbb{E}\left[\text{cost}(\hat{S})\right]$$

$$= \left\lceil \frac{2\delta^0 \sum_{i=1}^{b} \frac{\mathbb{P}(i\in\hat{S})\mathbb{E}\left[\left. L_{i,\hat{S}}^0 \right| i\in\hat{S}\right]}{\left(\mathbb{E}\left[\left. L_{i,\hat{S}}^1 \right| i\in\hat{S}\right]\right)^2}}{\varepsilon^2 \min_{i\in[b]}\left(\frac{\mathbb{P}(i\in\hat{S})}{\mathbb{E}\left[\left. L_{i,\hat{S}}^1 \right| i\in\hat{S}\right]}\right)^2} + \frac{2\delta^0}{\varepsilon \min_{i\in[b]} \frac{\mathbb{P}(i\in\hat{S})}{\mathbb{E}\left[\left. L_{i,\hat{S}}^1 \right| i\in\hat{S}\right]}} \right\rceil \left(c_{\text{ov}} + \sum_{i=1}^{b} c_i F_i + \sum_{i=1}^{b} c_i^\sharp Q_i\right)$$

$$= \left\lceil \frac{2\delta^0 \sum_{i=1}^{b} \frac{\mathbb{P}(i\in\hat{S})^2 \mathbb{E}\left[L_{i,\hat{S}}^0 \mathbb{I}(i\in\hat{S})\right]}{\left(\mathbb{E}\left[L_{i,\hat{S}}^1 \mathbb{I}(i\in\hat{S})\right]\right)^2}}{\varepsilon^2 \min_{i\in[b]}\left(\frac{\mathbb{P}(i\in\hat{S})^2}{\mathbb{E}\left[L_{i,\hat{S}}^1 \mathbb{I}(i\in\hat{S})\right]}\right)^2} + \frac{2\delta^0}{\varepsilon \min_{i\in[b]} \frac{\mathbb{P}(i\in\hat{S})^2}{\mathbb{E}\left[L_{i,\hat{S}}^1 \mathbb{I}(i\in\hat{S})\right]}} \right\rceil \left(c_{\text{ov}} + \sum_{i=1}^{b} c_i F_i + \sum_{i=1}^{b} c_i^\sharp Q_i\right).$$

We will consider two regimes.

**(1) The $\mathcal{O}\left(1/\varepsilon^2\right)$ term dominates.** Then the problem to be solved is

$$\min_{\mathcal{D}:\mathfrak{P}([b])\to[0,1],\sum_{S\subseteq[b]}\mathcal{D}(S)=1} \underbrace{\frac{\sum_{i=1}^{b}\frac{\mathbb{P}(i\in\hat{S})^2\mathbb{E}\left[L^0_{i,\hat{S}}\mathbb{I}(i\in\hat{S})\right]}{\left(\mathbb{E}\left[L^1_{i,\hat{S}}\mathbb{I}(i\in\hat{S})\right]\right)^2}}{\min_{i\in[b]}\left(\frac{\mathbb{P}(i\in\hat{S})^2}{\mathbb{E}\left[L^1_{i,\hat{S}}\mathbb{I}(i\in\hat{S})\right]}\right)^2}\left(c_{\mathrm{ov}}+\sum_{i=1}^{b}c_iF_i+\sum_{i=1}^{b}c_i^\sharp Q_i\right)}_{\propto\,\mathbb{E}\left[\mathrm{cost}_{\varepsilon^2}(K)\right]}. \tag{35}$$

**(2) The $\mathcal{O}\left(1/\varepsilon\right)$ term dominates.** Then the problem to be solved is

$$\min_{\mathcal{D}:\mathfrak{P}([b])\to[0,1],\sum_{S\subseteq[b]}\mathcal{D}(S)=1} \underbrace{\frac{1}{\min_{i\in[b]}\frac{\mathbb{P}(i\in\hat{S})^2}{\mathbb{E}\left[L^1_{i,\hat{S}}\mathbb{I}(i\in\hat{S})\right]}}\left(c_{\mathrm{ov}}+\sum_{i=1}^{b}c_iF_i+\sum_{i=1}^{b}c_i^\sharp Q_i\right)}_{\propto\,\mathbb{E}\left[\mathrm{cost}_{\varepsilon}(K)\right]}. \tag{36}$$

As in Section E.1, both tasks above involve optimization over probability distributions, which is intractable in general. Again, for certain parametric families, the objective simplifies to a linear-fractional program, which can be solved efficiently. Let us consider some specific examples.

### E.2.1 RANDOMIZED PROGRESSIVE TRAINING

Sample $\hat{s}\in\{1,\ldots,b\}$, where $p_i=\mathbb{P}\left(\hat{s}=i\right)$, and set $\hat{S}=\{\hat{s},\ldots,b\}$.

In this case, $\hat{S}=\{s,\ldots,b\}$ with probability $p_s$ for all $s\in[b]$, and hence

$$\mathbb{P}\left(i\in\hat{S}\right)=\mathbb{P}\left(s\le i\right)=\sum_{s=1}^{i}p_s,$$

and

$$\mathbb{E}\left[L^\alpha_{i,\hat{S}}\mathbb{I}\left(i\in\hat{S}\right)\right]=\sum_{s=1}^{i}p_sL^\alpha_{i,\{s,\ldots,b\}}$$

for $\alpha\in\{0,1\}$. Moreover, following the same steps as in Section E.1.1,

$$\mathbb{E}\left[\mathrm{cost}(\hat{S})\right]=c_{\mathrm{ov}}+\sum_{j=1}^{b}(c_j+c_j^\sharp)\sum_{i=1}^{j}p_i=\sum_{i=1}^{b}\left[c_{\mathrm{ov}}+\sum_{j\ge i}(c_j+c_j^\sharp)\right]p_i.$$

Hence, substituting into (35) and (36), the respective optimization problems reduce to minimizing

$$\mathbb{E}\left[\mathrm{cost}_{\varepsilon^2}(K)\right]\quad\propto\quad\frac{\sum_{i=1}^{b}\frac{\mathbb{P}(i\in\hat{S})^2\mathbb{E}\left[L^0_{i,\hat{S}}\mathbb{I}(i\in\hat{S})\right]}{\left(\mathbb{E}\left[L^1_{i,\hat{S}}\mathbb{I}(i\in\hat{S})\right]\right)^2}}{\min_{i\in[b]}\left(\frac{\mathbb{P}(i\in\hat{S})^2}{\mathbb{E}\left[L^1_{i,\hat{S}}\mathbb{I}(i\in\hat{S})\right]}\right)^2}\left(\sum_{i=1}^{b}\left[c_{\mathrm{ov}}+\sum_{j\ge i}(c_j+c_j^\sharp)\right]p_i\right)$$

$$=\quad\frac{\sum_{i=1}^{b}\frac{\left(\sum_{s=1}^{i}p_s\right)^2\sum_{s=1}^{i}p_sL^0_{i,\{s,\ldots,b\}}}{\left(\sum_{s=1}^{i}p_sL^1_{i,\{s,\ldots,b\}}\right)^2}}{\min_{i\in[b]}\left(\frac{\left(\sum_{s=1}^{i}p_s\right)^2}{\sum_{s=1}^{i}p_sL^1_{i,\{s,\ldots,b\}}}\right)^2}\left(\sum_{i=1}^{b}\left[c_{\mathrm{ov}}+\sum_{j\ge i}(c_j+c_j^\sharp)\right]p_i\right)$$

and

$$\mathbb{E}\left[\mathrm{cost}_{\varepsilon}(K)\right]\quad\propto\quad\frac{1}{\min_{i\in[b]}\frac{\mathbb{P}(i\in\hat{S})^2}{\mathbb{E}\left[L^1_{i,\hat{S}}\mathbb{I}(i\in\hat{S})\right]}}\left(\sum_{i=1}^{b}\left[c_{\mathrm{ov}}+\sum_{j\ge i}(c_j+c_j^\sharp)\right]p_i\right)$$

$$= \frac{1}{\min_{i \in [b]} \frac{\left(\sum_{s=1}^{i} p_s\right)^2}{\sum_{s=1}^{i} p_s L_{i,\{s,\ldots,b\}}^1}} \left( \sum_{i=1}^{b} \left[ c_{\mathrm{ov}} + \sum_{j \geq i} (c_j + c_j^\sharp) \right] p_i \right).$$

Let us first focus on $\mathbb{E}\left[\mathrm{cost}_\varepsilon(K)\right]$, following a similar reasoning to that in Section E.1.1. Denote $d_i := c_{\mathrm{ov}} + \sum_{j \geq i} (c_j + c_j^\sharp)$ for $i \in [b]$ and consider the linear fractional program

$$
\begin{aligned}
\min_{p,t} \quad & \frac{d_1 p_1 + \cdots + d_b p_b}{t} \\
\text{s.t.} \quad & p_1, \ldots, p_b \geq 0 \\
& p_1 + \cdots + p_b = 1 \\
& t \geq 0
\end{aligned}
\tag{37}
$$

$$
t \leq \frac{\left(\sum_{s=1}^{i} p_s\right)^2}{\sum_{s=1}^{i} p_s L_{i,\{s,\ldots,b\}}^1}, \quad i \in [b].
$$

This program can be written equivalently as

$$
\begin{aligned}
\min_{q} \quad & d_1 q_1 + \cdots + d_b q_b \\
\text{s.t.} \quad & q_1, \ldots, q_b \geq 0 \\
& g_i(q) := \left( \sum_{s=1}^{i} q_s \right)^2 - \sum_{s=1}^{i} q_s L_{i,\{s,\ldots,b\}}^1 \geq 0, \quad i \in [b].
\end{aligned}
\tag{38}
$$

We now write the KKT conditions for (38). Introduce multipliers $\lambda_i \geq 0$ for the constraints $g_i(q) \geq 0$ and multipliers $\eta_k \geq 0$ for the non-negativity constraints $q_k \geq 0$. The Lagrangian is

$$
\mathcal{L}(q, \lambda, s) = \sum_{i=1}^{b} d_i q_i - \sum_{i=1}^{b} \lambda_i g_i(q) - \sum_{k=1}^{b} \eta_k q_k.
$$

Fix $k \in [b]$. Differentiating $g_i$ with respect to $q_k$ yields

$$
\frac{\partial g_i(q)}{\partial q_k} = \begin{cases} 2 \sum_{s=1}^{i} q_s - L_{i,\{k,\ldots,b\}}^1, & k \leq i, \\ 0, & k > i. \end{cases}
\tag{39}
$$

Thus, stationarity $\nabla_q \mathcal{L}(q, \lambda, s) = 0$ yields, component-wise for $k \in [b]$,

$$
0 = d_k - \sum_{i=k}^{b} \lambda_i \left( 2 \sum_{s=1}^{i} q_s - L_{i,\{k,\ldots,b\}}^1 \right) - \eta_k,
\tag{40}
$$

and complementarity and sign conditions are

$$
\lambda_i g_i(q) = 0, \qquad \eta_i q_i = 0, \qquad \lambda_i, \eta_i \geq 0 \qquad i \in [b].
$$

**Lemma E.2.** Let $q^\star = (q_1^\star, \ldots, q_p^\star)$ be a local minimizer of (38) and let $\mathrm{supp}(q^\star) := \{k : q_k^\star > 0\}$. Then, for every index $k \in \mathrm{supp}(q^\star)$, there exists at least one index $i$ with $i \geq k$ such that $\lambda_i > 0$. In particular, not all $\lambda_i$ can be zero.

*Proof.* Fix $k \in \mathrm{supp}(q^\star)$. If $\lambda_i = 0$ for all $i \geq k$, then (40) at $k$ reduces to

$$0 = d_k - \eta_k.$$

By complementary slackness $\eta_k q_k^\star = 0$, and since $q_k^\star > 0$, we must have $\eta_k = 0$, so $d_k = 0$ as well. But by definition $d_k = c_{\mathrm{ov}} + \sum_{j \geq k} (c_j + c_j^\sharp)$, which is (under the model assumptions on the costs) strictly positive. Hence there must exist $i \geq k$ with $\lambda_i > 0$. In particular not all $\lambda_i$ are zero. $\square$

**Lemma E.3.** If $\text{supp}(q) = \{k\}$, then

$$q_k \geq \max_{i \in \{k,\ldots,b\}} L^1_{i,\{k,\ldots,b\}}.$$

*Proof.* When $q_j = 0$ for $j \neq k$, we have

$$\sum_{s=1}^{i} q_s = \begin{cases} 0 & i < k, \\ q_k & i \geq k. \end{cases}$$

For $i < k$, trivially $g_i(q) = 0$. For $i \geq k$, we have

$$g_i(q) = q_k^2 - q_k L^1_{i,\{k,\ldots,b\}} = q_k \left( q_k - L^1_{i,\{k,\ldots,b\}} \right).$$

Since $q_k > 0$, the constraint $g_i(q) \geq 0$ is equivalent to $q_k \geq L^1_{i,\{k,\ldots,b\}}$. This must hold for every $i \geq k$, and hence $q_k$ is at least the stated maximum. $\qquad\square$

**Lemma E.4.** Let $q^\star$ be a KKT point of (38) such that $\text{supp}(q^\star) = \{k\}$. Then

$$q_k^\star = \max_{i \geq k} L^1_{i,\{k,\ldots,b\}}.$$

*Proof.* By Lemma E.3 we already have $q_k^\star \geq \max_{i \geq k} L^1_{i,\{k,\ldots,b\}}$. Assume that the inequality is strict, that is,

$$q_k^\star > \max_{i \geq k} L^1_{i,\{k,\ldots,b\}}.$$

Then for every $i \geq k$ we have

$$g_i(q^\star) = q_k^\star \left( q_k^\star - L^1_{i,\{k,\ldots,b\}} \right) > 0.$$

Since by complementary slackness $\lambda_i g_i(q^\star) = 0$, we get $\lambda_i = 0$ for all $i \geq k$. Plugging this into the stationarity condition (40) yields

$$0 = d_k - \eta_k.$$

But $q_k^\star > 0$ forces $\eta_k = 0$, and thus $d_k = 0$, yielding a contradiction. Therefore, the strict inequality cannot hold, and we must have

$$q_k^\star = \max_{i \geq k} L^1_{i,\{k,\ldots,b\}}.$$

$\qquad\square$

**Theorem E.2.** Unless

$$L^1_{1,[b]} = \max_{i \in [b]} L^1_{i,[b]},$$

$(p_1, p_2, \ldots, p_b) = (1, 0, \ldots, 0)$ is *not* an optimal solution of (37).

*Proof.* According to Remark D.5, the sampling must be such that

$$\mathbb{P}\left( i \in \hat{S} \right) = \sum_{s=1}^{i} p_s > 0 \qquad \forall i \in [b],$$

and hence we must have $p_1 > 0$. Thus, the support of the solution $p^\star$ of (37) (which coincides with the support of the solution $q^\star$ of (38)) can only be a singleton if $\text{supp}(p^\star) = \text{supp}(q^\star) = \{1\}$. But then, by Lemma E.4, we must have

$$q_1^\star = \max_{i \in [b]} L^1_{i,[b]}, \quad q_2^\star = \ldots = q_p^\star = 0,$$

which in turn implies that

$$0 \leq g_i(q^\star) = \left( \sum_{s=1}^{i} q_s^\star \right)^2 - \sum_{s=1}^{i} q_s^\star L_{i,\{s,\ldots,b\}}^1 = q_1^\star \left( q_1^\star - L_{i,[b]}^1 \right)$$

for all $i \in [b]$, so in particular $q_1^\star \left( q_1^\star - L_{1,[b]}^1 \right) \geq 0$. Now, suppose $L_{1,[b]}^1 \neq \max_{i \in [b]} L_{i,[b]}^1$. Then the inequality is strict and, by complementary slackness, $\lambda_1 = \eta_1 = 0$, so

$$d_1 = \sum_{i=2}^{b} \lambda_i \left( 2q_1^\star - L_{i,\{1,\ldots,b\}}^1 \right) \leq \sum_{i=2}^{b} \lambda_i \left( 2q_1^\star - L_{i,\{2,\ldots,b\}}^1 \right) = d_2 - \eta_2 \leq d_2,$$

which is a contradiction. Thus, similar to the result in Remark E.1, we must have $L_{1,[b]}^1 = \max_{i \in [b]} L_{i,[b]}^1$. □

### E.2.2 $\tau$-NICE SAMPLING

Choose $S$ uniformly from all subsets of $[b]$ of size $\tau$.

For every $i \in [b]$ we have

$$\mathbb{P} \left( i \in \hat{S} \right) = \frac{\binom{b-1}{\tau-1}}{\binom{b}{\tau}} = \frac{\tau}{b}$$

and

$$\mathbb{E} \left[ L_{i,\hat{S}}^{\alpha} \mathbb{I} \left( i \in \hat{S} \right) \right] = \frac{1}{\binom{b}{\tau}} \sum_{\substack{S \subseteq [b] \\ |S| = \tau}} L_{i,S}^{\alpha} \mathbb{I} \left( i \in S \right).$$

for $\alpha \in \{0, 1\}$. Recall that

$$\mathbb{E} \left[ \mathrm{cost}(S) \right] = c_{\mathrm{ov}} + \sum_{j=1}^{b} c_j \left( 1 - \frac{\binom{b-j}{\tau}}{\binom{b}{\tau}} \right) + \frac{\tau}{b} \sum_{j=1}^{b} c_j^{\sharp}.$$

Hence, substituting into (35) and (36), the objective functions to minimize are

$$\mathbb{E} \left[ \mathrm{cost}_{\varepsilon^2}(K) \right]$$

$$\propto \frac{\sum_{i=1}^{b} \frac{\mathbb{P}(i \in \hat{S})^2 \mathbb{E} \left[ L_{i,\hat{S}}^0 \mathbb{I}(i \in \hat{S}) \right]}{\left( \mathbb{E} \left[ L_{i,\hat{S}}^1 \mathbb{I}(i \in \hat{S}) \right] \right)^2}}{\min_{i \in [b]} \left( \frac{\mathbb{P}(i \in \hat{S})^2}{\mathbb{E} \left[ L_{i,\hat{S}}^1 \mathbb{I}(i \in \hat{S}) \right]} \right)^2} \left( c_{\mathrm{ov}} + \sum_{j=1}^{b} c_j \left( 1 - \frac{\binom{b-j}{\tau}}{\binom{b}{\tau}} \right) + \frac{\tau}{b} \sum_{j=1}^{b} c_j^{\sharp} \right)$$

$$= \frac{\max_{i \in [b]} \left( \sum_{\substack{S \subseteq [b] \\ |S| = \tau}} L_{i,S}^1 \mathbb{I} \left( i \in S \right) \right)^2}{\frac{\tau}{b} \binom{b-1}{\tau-1}}$$

$$\times \sum_{i=1}^{b} \frac{\sum_{\substack{S \subseteq [b] \\ |S| = \tau}} L_{i,S}^0 \mathbb{I} \left( i \in S \right)}{\left( \sum_{\substack{S \subseteq [b] \\ |S| = \tau}} L_{i,S}^1 \mathbb{I} \left( i \in S \right) \right)^2} \left( c_{\mathrm{ov}} + \sum_{j=1}^{b} c_j \left( 1 - \frac{\binom{b-j}{\tau}}{\binom{b}{\tau}} \right) + \frac{\tau}{b} \sum_{j=1}^{b} c_j^{\sharp} \right)$$

and

$$\mathbb{E} \left[ \mathrm{cost}_{\varepsilon}(K) \right] \propto \frac{1}{\min_{i \in [b]} \frac{\mathbb{P}(i \in \hat{S})^2}{\mathbb{E} \left[ L_{i,\hat{S}}^1 \mathbb{I}(i \in \hat{S}) \right]}} \left( c_{\mathrm{ov}} + \sum_{j=1}^{b} c_j \left( 1 - \frac{\binom{b-j}{\tau}}{\binom{b}{\tau}} \right) + \frac{\tau}{b} \sum_{j=1}^{b} c_j^{\sharp} \right)$$

$$= \frac{\max_{i\in[b]}\sum_{\substack{S\subseteq[b]\\|S|=\tau}}L^1_{i,S}\mathbb{I}\,(i\in S)}{\left(\frac{\tau}{b}\right)^2\binom{b}{\tau}}\left(c_{\text{ov}}+\sum_{j=1}^{b}c_j\left(1-\frac{\binom{b-j}{\tau}}{\binom{b}{\tau}}\right)+\frac{\tau}{b}\sum_{j=1}^{b}c^\sharp_j\right).$$

As in Section E.1.2, in general there is no closed-form expression for $\tau$ minimizing these costs, but it can be shown that $\tau = b$ need not be optimal. Let us make a simplifying assumption that $L^0_{i,S}$ depends only on $i$ and $|S|$ and define $L^0_{i,\tau} := L^0_{i,S}$ for $|S| = \tau$. Then

$$\mathbb{E}\left[\text{cost}_{\varepsilon^2}(K)\right]$$

$$\propto \frac{\max_{i\in[b]}\left(\sum_{\substack{S\subseteq[b]\\|S|=\tau}}L^1_{i,\tau}\mathbb{I}\,(i\in S)\right)^2}{\frac{\tau}{b}\binom{b-1}{\tau-1}}$$

$$\times \sum_{i=1}^{b}\frac{\sum_{\substack{S\subseteq[b]\\|S|=\tau}}L^0_{i,\tau}\mathbb{I}\,(i\in S)}{\left(\sum_{\substack{S\subseteq[b]\\|S|=\tau}}L^1_{i,\tau}\mathbb{I}\,(i\in S)\right)^2}\left(c_{\text{ov}}+\sum_{j=1}^{b}c_j\left(1-\frac{\binom{b-j}{\tau}}{\binom{b}{\tau}}\right)+\frac{\tau}{b}\sum_{j=1}^{b}c^\sharp_j\right)$$

$$= \frac{\max_{i\in[b]}\left(L^1_{i,\tau}\binom{b-1}{\tau-1}\right)^2}{\frac{\tau}{b}\binom{b-1}{\tau-1}}\sum_{i=1}^{b}\frac{L^0_{i,\tau}\binom{b-1}{\tau-1}}{\left(L^1_{i,\tau}\binom{b-1}{\tau-1}\right)^2}\left(c_{\text{ov}}+\sum_{j=1}^{b}c_j\left(1-\frac{\binom{b-j}{\tau}}{\binom{b}{\tau}}\right)+\frac{\tau}{b}\sum_{j=1}^{b}c^\sharp_j\right)$$

$$= \underbrace{\max_{i\in[b]}\left(L^1_{i,\tau}\right)^2\sum_{i=1}^{b}\frac{L^0_{i,\tau}}{\left(L^1_{i,\tau}\right)^2}}_{:=A_{\varepsilon^2}(\tau)}\underbrace{\left(\frac{b}{\tau}c_{\text{ov}}+\sum_{j=1}^{b}c_j\left(\frac{b}{\tau}-\frac{\binom{b-j}{\tau}}{\binom{b-1}{\tau-1}}\right)+\sum_{j=1}^{b}c^\sharp_j\right)}_{:=B(\tau)}$$

and

$$\mathbb{E}\left[\text{cost}_{\varepsilon}(K)\right] \propto \frac{\max_{i\in[b]}\sum_{\substack{S\subseteq[b]\\|S|=\tau}}L^1_{i,\tau}\mathbb{I}\,(i\in S)}{\left(\frac{\tau}{b}\right)^2\binom{b}{\tau}}\left(c_{\text{ov}}+\sum_{j=1}^{b}c_j\left(1-\frac{\binom{b-j}{\tau}}{\binom{b}{\tau}}\right)+\frac{\tau}{b}\sum_{j=1}^{b}c^\sharp_j\right)$$

$$= \frac{\max_{i\in[b]}L^1_{i,\tau}\binom{b-1}{\tau-1}}{\left(\frac{\tau}{b}\right)^2\binom{b}{\tau}}\left(c_{\text{ov}}+\sum_{j=1}^{b}c_j\left(1-\frac{\binom{b-j}{\tau}}{\binom{b}{\tau}}\right)+\frac{\tau}{b}\sum_{j=1}^{b}c^\sharp_j\right)$$

$$= \underbrace{\max_{i\in[b]}L^1_{i,\tau}}_{:=A_{\varepsilon}(\tau)}\underbrace{\left(\frac{b}{\tau}c_{\text{ov}}+\sum_{j=1}^{b}c_j\left(\frac{b}{\tau}-\frac{\binom{b-j}{\tau}}{\binom{b-1}{\tau-1}}\right)+\sum_{j=1}^{b}c^\sharp_j\right)}_{:=B(\tau)}.$$

We have already established in Section E.1.2 that $B$ is decreasing in $\tau$. In fact, the expression to be minimized there is structurally very similar to these obtained above. Specifically, in the smooth case we had $\text{cost}_{\varepsilon}(\mathcal{D}) \propto A(\tau)B(\tau)$, where $A(\tau) := \max_{i\in[b]}L^0_{i,\tau}$. Since $L^1_{i,\tau}$, like $L^0_{i,\tau}$, is non-decreasing in $\tau$, the same reasoning as in Section E.1.2 applies to $\mathbb{E}\left[\text{cost}_{\varepsilon}(K)\right]$. On the other hand, the dependence of $A_{\varepsilon^2}(\tau)$ on $\tau$ can be arbitrary depending on the scaling between $L^0_{i,\tau}$ and $L^1_{i,\tau}$.

Consequently, both objectives $\mathbb{E}\left[\text{cost}_{\varepsilon}(K)\right]$ and $\mathbb{E}\left[\text{cost}_{\varepsilon^2}(K)\right]$ present a trade-off between a decreasing factor $B(\tau)$ and a (typically) increasing factor $A_1(\tau)$. If $A_{\varepsilon}(\tau)$ or $A_{\varepsilon^2}(\tau)$ grows sufficiently fast in $\tau$, it can compensate for the decrease of $B(\tau)$ and make a smaller $\tau$ (or even $\tau = 1$) optimal.

### E.2.3 $\tau$-SUBMODEL SAMPLING

Sample a starting index $\hat{s} \in \{1,\ldots,b-\tau+1\}$ with probability $p_i = \mathbb{P}\,(\hat{s} = i)$ and set $\hat{S} = \{\hat{s},\ldots,\hat{s}+\tau-1\}$.

For any fixed $i \in [b]$, we have

$$\mathbb{P}\left(i \in \hat{S}\right) = \sum_{j=\max\{1,i-\tau+1\}}^{\min\{i,b-\tau+1\}} p_j$$

and

$$\mathbb{E}\left[L_{i,\hat{S}}^1 \mathbb{I}\left(i \in \hat{S}\right)\right] = \sum_{j=\max\{1,i-\tau+1\}}^{\min\{i,b-\tau+1\}} p_j L_{i,\{j,\dots,j+\tau-1\}}^1.$$

We have also already shown that

$$\mathbb{E}\left[\text{cost}(\hat{S})\right] = c_{\text{ov}} + \sum_{j=1}^b c_j \left(\sum_{i=1}^{\min\{j,b-\tau+1\}} p_i\right) + \sum_{j=1}^b c_j^\sharp \left(\sum_{i=\max\{1,j-\tau+1\}}^{\min\{j,b-\tau+1\}} p_i\right).$$

**Partitioned $\tau$-submodel sampling.** Mimicking Section E.1.3, let us consider a special case of the above sampling scheme where the submodels partition $[b]$. For simplicity, we assume that $b$ is divisible by $\tau$ and let $m = b/\tau$. The algorithm is then equivalent to $\tau$-submodel sampling with starting-index distribution satisfying

$$p_{s_k} > 0 \quad (k = 1, \dots, m), \qquad p_j = 0 \text{ otherwise,}$$

where $s_k = (k-1)\tau + 1$, $k \in [m]$. For any fixed $i \in [b]$, the derivations above simplify to

$$\mathbb{P}\left(i \in \hat{S}\right) = \sum_{j=\max\{1,i-\tau+1\}}^{\min\{i,b-\tau+1\}} p_j = p_{s_{\lceil i/\tau \rceil}}$$

and

$$\mathbb{E}\left[L_{i,\hat{S}}^\alpha \mathbb{I}\left(i \in \hat{S}\right)\right] = \sum_{j=\max\{1,i-\tau+1\}}^{\min\{i,b-\tau+1\}} p_j L_{i,\{j,\dots,j+\tau-1\}}^\alpha = p_{s_{\lceil i/\tau \rceil}} L_{i,B_{\lceil i/\tau \rceil}}^\alpha$$

for $\alpha \in \{0,1\}$. Plugging this choice into (35) and (36) gives

$$\mathbb{E}\left[\text{cost}_{\varepsilon^2}(K)\right] \propto \frac{\sum_{i=1}^b \frac{\mathbb{P}(i\in\hat{S})^2 \mathbb{E}\left[L_{i,\hat{S}}^0 \mathbb{I}(i\in\hat{S})\right]}{\left(\mathbb{E}\left[L_{i,\hat{S}}^1 \mathbb{I}(i\in\hat{S})\right]\right)^2}}{\min_{i\in[b]} \left(\frac{\mathbb{P}(i\in\hat{S})^2}{\mathbb{E}\left[L_{i,\hat{S}}^1 \mathbb{I}(i\in\hat{S})\right]}\right)^2}$$

$$\times \left(c_{\text{ov}} + \sum_{j=1}^b c_j \left(\sum_{i=1}^{\min\{j,b-\tau+1\}} p_i\right) + \sum_{j=1}^b c_j^\sharp \left(\sum_{i=\max\{1,j-\tau+1\}}^{\min\{j,b-\tau+1\}} p_i\right)\right)$$

$$= \frac{\sum_{i=1}^b \frac{p_{s_{\lceil i/\tau \rceil}} L_{i,B_{\lceil i/\tau \rceil}}^0}{\left(L_{i,B_{\lceil i/\tau \rceil}}^1\right)^2}}{\min_{i\in[b]} \left(\frac{p_{s_{\lceil i/\tau \rceil}}}{L_{i,B_{\lceil i/\tau \rceil}}^1}\right)^2} \left(c_{\text{ov}} + \sum_{j=1}^b c_j \left(\sum_{k=1}^{\lceil j/\tau \rceil} p_{s_k}\right) + \sum_{j=1}^b c_j^\sharp p_{s_{\lceil j/\tau \rceil}}\right)$$

and

$$\mathbb{E}\left[\text{cost}_\varepsilon(K)\right] \propto \frac{1}{\min_{i\in[b]} \frac{\mathbb{P}(i\in\hat{S})^2}{\mathbb{E}\left[L_{i,\hat{S}}^1 \mathbb{I}(i\in\hat{S})\right]}}$$

$$\times \left(c_{\text{ov}} + \sum_{j=1}^b c_j \left(\sum_{i=1}^{\min\{j,b-\tau+1\}} p_i\right) + \sum_{j=1}^b c_j^\sharp \left(\sum_{i=\max\{1,j-\tau+1\}}^{\min\{j,b-\tau+1\}} p_i\right)\right)$$

$$= \frac{1}{\min_{i\in[b]} \frac{p_{s_{\lceil i/\tau\rceil}}}{L^1_{i,B_{\lceil i/\tau\rceil}}}} \left( c_{\mathrm{ov}} + \sum_{j=1}^{b} c_j \left( \sum_{k=1}^{\lceil j/\tau\rceil} p_{s_k} \right) + \sum_{j=1}^{b} c^\sharp_j p_{s_{\lceil j/\tau\rceil}} \right).$$

The expression for $\mathbb{E}\left[\mathrm{cost}_\varepsilon(K)\right]$ is entirely analogous to (25) derived in the smooth case, with $L^0_{i,B_{\lceil i/\tau\rceil}}$ replaced by $L^1_{i,B_{\lceil i/\tau\rceil}}$. Consequently, for a fixed $\tau$, the optimal block probabilities are

$$p^\star_{s_k} = \frac{\max_{i\in B_k} L^1_{i,B_k}}{\sum_{l=1}^{m} \max_{i\in B_l} L^1_{i,B_l}},$$

Again, the optimal number of blocks $m$ depends on the growth rate of the constants $L^1_{i,\tau}$ with respect to $\tau$ and on their interaction with the costs $c_j$ and $c^\sharp_j$. In general, the best choice of $m$ is not necessarily $b$.

The dependence of $\mathbb{E}\left[\mathrm{cost}_{\varepsilon^2}(K)\right]$ on $\tau$ is significantly more complex and governed by the relative scaling between $L^0_{i,\tau}$ and $L^1_{i,\tau}$.

### E.2.4 Arbitrary submodel sampling

Let $\{B_1,\ldots,B_m\}$ be a partition of $[b]$. Set $\hat{S} = B_i$ with probability $p_i$ (where $\sum_{i=1}^{m} p_i = 1$). For $i\in[b]$, $k(i)$ denotes the unique block with $i\in B_{k(i)}$ and $\underline{b}_k := \min B_k$.

For any $i\in[b]$, we have

$$\mathbb{P}\left(i\in\hat{S}\right) = p_{k(i)}$$

and

$$\mathbb{E}\left[ L^\alpha_{i,\hat{S}} \mathbb{I}\left(i\in\hat{S}\right) \right] = p_{k(i)} L^\alpha_{i,B_{k(i)}}$$

for $\alpha\in\{0,1\}$. Since

$$\mathbb{E}\left[\mathrm{cost}(\hat{S})\right] = c_{\mathrm{ov}} + \sum_{j=1}^{b} c_j \left( \sum_{k:\underline{b}_k\leq j} p_k \right) + \sum_{j=1}^{b} c^\sharp_j p_{k(j)},$$

the total expected costs become

$$\mathbb{E}\left[\mathrm{cost}_{\varepsilon^2}(K)\right] \quad \propto \quad \frac{\sum_{i=1}^{b} \frac{\mathbb{P}(i\in\hat{S})^2 \mathbb{E}\left[L^0_{i,\hat{S}}\mathbb{I}(i\in\hat{S})\right]}{\left(\mathbb{E}\left[L^1_{i,\hat{S}}\mathbb{I}(i\in\hat{S})\right]\right)^2}}{\min_{i\in[b]}\left(\frac{\mathbb{P}(i\in\hat{S})^2}{\mathbb{E}\left[L^1_{i,\hat{S}}\mathbb{I}(i\in\hat{S})\right]}\right)^2} \left( c_{\mathrm{ov}} + \sum_{j=1}^{b} c_j \left( \sum_{k:\underline{b}_k\leq j} p_k \right) + \sum_{j=1}^{b} c^\sharp_j p_{k(j)} \right)$$

$$= \frac{\sum_{i=1}^{b} \frac{p_{k(i)} L^0_{i,B_{k(i)}}}{\left(L^1_{i,B_{k(i)}}\right)^2}}{\min_{i\in[b]}\left(\frac{p_{k(i)}}{L^1_{i,B_{k(i)}}}\right)^2} \left( c_{\mathrm{ov}} + \sum_{j=1}^{b} c_j \left( \sum_{k:\underline{b}_k\leq j} p_k \right) + \sum_{j=1}^{b} c^\sharp_j p_{k(j)} \right)$$

and

$$\mathbb{E}\left[\mathrm{cost}_\varepsilon(K)\right] \quad \propto \quad \frac{1}{\min_{i\in[b]} \frac{\mathbb{P}(i\in\hat{S})^2}{\mathbb{E}\left[L^1_{i,\hat{S}}\mathbb{I}(i\in\hat{S})\right]}} \left( c_{\mathrm{ov}} + \sum_{j=1}^{b} c_j \left( \sum_{k:\underline{b}_k\leq j} p_k \right) + \sum_{j=1}^{b} c^\sharp_j p_{k(j)} \right)$$

$$= \frac{1}{\min_{i\in[b]} \frac{p_{k(i)}}{L^1_{i,B_{k(i)}}}} \left( c_{\mathrm{ov}} + \sum_{j=1}^{b} c_j \left( \sum_{k:\underline{b}_k\leq j} p_k \right) + \sum_{j=1}^{b} c^\sharp_j p_{k(j)} \right).$$

Following the reasoning in Section E.1.4, we find that for a fixed partition, the choice of probabilities minimizing $\mathbb{E}\left[\mathrm{cost}_\varepsilon(K)\right]$ is

$$p^\star_k = \frac{1/\delta^{\min}_k}{\sum_{l=1}^{m} 1/\delta^{\min}_l} = \frac{\max_{i\in B_k} L^0_{i,B_k}}{\sum_{l=1}^{m} \max_{i\in B_l} L^0_{i,B_l}},$$

and that submodel training can improve upon full model training in certain regimes.

## F   CONVERGENCE RESULTS – STOCHASTIC GRADIENT CASE

The proof of Theorem F.1 relies on two preliminary lemmas, which we state and prove first.

**Lemma F.1** (Descent Lemma). Let Assumption 4.4 hold and consider the update rule $X_i^{k+1} = \mathrm{LMO}_{\mathcal{B}(X_i^k, t_i^k)}(M_i^k)$, $i = 1, \ldots, b$, where $X^{k+1} = [X_1^{k+1}, \ldots, X_b^{k+1}] \in \mathcal{X}$, $X^k = [X_1^k, \ldots, X_b^k] \in \mathcal{X}$, $M^k = [M_1^k, \ldots, M_b^k] \in \mathcal{X}$ and $t_i^k > 0$. Then

$$
\begin{aligned}
f(X^{k+1}) &\leq f(X^k) + \sum_{i \in S^k} 2t_i^k \left\| \nabla_i f(X^k) - M_i^k \right\|_{(i)\star} - \sum_{i \in S^k} t_i^k \left\| \nabla_i f(X^k) \right\|_{(i)\star} \\
&\quad + \sum_{i \in S^k} \frac{L_{i,S^k}^0 + L_{i,S^k}^1 \left\| \nabla_i f(X^k) \right\|_{(i)\star}}{2} (t_i^k)^2.
\end{aligned}
$$

*Proof.* By Lemma H.3

$$
\begin{aligned}
&f(X^{k+1}) \\
&\leq f(X^k) + \left\langle \nabla f(X^k), X^{k+1} - X^k \right\rangle + \sum_{i \in S^k} \frac{L_{i,S^k}^0 + L_{i,S^k}^1 \left\| \nabla_i f(X^k) \right\|_{(i)\star}}{2} \left\| X_i^k - X_i^{k+1} \right\|_{(i)}^2 \\
&\overset{(48)}{=} f(X^k) + \sum_{i \in S^k} \left( \left\langle \nabla_i f(X^k) - M_i^k, X_i^{k+1} - X_i^k \right\rangle_{(i)} + \left\langle M_i^k, X_i^{k+1} - X_i^k \right\rangle_{(i)} \right) \\
&\quad + \sum_{i \in S^k} \frac{L_{i,S^k}^0 + L_{i,S^k}^1 \left\| \nabla_i f(X^k) \right\|_{(i)\star}}{2} (t_i^k)^2 \\
&\overset{(49)}{=} f(X^k) + \sum_{i \in S^k} \left( \left\langle \nabla_i f(X^k) - M_i^k, X_i^{k+1} - X_i^k \right\rangle_{(i)} - t_i^k \left\| M_i^k \right\|_{(i)\star} \right) \\
&\quad + \sum_{i \in S^k} \frac{L_{i,S^k}^0 + L_{i,S^k}^1 \left\| \nabla_i f(X^k) \right\|_{(i)\star}}{2} (t_i^k)^2 \\
&\leq f(X^k) + \sum_{i \in S^k} \left( t_i^k \left\| \nabla_i f(X^k) - M_i^k \right\|_{(i)\star} - t_i^k \left\| M_i^k \right\|_{(i)\star} \right. \\
&\qquad \left. + \frac{L_{i,S^k}^0 + L_{i,S^k}^1 \left\| \nabla_i f(X^k) \right\|_{(i)\star}}{2} (t_i^k)^2 \right),
\end{aligned}
$$

where in the last line we used the Cauchy-Schwarz inequality. Therefore, using triangle inequality, we get

$$
\begin{aligned}
&f(X^{k+1}) \\
&\leq f(X^k) + \sum_{i \in S^k} \left( t_i^k \left\| \nabla_i f(X^k) - M_i^k \right\|_{(i)\star} + t_i^k \left\| \nabla_i f(X^k) - M_i^k \right\|_{(i)\star} - t_i^k \left\| \nabla_i f(X^k) \right\|_{(i)\star} \right) \\
&\quad + \sum_{i \in S^k} \frac{L_{i,S^k}^0 + L_{i,S^k}^1 \left\| \nabla_i f(X^k) \right\|_{(i)\star}}{2} (t_i^k)^2 \\
&= f(X^k) + \sum_{i \in S^k} \left( 2t_i^k \left\| \nabla_i f(X^k) - M_i^k \right\|_{(i)\star} - t_i^k \left\| \nabla_i f(X^k) \right\|_{(i)\star} \right) \\
&\quad + \sum_{i \in S^k} \frac{L_{i,S^k}^0 + L_{i,S^k}^1 \left\| \nabla_i f(X^k) \right\|_{(i)\star}}{2} (t_i^k)^2
\end{aligned}
$$

as required. $\qquad \square$

**Lemma F.2.** Let Assumptions 4.4 and 4.5 hold. Then, the iterates of Algorithm 1 run with $t_i^k \equiv t_i$ satisfy

$$
\mathbb{E}\left[\left\|M_i^{k+1} - \nabla_i f(X^{k+1})\right\|_2\right]
$$

$$
\leq \quad \left(1 - \mathbb{E}\left[\mathbb{I}\left(i \in \hat{S}\right)\right]\beta_i\right)^{k+1} \mathbb{E}\left[\left\|M_i^0 - \nabla_i f(X^0)\right\|_2\right] + \frac{t_i \mathbb{E}\left[\left(1 - \mathbb{I}\left(i \in \hat{S}\right)\beta_i\right)L_{i,\hat{S}}^0\right]}{\underline{\rho}_i \beta_i \mathbb{E}\left[\mathbb{I}\left(i \in \hat{S}\right)\right]}
$$

$$
+ \frac{t_i}{\underline{\rho}_i}\mathbb{E}\left[\left(1 - \mathbb{I}\left(i \in \hat{S}\right)\beta_i\right)L_{i,\hat{S}}^1\right]\sum_{l=0}^{k}\left(1 - \mathbb{E}\left[\mathbb{I}\left(i \in \hat{S}\right)\right]\beta_i\right)^{k-l}\mathbb{E}\left[\left\|\nabla_i f(X^l)\right\|_{(i)\star}\right]
$$

$$
+ \sigma_i\sqrt{\beta_i}.
$$

*Proof.* The proof is inspired by Cutkosky & Mehta (2020, Theorem 1). First, using the momentum update rule, we have

$$
\begin{aligned}
M_i^{k+1} &= \left(1 - \mathbb{I}\left(i \in S^k\right)\beta_i\right)M_i^k + \mathbb{I}\left(i \in S^k\right)\beta_i \nabla_i f(X^{k+1}; \xi^{k+1}) \\
&= \left(1 - \beta_i^k\right)\left(M_i^k - \nabla_i f(X^k)\right) + \left(1 - \beta_i^k\right)\left(\nabla_i f(X^k) - \nabla_i f(X^{k+1})\right) \\
&\quad + \beta_i^k\left(\nabla_i f(X^{k+1}; \xi^{k+1}) - \nabla_i f(X^{k+1})\right) + \nabla_i f(X^{k+1}).
\end{aligned}
$$

where $\beta_i^k := \mathbb{I}\left(i \in S^k\right)\beta_i$. To simplify the notation, define $U_{1,i}^k := M_i^k - \nabla_i f(X^k)$, $U_{2,i}^k := \nabla_i f(X^k) - \nabla_i f(X^{k+1})$ and $U_{3,i}^k := \nabla_i f(X^k; \xi^k) - \nabla_i f(X^k)$. Then the above can be written as

$$
U_{1,i}^{k+1} = \left(1 - \beta_i^k\right)U_{1,i}^k + \left(1 - \beta_i^k\right)U_{2,i}^k + \beta_i^k U_{3,i}^{k+1}. \tag{41}
$$

Unrolling the recursion in (41), we get

$$
\begin{aligned}
U_{1,i}^{k+1} &= (1 - \beta_i^k)U_{1,i}^k + (1 - \beta_i^k)U_{2,i}^k + \beta_i^k U_{3,i}^{k+1} \\
&= \left(\prod_{m=0}^{k}(1 - \beta_i^m)\right)U_{1,i}^0 + \sum_{l=0}^{k}\left(\prod_{m=l}^{k}(1 - \beta_i^m)\right)U_{2,i}^l + \sum_{l=0}^{k}\left(\beta_i^l \prod_{m=l+1}^{k}(1 - \beta_i^m)\right)U_{3,i}^{l+1},
\end{aligned}
$$

where by convention $\prod_{m=a}^{b}(\cdot) = 1$ if $a > b$. Hence, taking norms and using the triangle inequality,

$$
\begin{aligned}
\mathbb{E}\left[\left\|U_{1,i}^{k+1}\right\|_2\right] &\leq \mathbb{E}\left[\left\|\left(\prod_{m=0}^{k}(1 - \beta_i^m)\right)U_{1,i}^0\right\|_2\right] + \mathbb{E}\left[\left\|\sum_{l=0}^{k}\left(\prod_{m=l}^{k}(1 - \beta_i^m)\right)U_{2,i}^l\right\|_2\right] \\
&\quad + \mathbb{E}\left[\left\|\sum_{l=0}^{k}\left(\beta_i^l \prod_{m=l+1}^{k}(1 - \beta_i^m)\right)U_{3,i}^{l+1}\right\|_2\right].
\end{aligned} \tag{42}
$$

Let us consider each of the terms separately. First, using the independence of $\{S^k\}_{k\geq 0}$, we have

$$
\begin{aligned}
\mathbb{E}\left[\left\|\left(\prod_{m=0}^{k}(1 - \beta_i^m)\right)U_{1,i}^0\right\|_2\right] &\leq \mathbb{E}\left[\left(\prod_{m=0}^{k}(1 - \beta_i^m)\right)\left\|U_{1,i}^0\right\|_2\right] \\
&= \mathbb{E}\left[\mathbb{E}\left[\left.\left(\prod_{m=0}^{k}(1 - \beta_i^m)\right)\right| X^k, M_i^k\right]\left\|U_{1,i}^0\right\|_2\right] \\
&= \prod_{m=0}^{k}\mathbb{E}\left[1 - \beta_i^m\right]\mathbb{E}\left[\left\|U_{1,i}^0\right\|_2\right]. \tag{43}
\end{aligned}
$$

Next, by Assumption 4.4

$$
\mathbb{E}\left[\left\|\sum_{l=0}^{k}\left(\prod_{m=l}^{k}(1 - \beta_i^m)\right)U_{2,i}^l\right\|_2\right]
$$

$$\leq \quad \mathbb{E}\left[\sum_{l=0}^{k}\left(\prod_{m=l}^{k}(1-\beta_i^m)\right)\left\|U_{2,i}^l\right\|_2\right]$$

$$\leq \quad \frac{1}{\underline{\rho}_i}\sum_{l=0}^{k}\mathbb{E}\left[\left(\prod_{m=l}^{k}(1-\beta_i^m)\right)\left\|\nabla_i f(X^l)-\nabla_i f(X^{l+1})\right\|_{(i)\star}\right]$$

$$\stackrel{(4.3)}{\leq} \quad \frac{1}{\underline{\rho}_i}\sum_{l=0}^{k}\mathbb{E}\left[\left(\prod_{m=l}^{k}(1-\beta_i^m)\right)\left(L_{i,S^l}^0+L_{i,S^l}^1\left\|\nabla_i f(X^l)\right\|_{(i)\star}\right)\left\|X_i^l-X_i^{l+1}\right\|_{(i)}\right].$$

Since the product $\prod_{m=l+1}^{k}(1-\beta_i^m)$ depends only on samplings at iterations $> l$, these factors are independent of the $\sigma$-algebra generated by $(X^l, S^l)$. Therefore,

$$\mathbb{E}\left[\left\|\sum_{l=0}^{k}\left(\prod_{m=l}^{k}(1-\beta_i^m)\right)U_{2,i}^l\right\|_2\right]$$

$$\leq \quad \frac{1}{\underline{\rho}_i}\sum_{l=0}^{k}\left(\left(\prod_{m=l+1}^{k}\mathbb{E}\left[1-\beta_i^m\right]\right)\mathbb{E}\left[(1-\beta_i^l)\left(L_{i,S^l}^0+L_{i,S^l}^1\left\|\nabla_i f(X^l)\right\|_{(i)\star}\right)\right]t_i\right),$$

where

$$\mathbb{E}\left[(1-\beta_i^l)\left(L_{i,S^l}^0+L_{i,S^l}^1\left\|\nabla_i f(X^l)\right\|_{(i)\star}\right)\right]$$

$$= \quad \mathbb{E}\left[(1-\beta_i^l)L_{i,S^l}^0\right]+\mathbb{E}\left[\mathbb{E}\left[(1-\beta_i^l)L_{i,S^l}^1\left\|\nabla_i f(X^l)\right\|_{(i)\star}\Big|X^l\right]\right]$$

$$= \quad \mathbb{E}\left[(1-\beta_i^l)L_{i,S^l}^0\right]+\mathbb{E}\left[(1-\beta_i^l)L_{i,S^l}^1\right]\mathbb{E}\left[\left\|\nabla_i f(X^l)\right\|_{(i)\star}\right].$$

Thus

$$\mathbb{E}\left[\left\|\sum_{l=0}^{k}\left(\prod_{m=l}^{k}(1-\beta_i^m)\right)U_{2,i}^l\right\|_2\right] \tag{44}$$

$$\leq \quad \frac{t_i}{\underline{\rho}_i}\sum_{l=0}^{k}\left(\prod_{m=l+1}^{k}\mathbb{E}\left[1-\beta_i^m\right]\right)\left(\mathbb{E}\left[(1-\beta_i^l)L_{i,S^l}^0\right]+\mathbb{E}\left[(1-\beta_i^l)L_{i,S^l}^1\right]\mathbb{E}\left[\left\|\nabla_i f(X^l)\right\|_{(i)\star}\right]\right).$$

Lastly, by Jensen's inequality, the last term can be bounded via

$$\mathbb{E}\left[\left\|\sum_{l=0}^{k}\left(\beta_i^l\prod_{m=l+1}^{k}(1-\beta_i^m)\right)U_{3,i}^{l+1}\right\|_2\right]$$

$$\leq \quad \sqrt{\mathbb{E}\left[\left\|\sum_{l=0}^{k}\underbrace{\left(\beta_i^l\prod_{m=l+1}^{k}(1-\beta_i^m)\right)}_{:=a_l}U_{3,i}^{l+1}\right\|_2^2\right]} = \sqrt{\mathbb{E}\left[\sum_{l,r=0}^{k}a_l a_r\left\langle U_{3,i}^{l+1},U_{3,i}^{r+1}\right\rangle\right]}$$

$$\stackrel{(4.5)}{=} \quad \sqrt{\mathbb{E}\left[\sum_{l=0}^{k}a_l^2\left\|U_{3,i}^{l+1}\right\|_2^2\right]} = \sqrt{\sum_{l=0}^{k}\mathbb{E}\left[\mathbb{E}\left[a_l^2\left\|U_{3,i}^{l+1}\right\|_2^2\Big|\{S^r\}_{r=l}^k,X^{l+1}\right]\right]}$$

$$= \quad \sqrt{\sum_{l=0}^{k}\mathbb{E}\left[a_l^2\mathbb{E}\left[\left\|U_{3,i}^{l+1}\right\|_2^2\Big|\{S^r\}_{r=l}^k,X^{l+1}\right]\right]} \stackrel{(4.5)}{\leq} \sigma_i\sqrt{\sum_{l=0}^{k}\mathbb{E}\left[a_l^2\right]}$$

$$= \quad \sigma_i\sqrt{\sum_{l=0}^{k}\left(\mathbb{E}\left[(\beta_i^l)^2\right]\prod_{m=l+1}^{k}\mathbb{E}\left[(1-\beta_i^m)^2\right]\right)}$$

$$= \quad \sigma_i\sqrt{\sum_{l=0}^{k}\left(\mathbb{E}\left[\mathbb{I}\left(i\in S^l\right)\beta_i^2\right]\prod_{m=l+1}^{k}\mathbb{E}\left[\left(1-\mathbb{I}\left(i\in S^m\right)\beta_i\right)^2\right]\right)}$$

$$
= \sigma_i \sqrt{\sum_{l=0}^{k} \left( \beta_i^2 \mathbb{E}\left[ \mathbb{I}\left(i \in \hat{S}\right) \right] \mathbb{E}\left[ \left(1 - \mathbb{I}\left(i \in \hat{S}\right)\beta_i\right)^2 \right]^{k-l} \right)}
$$

$$
\leq \sigma_i \sqrt{ \frac{\beta_i^2 \mathbb{E}\left[ \mathbb{I}\left(i \in \hat{S}\right) \right]}{1 - \mathbb{E}\left[ \left(1 - \mathbb{I}\left(i \in \hat{S}\right)\beta_i\right)^2 \right]} }
$$

$$
= \sigma_i \beta_i \sqrt{ \frac{\mathbb{E}\left[ \mathbb{I}\left(i \in \hat{S}\right) \right]}{\mathbb{E}\left[ (2 - \beta_i) \mathbb{I}\left(i \in \hat{S}\right) \beta_i \right]} } = \frac{\sigma_i \beta_i}{\sqrt{(2 - \beta_i)\,\beta_i}} \leq \sigma_i \sqrt{\beta_i}. \tag{45}
$$

Applying (43), (44) and (45) in (42) and noting that $\mathbb{E}\left[\beta_i^k\right] = \mathbb{E}\left[\mathbb{I}\left(i \in \hat{S}\right)\right] \beta_i$ yields

$$
\mathbb{E}\left[ \left\| U_{1,i}^{k+1} \right\|_2 \right]
$$

$$
\leq \prod_{m=0}^{k} \mathbb{E}\left[1 - \beta_i^m\right] \mathbb{E}\left[ \left\| U_{1,i}^0 \right\|_2 \right] + \frac{t_i}{\underline{\rho}_i} \sum_{l=0}^{k} \left( \prod_{m=l+1}^{k} \mathbb{E}\left[1 - \beta_i^m\right] \right) \mathbb{E}\left[ (1 - \beta_i^l) L_{i,S^l}^0 \right]
$$

$$
+ \frac{t_i}{\underline{\rho}_i} \sum_{l=0}^{k} \left( \prod_{m=l+1}^{k} \mathbb{E}\left[1 - \beta_i^m\right] \right) \mathbb{E}\left[ (1 - \beta_i^l) L_{i,S^l}^1 \right] \mathbb{E}\left[ \left\| \nabla_i f(X^l) \right\|_{(i)\star} \right] + \sigma_i \sqrt{\beta_i}
$$

$$
= \left( 1 - \mathbb{E}\left[ \mathbb{I}\left(i \in \hat{S}\right) \right] \beta_i \right)^{k+1} \mathbb{E}\left[ \left\| U_{1,i}^0 \right\|_2 \right]
$$

$$
+ \frac{t_i}{\underline{\rho}_i} \mathbb{E}\left[ \left(1 - \mathbb{I}\left(i \in \hat{S}\right)\beta_i\right) L_{i,\hat{S}}^0 \right] \sum_{l=0}^{k} \left( 1 - \mathbb{E}\left[ \mathbb{I}\left(i \in \hat{S}\right) \right] \beta_i \right)^{k-l}
$$

$$
+ \frac{t_i}{\underline{\rho}_i} \mathbb{E}\left[ \left(1 - \mathbb{I}\left(i \in \hat{S}\right)\beta_i\right) L_{i,\hat{S}}^1 \right] \sum_{l=0}^{k} \left( 1 - \mathbb{E}\left[ \mathbb{I}\left(i \in \hat{S}\right) \right] \beta_i \right)^{k-l} \mathbb{E}\left[ \left\| \nabla_i f(X^l) \right\|_{(i)\star} \right]
$$

$$
+ \sigma_i \sqrt{\beta_i}
$$

$$
\leq \left( 1 - \mathbb{E}\left[ \mathbb{I}\left(i \in \hat{S}\right) \right] \beta_i \right)^{k+1} \mathbb{E}\left[ \left\| U_{1,i}^0 \right\|_2 \right] + \frac{t_i}{\underline{\rho}_i \beta_i \mathbb{E}\left[ \mathbb{I}\left(i \in \hat{S}\right) \right]} \mathbb{E}\left[ \left(1 - \mathbb{I}\left(i \in \hat{S}\right)\beta_i\right) L_{i,\hat{S}}^0 \right]
$$

$$
+ \frac{t_i}{\underline{\rho}_i} \mathbb{E}\left[ \left(1 - \mathbb{I}\left(i \in \hat{S}\right)\beta_i\right) L_{i,\hat{S}}^1 \right] \sum_{l=0}^{k} \left( 1 - \mathbb{E}\left[ \mathbb{I}\left(i \in \hat{S}\right) \right] \beta_i \right)^{k-l} \mathbb{E}\left[ \left\| \nabla_i f(X^l) \right\|_{(i)\star} \right]
$$

$$
+ \sigma_i \sqrt{\beta_i}.
$$

$\square$

**Theorem F.1.** Let Assumptions 4.1, 4.4, and 4.5 hold. Let $\{X^k\}_{k=0}^{K-1}$, $K \geq 1$, be the iterates of Algorithm 1 initialized with $M_i^0 = \nabla_i f(X^0; \xi^0)$ and run with $\beta_i \equiv \beta = 1/(K+1)^{1/2}$ and

$$
0 < t_i^k \equiv t_i = \frac{\eta_i}{(K+1)^{3/4}}, \qquad i = 1, \ldots, b,
$$

where $\eta_i^2 \leq \min\left\{ \dfrac{(K+1)^{1/2}}{4\mathbb{E}\left[\mathbb{I}(i \in \hat{S}) L_{i,\hat{S}}^1\right] \mathbb{E}\left[\max\limits_{i \in [b]} L_{i,\hat{S}}^1\right]}, \dfrac{\underline{\rho}_i \min\limits_{i \in [b]}\left(\mathbb{E}[\mathbb{I}(i \in \hat{S})]\right)}{16 \bar{\rho}_i \mathbb{E}[\mathbb{I}(i \in \hat{S})] \mathbb{E}\left[\left(1 - \mathbb{I}(i \in \hat{S})\beta_i\right) L_{i,\hat{S}}^1\right] \mathbb{E}\left[\max\limits_{i \in [b]} L_{i,\hat{S}}^1\right]}, 1 \right\}.$

Then

$$
\min_{k=0,\ldots,K} \sum_{i=1}^{b} \frac{\mathbb{E}\left[ \mathbb{I}\left(i \in \hat{S}\right) \right] \eta_i}{\frac{1}{b} \sum_{l=1}^{b} \mathbb{E}\left[ \mathbb{I}\left(l \in \hat{S}\right) \right] \eta_l} \mathbb{E}\left[ \left\| \nabla_i f(X^k) \right\|_{(i)\star} \right]
$$

$$\leq \frac{3\delta^0}{(K+1)^{1/4}\frac{1}{b}\sum_{l=1}^{b}\mathbb{E}\left[\mathbb{I}\left(l\in\hat{S}\right)\right]\eta_l} + \frac{6}{(K+1)^{1/2}}\sum_{i=1}^{b}\frac{\eta_i\bar{\rho}_i\sigma_i}{\frac{1}{b}\sum_{l=1}^{b}\mathbb{E}\left[\mathbb{I}\left(l\in\hat{S}\right)\right]\eta_l}$$

$$+\sum_{i=1}^{b}\frac{\eta_i^2}{(K+1)^{1/4}\frac{1}{b}\sum_{l=1}^{b}\mathbb{E}\left[\mathbb{I}\left(l\in\hat{S}\right)\right]\eta_l}\frac{2\bar{\rho}_i}{\underline{\rho}_i}\left(\mathbb{E}\left[L_{i,\hat{S}}^0\right]+\mathbb{E}\left[L_{i,\hat{S}}^1\right]\mathbb{E}\left[\frac{L_{i,\hat{S}}^0}{L_{i,\hat{S}}^1}\right]\right)$$

$$+\sum_{i=1}^{b}\frac{\eta_i^2}{2(K+1)^{3/4}\frac{1}{b}\sum_{l=1}^{b}\mathbb{E}\left[\mathbb{I}\left(l\in\hat{S}\right)\right]\eta_l}$$

$$\times\left(\mathbb{E}\left[\mathbb{I}\left(i\in\hat{S}\right)L_{i,\hat{S}}^0\right]+\mathbb{E}\left[\mathbb{I}\left(i\in\hat{S}\right)L_{i,\hat{S}}^1\right]\mathbb{E}\left[\frac{L_{i,\hat{S}}^0}{L_{i,\hat{S}}^1}\right]\right)$$

$$+\sum_{i=1}^{b}\frac{2\mathbb{E}\left[\mathbb{I}\left(i\in\hat{S}\right)\right]\eta_i}{(K+1)^{1/4}\frac{1}{b}\sum_{l=1}^{b}\mathbb{E}\left[\mathbb{I}\left(l\in\hat{S}\right)\right]\eta_l}\bar{\rho}_i\sigma_i.$$

**Remark F.2.** For Theorem F.1 to be meaningful, the sampling must be such that $\mathbb{E}\left[\mathbb{I}\left(i\in\hat{S}\right)\right]\eta_i > 0$ for every $i\in[b]$. Thus, every layer has to be sampled with positive probability.

**Remark F.3.** Note that under Assumption 4.4, without loss of generality we can set $L_{i,S}^0 = L_{i,S}^1 = 0$ whenever $i\notin S$. Hence, in the case of RPT (see Section 4), we have

$$\mathbb{E}\left[\mathbb{I}\left(i\in\hat{S}\right)\right] = \sum_{s=1}^{i}p_s,$$

$$\mathbb{E}\left[\max_{i\in[b]}L_{i,\hat{S}}^1\right] = \sum_{s=1}^{b}p_s\max_{i\in[b]}L_{i,\{s,\ldots,b\}}^1,$$

$$\mathbb{E}\left[\frac{L_{i,\hat{S}}^0}{L_{i,\hat{S}}^1}\right] = \sum_{s=1}^{b}p_s\frac{L_{i,\{s,\ldots,b\}}^0}{L_{i,\{s,\ldots,b\}}^1} = \sum_{s=1}^{i}p_s\frac{L_{i,\{s,\ldots,b\}}^0}{L_{i,\{s,\ldots,b\}}^1},$$

$$\mathbb{E}\left[\left(1-\mathbb{I}\left(i\in\hat{S}\right)\beta_i\right)L_{i,\hat{S}}^1\right] = \sum_{s=1}^{b}p_s(1-\mathbb{I}\left(i\in\{s,\ldots,b\}\right)\beta_i)L_{i,\{s,\ldots,b\}}^1$$

$$= (1-\beta_i)\sum_{s=1}^{i}p_sL_{i,\{s,\ldots,b\}}^1,$$

$$\mathbb{E}\left[L_{i,\hat{S}}^\alpha\right] = \sum_{s=1}^{b}p_sL_{i,\{s,\ldots,b\}}^\alpha = \sum_{s=1}^{i}p_sL_{i,\{s,\ldots,b\}}^\alpha,$$

$$\mathbb{E}\left[L_{i,\hat{S}}^\alpha\mathbb{I}\left(i\in\hat{S}\right)\right] = \sum_{s=1}^{i}p_sL_{i,\{s,\ldots,b\}}^\alpha$$

for $\alpha\in\{0,1\}$. Substituting these expressions into the rate yields the result in Theorem 4.4.

*Proof of Theorem F.1.* Taking expectation conditional on $[X^k,\{M_i^k\}_{i\in[b]}]$ in Lemma F.1 gives

$$\mathbb{E}\left[f(X^{k+1})\middle|X^k,\{M_i^k\}_{i\in[b]}\right]$$

$$\leq f(X^k)+\mathbb{E}\left[\sum_{i\in S^k}2t_i\left\|\nabla_if(X^k)-M_i^k\right\|_{(i)\star}-\sum_{i\in S^k}t_i\left\|\nabla_if(X^k)\right\|_{(i)\star}\middle|X^k,\{M_i^k\}_{i\in[b]}\right]$$

$$+\mathbb{E}\left[\sum_{i\in S^k}\frac{L_{i,S}^0+L_{i,S}^1\left\|\nabla_if(X^k)\right\|_{(i)\star}}{2}t_i^2\middle|X^k,\{M_i^k\}_{i\in[b]}\right]$$

$$= f(X^k) + \sum_{i=1}^{b} \mathbb{E}\left[\mathbb{I}\left(i \in S^k\right)\left(2t_i \left\|\nabla_i f(X^k) - M_i^k\right\|_{(i)\star} - t_i \left\|\nabla_i f(X^k)\right\|_{(i)\star}\right) \Big| X^k, \{M_i^k\}_{i \in [b]}\right]$$

$$+ \sum_{i=1}^{b} \mathbb{E}\left[\mathbb{I}\left(i \in S^k\right) \frac{L_{i,S}^0 + L_{i,S}^1 \left\|\nabla_i f(X^k)\right\|_{(i)\star}}{2} t_i^2 \Big| X^k, \{M_i^k\}_{i \in [b]}\right]$$

$$\leq f(X^k) + \sum_{i=1}^{b} \mathbb{E}\left[\mathbb{I}\left(i \in \hat{S}\right)\right]\left(2t_i \bar{\rho}_i \left\|\nabla_i f(X^k) - M_i^k\right\|_2 - t_i \left\|\nabla_i f(X^k)\right\|_{(i)\star}\right)$$

$$+ \frac{1}{2}\sum_{i=1}^{b} \mathbb{E}\left[\mathbb{I}\left(i \in \hat{S}\right) L_{i,\hat{S}}^0\right] t_i^2 + \frac{1}{2}\sum_{i=1}^{b} \mathbb{E}\left[\mathbb{I}\left(i \in \hat{S}\right) L_{i,\hat{S}}^1\right] t_i^2 \left\|\nabla_i f(X^k)\right\|_{(i)\star}.$$

Hence, taking full expectation

$$\mathbb{E}\left[f(X^{k+1})\right]$$

$$\leq \quad \mathbb{E}\left[f(X^k)\right] + \sum_{i=1}^{b} 2t_i \bar{\rho}_i \mathbb{E}\left[\mathbb{I}\left(i \in \hat{S}\right)\right] \mathbb{E}\left[\left\|\nabla_i f(X^k) - M_i^k\right\|_2\right]$$

$$- \sum_{i=1}^{b} \mathbb{E}\left[\mathbb{I}\left(i \in \hat{S}\right)\right] t_i \mathbb{E}\left[\left\|\nabla_i f(X^k)\right\|_{(i)\star}\right]$$

$$+ \frac{1}{2}\sum_{i=1}^{b} \mathbb{E}\left[\mathbb{I}\left(i \in \hat{S}\right) L_{i,\hat{S}}^0\right] t_i^2 + \frac{1}{2}\sum_{i=1}^{b} \mathbb{E}\left[\mathbb{I}\left(i \in \hat{S}\right) L_{i,\hat{S}}^1\right] t_i^2 \mathbb{E}\left[\left\|\nabla_i f(X^k)\right\|_{(i)\star}\right].$$

To simplify the notation, let $\delta^k := \mathbb{E}\left[f(X^k) - f^\star\right]$ and $P_i^k := \mathbb{E}\left[\left\|\nabla_i f(X^k) - M_i^k\right\|_2\right]$. Then, according to the descent inequality above and Lemma F.2

$$\delta^{k+1} \leq \quad \delta^k - \sum_{i=1}^{b} t_i \mathbb{E}\left[\mathbb{I}\left(i \in \hat{S}\right)\right] \mathbb{E}\left[\left\|\nabla_i f(X^k)\right\|_{(i)\star}\right] + \sum_{i=1}^{b} 2t_i \bar{\rho}_i \mathbb{E}\left[\mathbb{I}\left(i \in \hat{S}\right)\right] P_i^k$$

$$+ \frac{1}{2}\sum_{i=1}^{b} t_i^2 \mathbb{E}\left[\mathbb{I}\left(i \in \hat{S}\right) L_{i,\hat{S}}^0\right] + \frac{1}{2}\sum_{i=1}^{b} t_i^2 \mathbb{E}\left[\mathbb{I}\left(i \in \hat{S}\right) L_{i,\hat{S}}^1\right] \mathbb{E}\left[\left\|\nabla_i f(X^k)\right\|_{(i)\star}\right], \quad (46)$$

$$P_i^k \leq \quad \left(1 - \mathbb{E}\left[\mathbb{I}\left(i \in \hat{S}\right)\right]\beta_i\right)^k P_i^0 + \frac{t_i \mathbb{E}\left[\left(1 - \mathbb{I}\left(i \in \hat{S}\right)\beta_i\right) L_{i,\hat{S}}^0\right]}{\underline{\rho}_i \beta_i \mathbb{E}\left[\mathbb{I}\left(i \in \hat{S}\right)\right]}$$

$$+ \frac{t_i}{\underline{\rho}_i} \mathbb{E}\left[\left(1 - \mathbb{I}\left(i \in \hat{S}\right)\beta_i\right) L_{i,\hat{S}}^1\right] \sum_{l=0}^{k-1}\left(1 - \mathbb{E}\left[\mathbb{I}\left(i \in \hat{S}\right)\right]\beta_i\right)^{k-1-l} \mathbb{E}\left[\left\|\nabla_i f(X^l)\right\|_{(i)\star}\right]$$

$$+ \sigma_i \sqrt{\beta_i}. \quad (47)$$

Applying (47) in (46),

$$\delta^{k+1}$$

$$\leq \quad \delta^k - \sum_{i=1}^{b} t_i \mathbb{E}\left[\mathbb{I}\left(i \in \hat{S}\right)\right] \mathbb{E}\left[\left\|\nabla_i f(X^k)\right\|_{(i)\star}\right]$$

$$+ \sum_{i=1}^{b} 2t_i \bar{\rho}_i \mathbb{E}\left[\mathbb{I}\left(i \in \hat{S}\right)\right]\left(\left(1 - \mathbb{E}\left[\mathbb{I}\left(i \in \hat{S}\right)\right]\beta_i\right)^k P_i^0 + \frac{t_i \mathbb{E}\left[\left(1 - \mathbb{I}\left(i \in \hat{S}\right)\beta_i\right) L_{i,\hat{S}}^0\right]}{\underline{\rho}_i \beta_i \mathbb{E}\left[\mathbb{I}\left(i \in \hat{S}\right)\right]}\right)$$

$$+ \sum_{i=1}^{b}\left(\frac{2t_i^2 \bar{\rho}_i}{\underline{\rho}_i} \mathbb{E}\left[\mathbb{I}\left(i \in \hat{S}\right)\right] \mathbb{E}\left[\left(1 - \mathbb{I}\left(i \in \hat{S}\right)\beta_i\right) L_{i,\hat{S}}^1\right]\right.$$

$$\left. \times \sum_{l=0}^{k-1}\left(1 - \mathbb{E}\left[\mathbb{I}\left(i \in \hat{S}\right)\right]\beta_i\right)^{k-1-l} \mathbb{E}\left[\left\|\nabla_i f(X^l)\right\|_{(i)\star}\right]\right)$$

$$
+ \sum_{i=1}^{b} 2 t_i \bar{\rho}_i \mathbb{E}\left[\mathbb{I}\left(i \in \hat{S}\right)\right] \sigma_i \sqrt{\beta_i}
$$

$$
+ \frac{1}{2} \sum_{i=1}^{b} t_i^2 \mathbb{E}\left[\mathbb{I}\left(i \in \hat{S}\right) L_{i,\hat{S}}^0\right] + \frac{1}{2} \sum_{i=1}^{b} t_i^2 \mathbb{E}\left[\mathbb{I}\left(i \in \hat{S}\right) L_{i,\hat{S}}^1\right] \mathbb{E}\left[\left\|\nabla_i f(X^k)\right\|_{(i)\star}\right]
$$

$$
= \delta^k - \sum_{i=1}^{b} t_i \mathbb{E}\left[\mathbb{I}\left(i \in \hat{S}\right)\right] \mathbb{E}\left[\left\|\nabla_i f(X^k)\right\|_{(i)\star}\right]
$$

$$
+ \sum_{i=1}^{b} 2 t_i \bar{\rho}_i \mathbb{E}\left[\mathbb{I}\left(i \in \hat{S}\right)\right] \left(1 - \mathbb{E}\left[\mathbb{I}\left(i \in \hat{S}\right)\right] \beta_i\right)^k P_i^0
$$

$$
+ \sum_{i=1}^{b} \left( \frac{2 t_i^2 \bar{\rho}_i}{\underline{\rho}_i} \mathbb{E}\left[\mathbb{I}\left(i \in \hat{S}\right)\right] \mathbb{E}\left[\left(1 - \mathbb{I}\left(i \in \hat{S}\right) \beta_i\right) L_{i,\hat{S}}^1\right] \right.
$$

$$
\left. \times \sum_{l=0}^{k-1} \left(1 - \mathbb{E}\left[\mathbb{I}\left(i \in \hat{S}\right)\right] \beta_i\right)^{k-1-l} \mathbb{E}\left[\left\|\nabla_i f(X^l)\right\|_{(i)\star}\right] \right)
$$

$$
+ \frac{1}{2} \sum_{i=1}^{b} t_i^2 \mathbb{E}\left[\mathbb{I}\left(i \in \hat{S}\right) L_{i,\hat{S}}^1\right] \mathbb{E}\left[\left\|\nabla_i f(X^k)\right\|_{(i)\star}\right]
$$

$$
+ \frac{1}{2} \sum_{i=1}^{b} t_i^2 \mathbb{E}\left[\mathbb{I}\left(i \in \hat{S}\right) L_{i,\hat{S}}^0\right] + \sum_{i=1}^{b} \frac{2 t_i^2 \bar{\rho}_i \mathbb{E}\left[\left(1 - \mathbb{I}\left(i \in \hat{S}\right) \beta_i\right) L_{i,\hat{S}}^0\right]}{\beta_i \underline{\rho}_i}
$$

$$
+ \sum_{i=1}^{b} 2 t_i \bar{\rho}_i \mathbb{E}\left[\mathbb{I}\left(i \in \hat{S}\right)\right] \sigma_i \sqrt{\beta_i}.
$$

Let us now look at the terms involving the gradient norms. Using Lemma H.5, we get

$$
\sum_{i=1}^{b} \left( t_i^2 \underbrace{\frac{2 \bar{\rho}_i}{\underline{\rho}_i} \mathbb{E}\left[\mathbb{I}\left(i \in \hat{S}\right)\right] \mathbb{E}\left[\left(1 - \mathbb{I}\left(i \in \hat{S}\right) \beta_i\right) L_{i,\hat{S}}^1\right]}_{:=b_i} \right.
$$

$$
\left. \times \sum_{l=0}^{k-1} \left(1 - \mathbb{E}\left[\mathbb{I}\left(i \in \hat{S}\right)\right] \beta_i\right)^{k-1-l} \mathbb{E}\left[\left\|\nabla_i f(X^l)\right\|_{(i)\star}\right] \right)
$$

$$
= \sum_{l=0}^{k-1} \sum_{i=1}^{b} t_i^2 b_i \left(1 - \mathbb{E}\left[\mathbb{I}\left(i \in \hat{S}\right)\right] \beta_i\right)^{k-1-l} \mathbb{E}\left[\left\|\nabla_i f(X^l)\right\|_{(i)\star}\right]
$$

$$
\leq \sum_{l=0}^{k-1} \mathbb{E}\left[4 \max_{i \in [b]} \left(t_i^2 b_i \left(1 - \mathbb{E}\left[\mathbb{I}\left(i \in \hat{S}\right)\right] \beta_i\right)^{k-1-l} L_{i,S^l}^1\right) \left(f(X^l) - f^\star\right)\right]
$$

$$
+ \mathbb{E}\left[\sum_{i=1}^{b} \frac{t_i^2 b_i \left(1 - \mathbb{E}\left[\mathbb{I}\left(i \in \hat{S}\right)\right] \beta_i\right)^{k-1-l} L_{i,S^l}^0}{L_{i,S^l}^1}\right]
$$

$$
\leq 4 \sum_{l=0}^{k-1} \mathbb{E}\left[\mathbb{E}\left[\max_{i \in [b]} \left(t_i^2 b_i \left(1 - \mathbb{E}\left[\mathbb{I}\left(i \in \hat{S}\right)\right] \beta_i\right)^{k-1-l}\right) \max_{i \in [b]} \left(L_{i,S^l}^1\right) \left(f(X^l) - f^\star\right) \middle| X^l\right]\right]
$$

$$
+ \sum_{l=0}^{k-1} \sum_{i=1}^{b} t_i^2 b_i \left(1 - \mathbb{E}\left[\mathbb{I}\left(i \in \hat{S}\right)\right] \beta_i\right)^{k-1-l} \mathbb{E}\left[\frac{L_{i,\hat{S}}^0}{L_{i,\hat{S}}^1}\right]
$$

$$
= 4 \sum_{l=0}^{k-1} \max_{i \in [b]} \left(t_i^2 b_i \left(1 - \mathbb{E}\left[\mathbb{I}\left(i \in \hat{S}\right)\right] \beta_i\right)^{k-1-l}\right) \mathbb{E}\left[\max_{i \in [b]} L_{i,\hat{S}}^1\right] \delta^l
$$

$$+ \sum_{i=1}^{b} t_i^2 b_i \mathbb{E}\left[\frac{L_{i,\hat{S}}^0}{L_{i,\hat{S}}^1}\right] \sum_{l=0}^{k-1} \left(1 - \mathbb{E}\left[\mathbb{I}\left(i \in \hat{S}\right)\right] \beta_i\right)^{k-1-l}$$

$$\leq 4 \max_{i \in [b]} \left(t_i^2 b_i\right) \mathbb{E}\left[\max_{i \in [b]} L_{i,\hat{S}}^1 \sum_{l=0}^{k-1} \max_{i \in [b]} \left(\left(1 - \mathbb{E}\left[\mathbb{I}\left(i \in \hat{S}\right)\right] \beta_i\right)^{k-1-l}\right) \delta^l\right]$$

$$+ \sum_{i=1}^{b} \frac{t_i^2 b_i}{\mathbb{E}\left[\mathbb{I}\left(i \in \hat{S}\right)\right] \beta_i} \mathbb{E}\left[\frac{L_{i,\hat{S}}^0}{L_{i,\hat{S}}^1}\right]$$

$$= 4 \max_{i \in [b]} \left(t_i^2 b_i\right) \mathbb{E}\left[\max_{i \in [b]} L_{i,\hat{S}}^1 \sum_{l=0}^{k-1} \left(1 - \min_{i \in [b]} \left(\mathbb{E}\left[\mathbb{I}\left(i \in \hat{S}\right)\right] \beta_i\right)\right)^{k-1-l} \delta^l\right]$$

$$+ \sum_{i=1}^{b} \frac{t_i^2 b_i}{\mathbb{E}\left[\mathbb{I}\left(i \in \hat{S}\right)\right] \beta_i} \mathbb{E}\left[\frac{L_{i,\hat{S}}^0}{L_{i,\hat{S}}^1}\right],$$

and

$$\sum_{i=1}^{b} t_i^2 \mathbb{E}\left[\mathbb{I}\left(i \in \hat{S}\right) L_{i,\hat{S}}^1\right] \mathbb{E}\left[\left\|\nabla_i f(X^k)\right\|_{(i)\star}\right]$$

$$\leq \mathbb{E}\left[4 \max_{i \in [b]} \left(t_i^2 \mathbb{E}\left[\mathbb{I}\left(i \in \hat{S}\right) L_{i,\hat{S}}^1\right] L_{i,S^k}^1\right) \left(f(X^k) - f^\star\right) + \sum_{i=1}^{b} \frac{t_i^2 \mathbb{E}\left[\mathbb{I}\left(i \in \hat{S}\right) L_{i,\hat{S}}^1\right] L_{i,S^k}^0}{L_{i,S^k}^1}\right]$$

$$\leq 4 \max_{i \in [b]} \left(\mathbb{E}\left[\mathbb{I}\left(i \in \hat{S}\right) L_{i,\hat{S}}^1\right]\right) \mathbb{E}\left[\max_{i \in [b]} \left(t_i^2 L_{i,S^k}^1\right) \left(f(X^k) - f^\star\right)\right]$$

$$+ \sum_{i=1}^{b} t_i^2 \mathbb{E}\left[\mathbb{I}\left(i \in \hat{S}\right) L_{i,\hat{S}}^1\right] \mathbb{E}\left[\frac{L_{i,S^k}^0}{L_{i,S^k}^1}\right]$$

$$= 4 \max_{i \in [b]} \left(\mathbb{E}\left[\mathbb{I}\left(i \in \hat{S}\right) L_{i,\hat{S}}^1\right]\right) \mathbb{E}\left[\mathbb{E}\left[\max_{i \in [b]} \left(t_i^2 L_{i,S^k}^1\right) \left(f(X^k) - f^\star\right) \Big| X^k\right]\right]$$

$$+ \sum_{i=1}^{b} t_i^2 \mathbb{E}\left[\mathbb{I}\left(i \in \hat{S}\right) L_{i,\hat{S}}^1\right] \mathbb{E}\left[\frac{L_{i,\hat{S}}^0}{L_{i,\hat{S}}^1}\right]$$

$$= 4 \max_{i \in [b]} \left(\mathbb{E}\left[\mathbb{I}\left(i \in \hat{S}\right) L_{i,\hat{S}}^1\right]\right) \mathbb{E}\left[\max_{i \in [b]} \left(t_i^2 L_{i,\hat{S}}^1\right)\right] \delta^k$$

$$+ \sum_{i=1}^{b} t_i^2 \mathbb{E}\left[\mathbb{I}\left(i \in \hat{S}\right) L_{i,\hat{S}}^1\right] \mathbb{E}\left[\frac{L_{i,\hat{S}}^0}{L_{i,\hat{S}}^1}\right].$$

Thus

$$\delta^{k+1}$$

$$\leq \delta^k - \sum_{i=1}^{b} t_i \mathbb{E}\left[\mathbb{I}\left(i \in \hat{S}\right)\right] \mathbb{E}\left[\left\|\nabla_i f(X^k)\right\|_{(i)\star}\right]$$

$$+ \sum_{i=1}^{b} 2 t_i \bar{\rho}_i \mathbb{E}\left[\mathbb{I}\left(i \in \hat{S}\right)\right] \left(1 - \mathbb{E}\left[\mathbb{I}\left(i \in \hat{S}\right)\right] \beta_i\right)^k P_i^0$$

$$+ 4 \max_{i \in [b]} \left(t_i^2 b_i\right) \mathbb{E}\left[\max_{i \in [b]} L_{i,\hat{S}}^1 \sum_{l=0}^{k-1} \left(1 - \min_{i \in [b]} \left(\mathbb{E}\left[\mathbb{I}\left(i \in \hat{S}\right)\right] \beta_i\right)\right)^{k-1-l} \delta^l\right]$$

$$+ \sum_{i=1}^{b} \frac{t_i^2 b_i}{\mathbb{E}\left[\mathbb{I}\left(i \in \hat{S}\right)\right] \beta_i} \mathbb{E}\left[\frac{L_{i,\hat{S}}^0}{L_{i,\hat{S}}^1}\right] + 2 \max_{i \in [b]} \left(\mathbb{E}\left[\mathbb{I}\left(i \in \hat{S}\right) L_{i,\hat{S}}^1\right]\right) \mathbb{E}\left[\max_{i \in [b]} \left(t_i^2 L_{i,\hat{S}}^1\right)\right] \delta^k$$

$$+\frac{1}{2}\sum_{i=1}^{b}t_i^2\mathbb{E}\left[\mathbb{I}\left(i\in\hat{S}\right)L_{i,\hat{S}}^1\right]\mathbb{E}\left[\frac{L_{i,\hat{S}}^0}{L_{i,\hat{S}}^1}\right]$$

$$+\frac{1}{2}\sum_{i=1}^{b}t_i^2\mathbb{E}\left[\mathbb{I}\left(i\in\hat{S}\right)L_{i,\hat{S}}^0\right]+\sum_{i=1}^{b}\frac{2t_i^2\bar{\rho}_i\mathbb{E}\left[\left(1-\mathbb{I}\left(i\in\hat{S}\right)\beta_i\right)L_{i,\hat{S}}^0\right]}{\beta_i\underline{\rho}_i}$$

$$+\sum_{i=1}^{b}2t_i\bar{\rho}_i\mathbb{E}\left[\mathbb{I}\left(i\in\hat{S}\right)\right]\sigma_i\sqrt{\beta_i}$$

$$\leq\quad\left(1+2\max_{i\in[b]}\left(\mathbb{E}\left[\mathbb{I}\left(i\in\hat{S}\right)L_{i,\hat{S}}^1\right]\right)\mathbb{E}\left[\max_{i\in[b]}\left(t_i^2L_{i,\hat{S}}^1\right)\right]\right)\delta^k$$

$$-\sum_{i=1}^{b}t_i\mathbb{E}\left[\mathbb{I}\left(i\in\hat{S}\right)\right]\mathbb{E}\left[\left\|\nabla_if(X^k)\right\|_{(i)\star}\right]$$

$$+4\max_{i\in[b]}\left(t_i^2b_i\right)\mathbb{E}\left[\max_{i\in[b]}L_{i,\hat{S}}^1\right]\sum_{l=0}^{k-1}\left(1-\min_{i\in[b]}\left(\mathbb{E}\left[\mathbb{I}\left(i\in\hat{S}\right)\right]\beta_i\right)\right)^{k-1-l}\delta^l$$

$$+\sum_{i=1}^{b}2t_i\bar{\rho}_i\mathbb{E}\left[\mathbb{I}\left(i\in\hat{S}\right)\right]\left(1-\mathbb{E}\left[\mathbb{I}\left(i\in\hat{S}\right)\right]\beta_i\right)^kP_i^0$$

$$+\sum_{i=1}^{b}t_i^2\left(\frac{2\bar{\rho}_i\mathbb{E}\left[\left(1-\mathbb{I}\left(i\in\hat{S}\right)\beta_i\right)L_{i,\hat{S}}^1\right]}{\beta_i\underline{\rho}_i}+\frac{\mathbb{E}\left[\mathbb{I}\left(i\in\hat{S}\right)L_{i,\hat{S}}^1\right]}{2}\right)\mathbb{E}\left[\frac{L_{i,\hat{S}}^0}{L_{i,\hat{S}}^1}\right]$$

$$+\sum_{i=1}^{b}t_i^2\left(\frac{2\bar{\rho}_i\mathbb{E}\left[\left(1-\mathbb{I}\left(i\in\hat{S}\right)\beta_i\right)L_{i,\hat{S}}^0\right]}{\beta_i\underline{\rho}_i}+\frac{\mathbb{E}\left[\mathbb{I}\left(i\in\hat{S}\right)L_{i,\hat{S}}^0\right]}{2}\right)$$

$$+2\sum_{i=1}^{b}t_i\mathbb{E}\left[\mathbb{I}\left(i\in\hat{S}\right)\right]\bar{\rho}_i\sigma_i\sqrt{\beta_i}$$

$$=\quad\left(1+c_1\right)\delta^k-\sum_{i=1}^{b}t_i\mathbb{E}\left[\mathbb{I}\left(i\in\hat{S}\right)\right]\mathbb{E}\left[\left\|\nabla_if(X^k)\right\|_{(i)\star}\right]$$

$$+c_2\sum_{l=0}^{k-1}\left(1-\min_{i\in[b]}\left(\mathbb{E}\left[\mathbb{I}\left(i\in\hat{S}\right)\right]\beta_i\right)\right)^{k-1-l}\delta^l$$

$$+\sum_{i=1}^{b}2t_i\bar{\rho}_i\mathbb{E}\left[\mathbb{I}\left(i\in\hat{S}\right)\right]\left(1-\mathbb{E}\left[\mathbb{I}\left(i\in\hat{S}\right)\right]\beta_i\right)^kP_i^0+\sum_{i=1}^{b}t_i^2c_{3,i}+\sum_{i=1}^{b}t_ic_{4,i},$$

where

$$c_1\quad:=\quad 2\max_{i\in[b]}\left(\mathbb{E}\left[\mathbb{I}\left(i\in\hat{S}\right)L_{i,\hat{S}}^1\right]\right)\mathbb{E}\left[\max_{i\in[b]}\left(t_i^2L_{i,\hat{S}}^1\right)\right],$$

$$c_2\quad:=\quad 4\max_{i\in[b]}\left(t_i^2b_i\right)\mathbb{E}\left[\max_{i\in[b]}L_{i,\hat{S}}^1\right],$$

$$c_{3,i}\quad:=\quad\left(\frac{2\bar{\rho}_i\mathbb{E}\left[\left(1-\mathbb{I}\left(i\in\hat{S}\right)\beta_i\right)L_{i,\hat{S}}^1\right]}{\beta_i\underline{\rho}_i}+\frac{\mathbb{E}\left[\mathbb{I}\left(i\in\hat{S}\right)L_{i,\hat{S}}^1\right]}{2}\right)\mathbb{E}\left[\frac{L_{i,\hat{S}}^0}{L_{i,\hat{S}}^1}\right]$$

$$+\left(\frac{2\bar{\rho}_i\mathbb{E}\left[\left(1-\mathbb{I}\left(i\in\hat{S}\right)\beta_i\right)L_{i,\hat{S}}^0\right]}{\beta_i\underline{\rho}_i}+\frac{\mathbb{E}\left[\mathbb{I}\left(i\in\hat{S}\right)L_{i,\hat{S}}^0\right]}{2}\right),$$

$$c_{4,i}\quad:=\quad 2\mathbb{E}\left[\mathbb{I}\left(i\in\hat{S}\right)\right]\bar{\rho}_i\sigma_i\sqrt{\beta_i}.$$

Now, let us introduce a weighting sequence $w^k := w^{k-1}\left(1 + c_1 + \frac{c_2}{\min_{i \in [b]}\left(\mathbb{E}[\mathbb{I}(i \in \hat{S})]\beta_i\right)}\right)^{-1}$, where $w^{-1} = 1$ and $W^K := \sum_{k=0}^{K} w^k$. Then, multiplying the inequality above by $w^k$ and summing over the first $K + 1$ iterations gives

$$
\begin{aligned}
\sum_{k=0}^{K} w^k \delta^{k+1} &\leq \sum_{k=0}^{K} w^k (1 + c_1) \delta^k - \sum_{k=0}^{K} w^k \sum_{i=1}^{b} t_i \mathbb{E}\left[\mathbb{I}\left(i \in \hat{S}\right)\right] \mathbb{E}\left[\left\|\nabla_i f(X^k)\right\|_{(i)\star}\right] \\
&\quad + \sum_{k=0}^{K} w^k c_2 \sum_{l=0}^{k-1} \left(1 - \min_{i \in [b]}\left(\mathbb{E}\left[\mathbb{I}\left(i \in \hat{S}\right)\right]\beta_i\right)\right)^{k-1-l} \delta^l \\
&\quad + \sum_{k=0}^{K} w^k \sum_{i=1}^{b} 2 t_i \bar{\rho}_i \mathbb{E}\left[\mathbb{I}\left(i \in \hat{S}\right)\right] \left(1 - \mathbb{E}\left[\mathbb{I}\left(i \in \hat{S}\right)\right]\beta_i\right)^k P_i^0 \\
&\quad + \sum_{k=0}^{K} w^k \sum_{i=1}^{b} t_i^2 c_{3,i} + \sum_{k=0}^{K} w^k \sum_{i=1}^{b} t_i c_{4,i} \\
&\leq (1 + c_1) \sum_{k=0}^{K} w^k \delta^k - \sum_{k=0}^{K} w^k \sum_{i=1}^{b} t_i \mathbb{E}\left[\mathbb{I}\left(i \in \hat{S}\right)\right] \mathbb{E}\left[\left\|\nabla_i f(X^k)\right\|_{(i)\star}\right] \\
&\quad + c_2 \sum_{l=0}^{K-1} \sum_{k=l+1}^{K} w^k \left(1 - \min_{i \in [b]}\left(\mathbb{E}\left[\mathbb{I}\left(i \in \hat{S}\right)\right]\beta_i\right)\right)^{k-1-l} \delta^l \\
&\quad + \sum_{i=1}^{b} 2 t_i \bar{\rho}_i \mathbb{E}\left[\mathbb{I}\left(i \in \hat{S}\right)\right] P_i^0 \sum_{k=0}^{K} \left(1 - \mathbb{E}\left[\mathbb{I}\left(i \in \hat{S}\right)\right]\beta_i\right)^k \\
&\quad + W^K \sum_{i=1}^{b} t_i^2 c_{3,i} + W^K \sum_{i=1}^{b} t_i c_{4,i},
\end{aligned}
$$

where in the last line we used the fact that $w^k \leq w^{k-1} \leq w^{-1} = 1$. Therefore,

$$
\begin{aligned}
\sum_{k=0}^{K} w^k \delta^{k+1} &\leq (1 + c_1) \sum_{k=0}^{K} w^k \delta^k - \sum_{k=0}^{K} w^k \sum_{i=1}^{b} t_i \mathbb{E}\left[\mathbb{I}\left(i \in \hat{S}\right)\right] \mathbb{E}\left[\left\|\nabla_i f(X^k)\right\|_{(i)\star}\right] \\
&\quad + c_2 \sum_{l=0}^{K-1} \sum_{k=l+1}^{K} w^l \left(1 - \min_{i \in [b]}\left(\mathbb{E}\left[\mathbb{I}\left(i \in \hat{S}\right)\right]\beta_i\right)\right)^{k-1-l} \delta^l \\
&\quad + \sum_{i=1}^{b} \frac{2 t_i \bar{\rho}_i \mathbb{E}\left[\mathbb{I}\left(i \in \hat{S}\right)\right]}{\mathbb{E}\left[\mathbb{I}\left(i \in \hat{S}\right)\right]\beta_i} P_i^0 + W^K \sum_{i=1}^{b} t_i^2 c_{3,i} + W^K \sum_{i=1}^{b} t_i c_{4,i} \\
&\leq (1 + c_1) \sum_{k=0}^{K} w^k \delta^k - \sum_{k=0}^{K} w^k \sum_{i=1}^{b} t_i \mathbb{E}\left[\mathbb{I}\left(i \in \hat{S}\right)\right] \mathbb{E}\left[\left\|\nabla_i f(X^k)\right\|_{(i)\star}\right] \\
&\quad + c_2 \sum_{l=0}^{K-1} w^l \delta^l \sum_{k=l+1}^{K} \left(1 - \min_{i \in [b]}\left(\mathbb{E}\left[\mathbb{I}\left(i \in \hat{S}\right)\right]\beta_i\right)\right)^{k-1-l} \\
&\quad + \sum_{i=1}^{b} \frac{2 t_i \bar{\rho}_i}{\beta_i} P_i^0 + W^K \sum_{i=1}^{b} t_i^2 c_{3,i} + W^K \sum_{i=1}^{b} t_i c_{4,i} \\
&\leq (1 + c_1) \sum_{k=0}^{K} w^k \delta^k - \sum_{k=0}^{K} w^k \sum_{i=1}^{b} t_i \mathbb{E}\left[\mathbb{I}\left(i \in \hat{S}\right)\right] \mathbb{E}\left[\left\|\nabla_i f(X^k)\right\|_{(i)\star}\right] \\
&\quad + c_2 \sum_{l=0}^{K-1} w^l \delta^l \sum_{k=0}^{\infty} \left(1 - \min_{i \in [b]}\left(\mathbb{E}\left[\mathbb{I}\left(i \in \hat{S}\right)\right]\beta_i\right)\right)^k
\end{aligned}
$$

$$+ \sum_{i=1}^{b} \frac{2t_i \bar{\rho}_i}{\beta_i} P_i^0 + W^K \sum_{i=1}^{b} t_i^2 c_{3,i} + W^K \sum_{i=1}^{b} t_i c_{4,i}$$

$$\leq \left( 1 + c_1 + \frac{c_2}{\min_{i \in [b]} \left( \mathbb{E}\left[ \mathbb{I}\left( i \in \hat{S} \right) \right] \beta_i \right)} \right) \sum_{k=0}^{K} w^k \delta^k$$

$$- \sum_{k=0}^{K} w^k \sum_{i=1}^{b} t_i \mathbb{E}\left[ \mathbb{I}\left( i \in \hat{S} \right) \right] \mathbb{E}\left[ \left\| \nabla_i f(X^k) \right\|_{(i)\star} \right]$$

$$+ \sum_{i=1}^{b} \frac{2t_i \bar{\rho}_i}{\beta_i} P_i^0 + W^K \sum_{i=1}^{b} t_i^2 c_{3,i} + W^K \sum_{i=1}^{b} t_i c_{4,i}$$

$$= \sum_{k=0}^{K} w^{k-1} \delta^k - \sum_{k=0}^{K} w^k \sum_{i=1}^{b} t_i \mathbb{E}\left[ \mathbb{I}\left( i \in \hat{S} \right) \right] \mathbb{E}\left[ \left\| \nabla_i f(X^k) \right\|_{(i)\star} \right]$$

$$+ \sum_{i=1}^{b} \frac{2t_i \bar{\rho}_i}{\beta_i} P_i^0 + W^K \sum_{i=1}^{b} t_i^2 c_{3,i} + W^K \sum_{i=1}^{b} t_i c_{4,i}.$$

Rearranging the terms and dividing by $W^K$, we get

$$\min_{k=0,\dots,K} \sum_{i=1}^{b} t_i \mathbb{E}\left[ \mathbb{I}\left( i \in \hat{S} \right) \right] \mathbb{E}\left[ \left\| \nabla_i f(X^k) \right\|_{(i)\star} \right]$$

$$\leq \sum_{k=0}^{K} \sum_{i=1}^{b} \frac{w^k}{W^K} t_i \mathbb{E}\left[ \mathbb{I}\left( i \in \hat{S} \right) \right] \mathbb{E}\left[ \left\| \nabla_i f(X^k) \right\|_{(i)\star} \right]$$

$$\leq \frac{1}{W^K} \sum_{k=0}^{K} \left( w^{k-1} \delta^k - w^k \delta^{k+1} \right) + \frac{1}{W^K} \sum_{i=1}^{b} \frac{2t_i \bar{\rho}_i}{\beta_i} P_i^0 + \sum_{i=1}^{b} t_i^2 c_{3,i} + \sum_{i=1}^{b} t_i c_{4,i}$$

$$\leq \frac{\delta^0}{W^K} + \frac{1}{W^K} \sum_{i=1}^{b} \frac{2t_i \bar{\rho}_i}{\beta_i} P_i^0 + \sum_{i=1}^{b} t_i^2 c_{3,i} + \sum_{i=1}^{b} t_i c_{4,i}.$$

Now, note that taking $t_i = \frac{\eta_i}{(K+1)^{3/4}}$, where

$$\eta_i^2 \leq \min \left\{ \frac{(K+1)^{1/2}}{4\mathbb{E}\left[ \mathbb{I}\left( i \in \hat{S} \right) L_{i,\hat{S}}^1 \right] \mathbb{E}\left[ \max_{i \in [b]} L_{i,\hat{S}}^1 \right]}, \frac{(K+1)^{1/2} \underline{\rho}_i \min_{i \in [b]}\left( \mathbb{E}\left[ \mathbb{I}(i \in \hat{S}) \right] \beta_i \right)}{16\bar{\rho}_i \mathbb{E}\left[ \mathbb{I}(i \in \hat{S}) \right] \mathbb{E}\left[ \left( 1 - \mathbb{I}(i \in \hat{S}) \beta_i \right) L_{i,\hat{S}}^1 \right] \mathbb{E}\left[ \max_{i \in [b]} L_{i,\hat{S}}^1 \right]}, 1 \right\},$$

we have

$$2(K+1) \max_{i \in [b]} \left( \mathbb{E}\left[ \mathbb{I}\left( i \in \hat{S} \right) L_{i,\hat{S}}^1 \right] \right) \mathbb{E}\left[ \max_{i \in [b]} \left( t_i^2 L_{i,\hat{S}}^1 \right) \right] \leq \frac{1}{2},$$

$$(K+1) \frac{4 \max_{i \in [b]} \left( t_i^2 b_i \right) \mathbb{E}\left[ \max_{i \in [b]} L_{i,\hat{S}}^1 \right]}{\min_{i \in [b]} \left( \mathbb{E}\left[ \mathbb{I}\left( i \in \hat{S} \right) \right] \beta_i \right)} \leq \frac{1}{2},$$

and hence $(K+1)\left( c_1 + \frac{c_2}{\min_{i \in [b]}\left( \mathbb{E}\left[ \mathbb{I}(i \in \hat{S}) \right] \beta_i \right)} \right) \leq 1$. Thus, the weights satisfy

$$W^K = \sum_{k=0}^{K} w^k \geq (K+1) w^K = \frac{(K+1) w^{-1}}{\left( 1 + c_1 + \frac{c_2}{\min_{i \in [b]}\left( \mathbb{E}\left[ \mathbb{I}(i \in \hat{S}) \right] \beta_i \right)} \right)^{K+1}}$$

$$\geq \frac{K+1}{\exp\left( (K+1)\left( c_1 + \frac{c_2}{\min_{i \in [b]}\left( \mathbb{E}\left[ \mathbb{I}(i \in \hat{S}) \right] \beta_i \right)} \right) \right)} \geq \frac{K+1}{\exp(1)} \geq \frac{K+1}{3},$$

meaning that

$$\min_{k=0,\ldots,K} \sum_{i=1}^{b} t_i \mathbb{E}\left[\mathbb{I}\left(i \in \hat{S}\right)\right] \mathbb{E}\left[\left\|\nabla_i f(X^k)\right\|_{(i)\star}\right]$$

$$\leq \frac{3\delta^0}{K+1} + \frac{6}{K+1}\sum_{i=1}^{b} \frac{t_i \bar{\rho}_i}{\beta_i} P_i^0 + \sum_{i=1}^{b} t_i^2 c_{3,i} + \sum_{i=1}^{b} t_i c_{4,i}$$

$$= \frac{3\delta^0}{K+1} + \frac{6}{K+1}\sum_{i=1}^{b} \frac{\eta_i \bar{\rho}_i}{\beta_i(K+1)^{3/4}} P_i^0$$

$$+ \sum_{i=1}^{b} \frac{\eta_i^2}{(K+1)^{3/2}}\frac{2\bar{\rho}_i}{\beta_i \underline{\rho}_i}\left(\mathbb{E}\left[\left(1 - \mathbb{I}\left(i \in \hat{S}\right)\beta_i\right)L_{i,\hat{S}}^0\right] + \mathbb{E}\left[\left(1 - \mathbb{I}\left(i \in \hat{S}\right)\beta_i\right)L_{i,\hat{S}}^1\right]\mathbb{E}\left[\frac{L_{i,\hat{S}}^0}{L_{i,\hat{S}}^1}\right]\right)$$

$$+ \sum_{i=1}^{b} \frac{\eta_i^2}{2(K+1)^{3/2}}\left(\mathbb{E}\left[\mathbb{I}\left(i \in \hat{S}\right)L_{i,\hat{S}}^0\right] + \mathbb{E}\left[\mathbb{I}\left(i \in \hat{S}\right)L_{i,\hat{S}}^1\right]\mathbb{E}\left[\frac{L_{i,\hat{S}}^0}{L_{i,\hat{S}}^1}\right]\right)$$

$$+ \sum_{i=1}^{b} \frac{2\eta_i}{(K+1)^{3/4}}\mathbb{E}\left[\mathbb{I}\left(i \in \hat{S}\right)\right]\bar{\rho}_i \sigma_i \sqrt{\beta_i}.$$

Dividing by $\frac{1}{b}\sum_{l=1}^{b}\mathbb{E}\left[\mathbb{I}\left(l \in \hat{S}\right)\right]t_l = \frac{1}{(K+1)^{3/4}}\frac{1}{b}\sum_{l=1}^{b}\mathbb{E}\left[\mathbb{I}\left(l \in \hat{S}\right)\right]\eta_l$ gives

$$\min_{k=0,\ldots,K} \sum_{i=1}^{b} \frac{\mathbb{E}\left[\mathbb{I}\left(i \in \hat{S}\right)\right]\eta_i}{\frac{1}{b}\sum_{l=1}^{b}\mathbb{E}\left[\mathbb{I}\left(l \in \hat{S}\right)\right]\eta_l}\mathbb{E}\left[\left\|\nabla_i f(X^k)\right\|_{(i)\star}\right]$$

$$\leq \frac{3\delta^0}{(K+1)^{1/4}\frac{1}{b}\sum_{l=1}^{b}\mathbb{E}\left[\mathbb{I}\left(l \in \hat{S}\right)\right]\eta_l} + \frac{6}{K+1}\sum_{i=1}^{b} \frac{\eta_i \bar{\rho}_i}{\beta_i \frac{1}{b}\sum_{l=1}^{b}\mathbb{E}\left[\mathbb{I}\left(l \in \hat{S}\right)\right]\eta_l} P_i^0$$

$$+ \sum_{i=1}^{b} \frac{\eta_i^2}{(K+1)^{3/4}\frac{1}{b}\sum_{l=1}^{b}\mathbb{E}\left[\mathbb{I}\left(l \in \hat{S}\right)\right]\eta_l}\frac{2\bar{\rho}_i}{\beta_i \underline{\rho}_i}$$

$$+ \times \left(\mathbb{E}\left[\left(1 - \mathbb{I}\left(i \in \hat{S}\right)\beta_i\right)L_{i,\hat{S}}^0\right] + \mathbb{E}\left[\left(1 - \mathbb{I}\left(i \in \hat{S}\right)\beta_i\right)L_{i,\hat{S}}^1\right]\mathbb{E}\left[\frac{L_{i,\hat{S}}^0}{L_{i,\hat{S}}^1}\right]\right)$$

$$+ \sum_{i=1}^{b} \frac{\eta_i^2}{2(K+1)^{3/4}\frac{1}{b}\sum_{l=1}^{b}\mathbb{E}\left[\mathbb{I}\left(l \in \hat{S}\right)\right]\eta_l}\left(\mathbb{E}\left[\mathbb{I}\left(i \in \hat{S}\right)L_{i,\hat{S}}^0\right] + \mathbb{E}\left[\mathbb{I}\left(i \in \hat{S}\right)L_{i,\hat{S}}^1\right]\mathbb{E}\left[\frac{L_{i,\hat{S}}^0}{L_{i,\hat{S}}^1}\right]\right)$$

$$+ \sum_{i=1}^{b} \frac{2\mathbb{E}\left[\mathbb{I}\left(i \in \hat{S}\right)\right]\eta_i}{\frac{1}{b}\sum_{l=1}^{b}\mathbb{E}\left[\mathbb{I}\left(l \in \hat{S}\right)\right]\eta_l}\bar{\rho}_i \sigma_i \sqrt{\beta_i}$$

$$\leq \frac{3\delta^0}{(K+1)^{1/4}\frac{1}{b}\sum_{l=1}^{b}\mathbb{E}\left[\mathbb{I}\left(l \in \hat{S}\right)\right]\eta_l} + \frac{6}{(K+1)^{1/2}}\sum_{i=1}^{b} \frac{\eta_i \bar{\rho}_i}{\frac{1}{b}\sum_{l=1}^{b}\mathbb{E}\left[\mathbb{I}\left(l \in \hat{S}\right)\right]\eta_l} P_i^0$$

$$+ \sum_{i=1}^{b} \frac{\eta_i^2}{(K+1)^{1/4}\frac{1}{b}\sum_{l=1}^{b}\mathbb{E}\left[\mathbb{I}\left(l \in \hat{S}\right)\right]\eta_l}\frac{2\bar{\rho}_i}{\underline{\rho}_i}\left(\mathbb{E}\left[L_{i,\hat{S}}^0\right] + \mathbb{E}\left[L_{i,\hat{S}}^1\right]\mathbb{E}\left[\frac{L_{i,\hat{S}}^0}{L_{i,\hat{S}}^1}\right]\right)$$

$$+ \sum_{i=1}^{b} \frac{\eta_i^2}{2(K+1)^{3/4}\frac{1}{b}\sum_{l=1}^{b}\mathbb{E}\left[\mathbb{I}\left(l \in \hat{S}\right)\right]\eta_l}\left(\mathbb{E}\left[\mathbb{I}\left(i \in \hat{S}\right)L_{i,\hat{S}}^0\right] + \mathbb{E}\left[\mathbb{I}\left(i \in \hat{S}\right)L_{i,\hat{S}}^1\right]\mathbb{E}\left[\frac{L_{i,\hat{S}}^0}{L_{i,\hat{S}}^1}\right]\right)$$

$$+ \sum_{i=1}^{b} \frac{2\mathbb{E}\left[\mathbb{I}\left(i \in \hat{S}\right)\right]\eta_i}{(K+1)^{1/4}\frac{1}{b}\sum_{l=1}^{b}\mathbb{E}\left[\mathbb{I}\left(l \in \hat{S}\right)\right]\eta_l}\bar{\rho}_i \sigma_i,$$

where in the last equality we set $\beta_i = \frac{1}{(K+1)^{1/2}}$. Finally, the initialization $M_i^0 = \nabla_i f(X^0; \xi^0)$ guarantees that

$$P_i^0 := \mathbb{E}\left[\left\|\nabla_i f(X^0) - M_i^0\right\|_2\right] \leq \sqrt{\mathbb{E}\left[\left\|\nabla_i f(X^0) - \nabla_i f(X^0; \xi^0)\right\|_2^2\right]} \overset{(4.5)}{\leq} \sigma_i.$$

$\square$

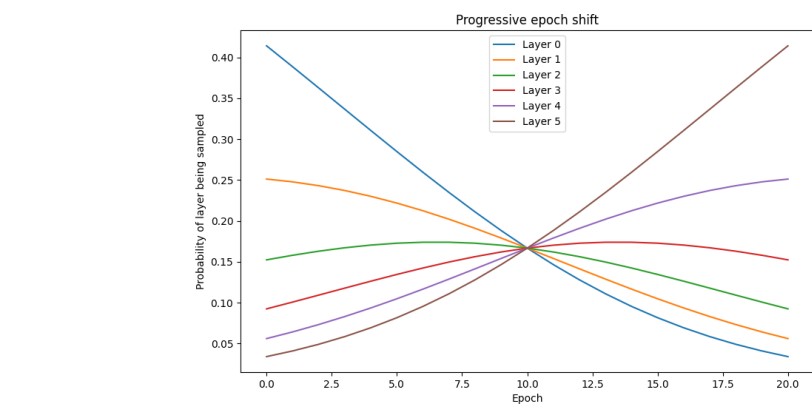

Figure 4: Evolution of the layer sampling distribution as a function of the epochs. Shallow layers are more trained in the first epochs but their probabilities of being sampled decrease with the epochs. This effect can be amplified or reduced by varying the value of $\alpha$; here we chose $\alpha = 0.5$.

## G    EXPERIMENTS

We evaluate Drop-Muon on five benchmarks–NanoGPT (Karpathy, 2023) on the FineWeb10B dataset (Penedo et al., 2024), ResNet50 on Tiny ImageNet, and 3-layer convolutional neural networks (CNNs) of varying capacity on MNIST, Fashion-MNIST, and CIFAR-10.

Experiments were run on various NVIDIA GPUs based on availability. MNIST and Fashion-MNIST experiments were run on Tesla V100-SXM2-32GB. CIFAR-10 experiments were performed on a mix of A100-SXM4-80GB, Tesla P100-PCIE-16GB, and GeForce GTX 1080 Ti GPUs. NanoGPT experiments were conducted on 4 NVIDIA Tesla V100-SXM2-32GB GPUs or 4 NVIDIA A100-SXM4-80GB in a Distributed Data Parallel (DDP) setup. Resnet50 experiments were conducted on 1 NVIDIA Tesla V100-SXM2-32GB GPU.

### G.1    LAYER SAMPLING DISTRIBUTIONS

The choice of layer distribution determines which layers are updated at each iteration and is central to the behavior of Drop-Muon. We considered the following strategies:

**Uniform Distribution.**    Each layer is sampled with equal probability ($p_i = 1/b$ for all $i \in [b]$ in the notation from Section 4).

**Linear Distribution.**    Sampling probability decreases linearly with the layer index.

**Epoch-Shift Distribution.**    Drop-Muon can dynamically adjust which layers are updated at each iteration. The strategy we consider is the *epoch-shift* distribution, which biases training towards shallow layers in the early epochs, gradually shifting focus to deeper layers as training progresses (see Figure 4). Such a schedule balances early feature extraction with later complex feature learning, improving convergence in practice.

Formally, let $b$ denote the total number of layers, $\alpha$ be a constant controlling the sharpness of the bias, and define the training progress as

$$\text{progress} = \frac{\text{epoch}}{\text{max\_epochs}} \in [0, 1].$$

The weight for each layer $i$ is given by

$$w_i = \exp\left(\alpha \left[(1 - \text{progress})(b - 1 - i) + \text{progress} \cdot i\right]\right).$$

We then normalize the weights to obtain a valid probability distribution:

$$W = \sum_{i=0}^{b-1} w_i, \qquad p_i = \frac{w_i}{W}.$$

By adjusting $\alpha$, one can control the rate at which training emphasis shifts from shallow to deeper layers. In Figure 4, we set $\alpha = 0.5$.

**Gaussian-shift Distribution.** The Gaussian-shift distribution places a Gaussian curve over the layer indices, with its mean $\mu$ moving smoothly from the first layer to the last as training progresses. Early in training, the Gaussian is centered near layer 0, so lower layers are sampled more often; later, the bump shifts upward, gradually prioritizing higher layers. The standard deviation $\sigma$ controls how concentrated or spread-out the sampling around the current layer is.

## G.2 EVALUATION METRICS

When reporting results aggregated over runs with multiple random seeds, we use two complementary procedures:

- **Normalized curve averaging:** We normalize wall-clock time for each run so that Muon always ends at $t = 1$, interpolate both Muon and Drop-Muon curves onto this grid, and average over seeds.
- **Time-to-target evaluation:** We measure the wall-clock time to reach fixed accuracy thresholds (e.g. $60\%, 70\%, 80\%, ...$), reporting the ratio between Muon and Drop-Muon. This metric is interpolation-free and directly quantifies practical speedup.

## G.3 NANOGPT EXPERIMENTS

All tuning runs used $3/8$ of the `FineWeb` dataset. We considered the following grid:

- Batch size: $\{64, 128, 256, 512\}$.
- Learning rate constant phase value: $\{0.00026, 0.00036, 0.00046, 0.00056\}$
- Gaussian shift parameter: $\sigma \in \{0.05, 0.25, 0.45, 0.60, 0.85\}$.
- Late wake-up parameter: $\{0.05, 0.25, 0.45\}$.

Tables 1 and 2 summarize the model and optimizer hyperparameters.

Table 1: `NanoGPT`-124M model configuration.

| Hyperparameter | Value |
|---|---|
| Total Parameters | 124M |
| Vocabulary Size | 50,304 |
| Number of Transformer Layers | 12 |
| Attention Heads | 6 |
| Hidden Size | 768 |
| FFN Hidden Size | 3,072 |
| Positional Embedding | RoPE (Su et al., 2024) |
| Activation Function | Squared ReLU (So et al., 2021) |
| Normalization | RMSNorm (Zhang & Sennrich, 2019) |
| Bias Parameters | None |

## G.4 RESNET50 EXPERIMENTS

The learning rates and batch sizes were tuned over the sets $\{0.2, 0.15, 0.05, 0.01, 0.005, 0.001\}$ and $\{2, 64, 128, 256, 512, 1024\}$.

The results are presented in Section 6.

Table 2: Optimizer configuration.

| Hyperparameter | Value |
|---|---|
| Sequence Length | 1024 |
| Batch Size | 256 |
| Optimizer | Drop-Muon |
| Weight Decay | 0 |
| Hidden Layer Norm | Spectral norm |
| Hidden Layer Scale | 50 |
| Newton–Schulz Iterations | 5 |
| Embedding and Head Layers Norm | $\ell_\infty$ norm |
| Embedding and Head Layers Scale | 3000 |
| Initial Learning Rate | For non-compressed: $3.6 \times 10^{-4}$ |
| Learning Rate Schedule | Constant followed by linear decreasing |
| Learning Rate Constant Phase Length | 40% of tokens |
| Momentum | 0.9 |

### G.5 CNN EXPERIMENTS

We next evaluate our algorithm on small, controlled CNN experiments using MNIST, Fashion-MNIST, and CIFAR-10. For each task, we conduct a grid search over the following hyperparameters:

**Batch size and learning rate.** We test batch sizes $\{64, 512, 8192, 16384, 32768\}$ and learning rates $\{0.1, 0.01, 0.001\}$.

**Model depth and capacity.** To assess the impact of network complexity, we train multiple 3-layer CNNs with varying channel configurations: $[8, 16, 32]$, $[16, 32, 64]$, $[64, 128, 256]$, $[128, 256, 512]$, $[64, 16, 8]$, and $[256, 128, 64]$.

**Fixed parameters.** We fix the number of epochs to 20 or 50, momentum to 0.5, and the number of Newton-Schulz iterations to 5 (see Section A.1).

For all $i \in [b]$, we set $\|\cdot\|_{(i)} = \|\cdot\|_{2 \to 2}$, i.e., the spectral norm, consistent with Muon. We study RPT sampling strategy with several smallest-index sampling rules, including uniform and an epoch-shift distribution that gradually shifts probability mass from shallow to deep layers as training progresses. Our baseline is again standard Muon (i.e., Drop-Muon with spectral norms and full-network training) run under identical hyperparameter configurations. This design allows us to isolate the effect of partial network updates while controlling for all other factors. The observed speed-up over Muon could be even greater with dedicated tuning: in our experiments, both methods use the same constant learning rates, whereas theory (Theorems 4.1, 4.2, and 4.4) and the practice of coordinate descent (Section A.3) suggest that Drop-Muon can safely employ larger stepsizes than the full-network Muon baseline. Here, we kept the learning rates identical for both optimizers to isolate the wall-clock benefit of partial layer updates, but additional gains are likely achievable.

For each configuration, we record accuracy as a function of both epochs and wall-clock time.

### G.5.1 MNIST

We first evaluate Drop-Muon on the MNIST dataset. Figure 5 shows a representative run comparing standard full-network Muon training with Drop-Muon using a uniform layer sampling distribution. Although the per-epoch train accuracy of Drop-Muon is initially lower than that of full-layer Muon, the wall-clock training time tells a different story: for training times up to approximately 150 seconds, Drop-Muon consistently achieves higher accuracy. In practice, this means that reaching a train accuracy of 95% or less is faster with Drop-Muon, highlighting its efficiency advantage.

To account for variability across independent runs, we evaluate two aggregation strategies. Figure 6 shows normalized curve averaging, where each run is rescaled to a common time grid. Figure 7

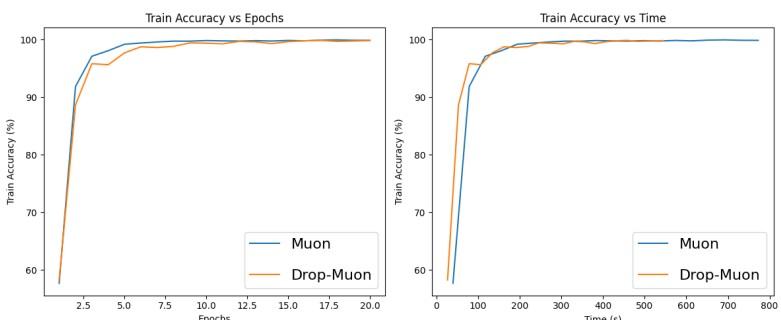

Figure 5: Evolution of the training accuracy for Muon and Drop-Muon with uniform index sampling on `MNIST`. Batch size $= 8192$, learning rate $= 0.1$, channels $= [64, 128, 256]$.

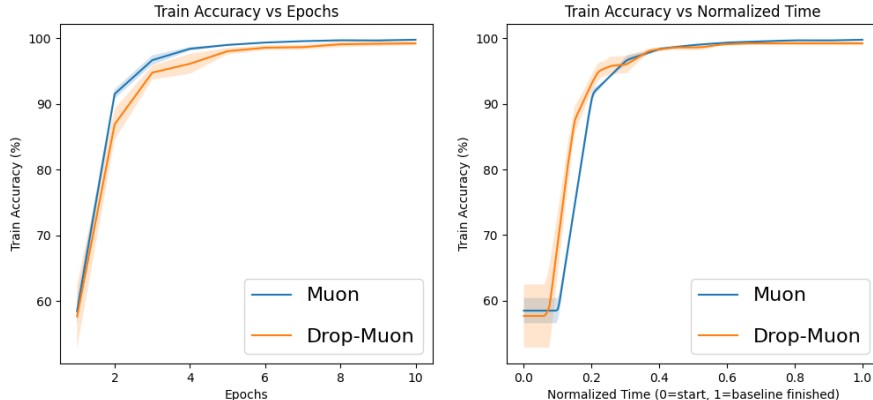

Figure 6: Normalized curve averaging of several runs of Muon and Drop-Muon with uniform index sampling on `MNIST`. Batch size $= 8192$, learning rate $= 0.1$, channels $= [64, 128, 256]$.

(left) presents averaged time-to-target evaluation across multiple seeds, reporting the ratio of Muon time to Drop-Muon time for fixed accuracy thresholds.

The time-to-target results clearly demonstrate a speedup of up to $1.4\times$ across thresholds of $60\%, 70\%, 80\%$, and $99\%$, with slightly lower speedup for the $90\%$ threshold. We note variability across seeds, emphasizing the stochastic nature of layer sampling. Despite this, Drop-Muon consistently provides practical acceleration across training.

### G.5.2 FASHION-MNIST

We repeat the experiment on `Fashion-MNIST`. Drop-Muon again delivers meaningful acceleration: as before, its per-epoch convergence is slightly slower, but it overtakes Muon in wall-clock time. Figure 7 (right) shows aggregated time-to-target results over multiple seeds, with Drop-Muon achieving roughly $1.2\times$ faster convergence on average across accuracy thresholds (we omit the $99\%$ threshold since neither method reaches it).

Figure 8 shows a typical run comparing standard Muon training with Drop-Muon using the epoch-shift layer sampling distribution. Drop-Muon achieves faster progress in wall-clock time.

Normalized curve averaging across multiple seeds is presented in Figure 9. Although the curves appear smoother at higher accuracy, this should not be interpreted as a reduction in variance; early-stage training is inherently noisier due to layer sampling, whereas later training reflects fewer updates per wall-clock time.

Overall, `Fashion-MNIST` experiments reinforce that Drop-Muon provides consistent practical speedups, similar to `MNIST`, while maintaining comparable final accuracy.

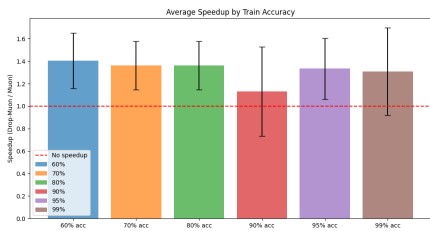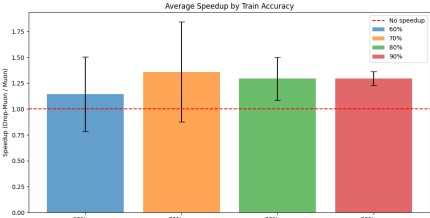

Figure 7: Averaged time-to-target speed-up over multiple runs comparing Muon and Drop-Muon with epoch-shift index sampling. Left: MNIST with batch size = 8192, learning rate = 0.1, and channels = [64, 128, 256]. Right: Fashion-MNIST with batch size = 32768, learning rate = 0.1, and channels = [64, 128, 256].

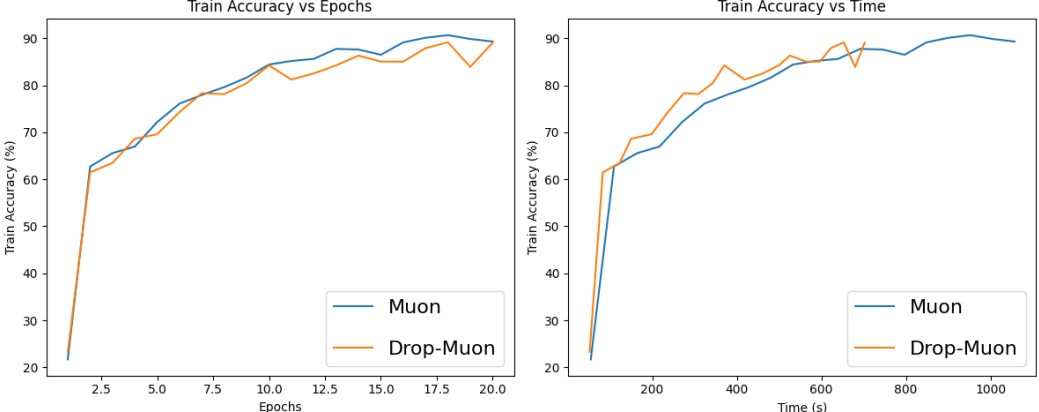

Figure 8: Evolution of the training accuracy for Muon and Drop-Muon with epoch-shift index sampling on Fashion-MNIST. Batch size = 32768, learning rate = 0.1, channels = [64, 128, 256].

### G.5.3   CIFAR-10

In the next set of experiments with CIFAR-10, we use a CNN with channels [128, 256, 512] and insert batch normalization layers between each convolution and ReLU activation to improve training stability.

Figure 10 shows an example run comparing standard full-layer Muon training with Drop-Muon using the epoch-shift layer sampling distribution. While Drop-Muon has slightly lower per-epoch training accuracy, it achieves faster progress in terms of wall-clock time. Indeed, although the absolute gain is smaller than on MNIST or Fashion-MNIST, Drop-Muon still reaches 90% training accuracy earlier in wall-clock time. Figure 11 shows normalized curve averaging across multiple seeds (the flat end of the curve reflects the earlier completion of Drop-Muon).

Time-to-target results in Figure 12 indicate a notable speedup at the 90% train accuracy threshold. However, for lower thresholds (60%, 70%, 80%), speedups are minimal or absent for this configuration. We attribute this limitation primarily to sub-optimal hyperparameter choices.

### G.6   DISCUSSION AND PRACTICAL REMARKS

We summarize several key observations and practical considerations arising from our experiments:

- **Simplicity of implementation:** Drop-Muon can be implemented in just a few lines of code, making it easy to integrate into existing training pipelines.

- **Further benefits from tuning:** Performance can be further improved through dedicated tuning of hyperparameters (see Section 6).

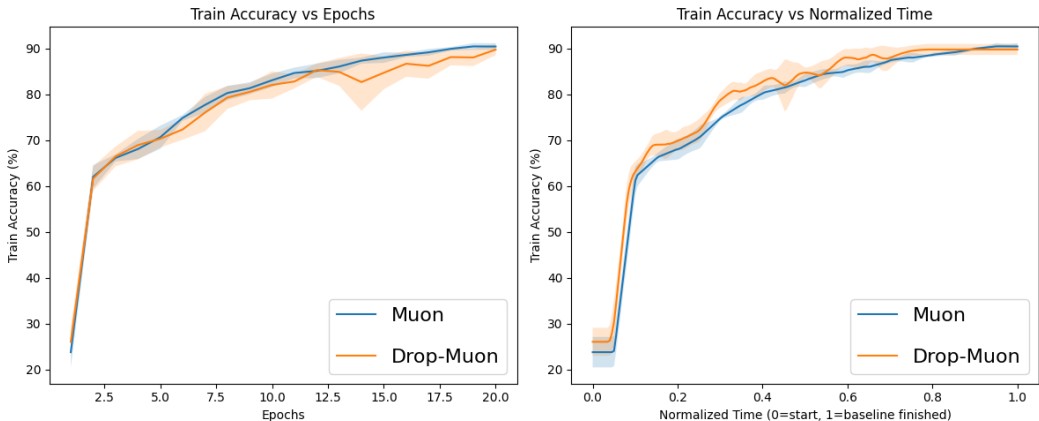

Figure 9: Normalized curve averaging of several runs of Muon and Drop-Muon with epoch-shift index sampling on Fashion-MNIST. Batch size $= 32768$, learning rate $= 0.1$, channels $= [64, 128, 256]$.

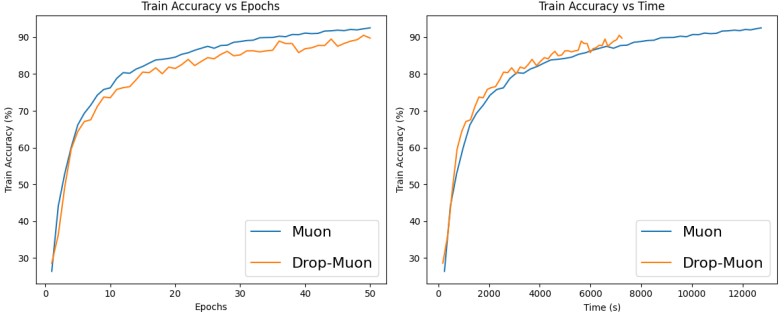

Figure 10: Evolution of the training accuracy for Muon and Drop-Muon with epoch-shift index sampling on CIFAR-10. Batch size $= 8192$, learning rate $= 0.1$, channels $= [128, 256, 512]$.

- **Potential for implementation improvements:**
  - Turning off gradient computations at every iteration can be costly for large models. Sampling the cutoff layer only every few iterations can reduce overhead and improve efficiency.
  - When the batch size or model size is small, the gradient computation may be dominated by the Newton-Schulz routine. In such cases, computing all gradients but orthogonalizing only a subset can be advantageous.

- **Variance across seeds:** The method exhibits non-negligible variance, particularly on smaller datasets. Stabilization mechanisms could mitigate this.

- **Adaptive learning of sampling distributions:** A promising future direction is to learn the layer sampling distribution online, so that it automatically adapts to the training dynamics of a given dataset and architecture.

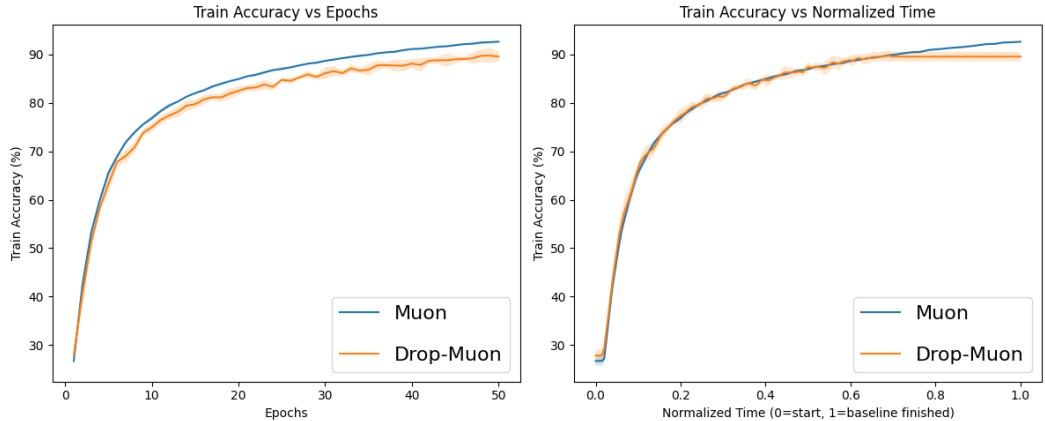

Figure 11: Normalized curve averaging of several runs of Muon and Drop-Muon with epoch-shift index sampling on CIFAR-10. Batch size $= 8192$, learning rate $= 0.1$, channels $= [128, 256, 512]$.

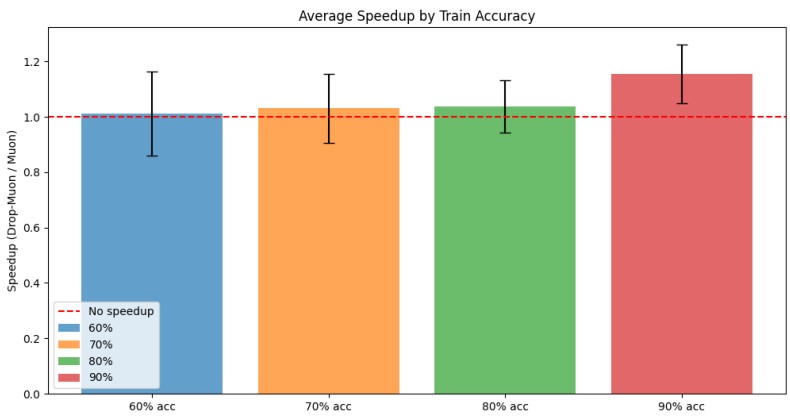

Figure 12: Averaged time-to-target speed-up over multiple runs comparing Muon and Drop-Muon with epoch-shift index sampling on CIFAR-10 with batch size $8192$, learning rate $0.1$, and channels $[128, 256, 512]$.

## H USEFUL FACTS

For all $X, Y \in \mathcal{X}$ and $t > 0$, we have:

$$\left\| \text{LMO}_{\mathcal{B}(0,t)}(G) \right\| = t \tag{48}$$

$$\left\langle G, \text{LMO}_{\mathcal{B}(X,t)}(G) \right\rangle = -t \left\| G \right\|_\star \tag{49}$$

$$\left\langle X, X^\sharp \right\rangle = \left\| X^\sharp \right\|^2, \tag{50}$$

$$\left\| X \right\|_\star = \left\| X^\sharp \right\|, \tag{51}$$

$$\left\langle H, X \right\rangle = \left\| X \right\|_\star, \quad \left\| H \right\| = 1, \quad \forall X \neq 0. \tag{52}$$

where $H \in \partial \left\| \cdot \right\|_\star (X)$ belongs to the subdifferential of the dual norm.

**Lemma H.1** (Variance decomposition). For any random vector $X \in \mathcal{X}$ and any non-random $c \in \mathcal{X}$, we have

$$\mathbb{E}\left[ \left\| X - c \right\|_2^2 \right] = \mathbb{E}\left[ \left\| X - \mathbb{E}[X] \right\|_2^2 \right] + \left\| \mathbb{E}[X] - c \right\|_2^2.$$

**Lemma H.2** ([Riabinin et al. (2025b), Lemma 3](#)). Suppose that $x_1, \ldots, x_p, y_1, \ldots, y_p \in \mathbb{R}$, $\max_{i \in [b]} |x_i| > 0$ and $z_1, \ldots, z_p > 0$. Then

$$\sum_{i=1}^{b} \frac{y_i^2}{z_i} \geq \frac{\left( \sum_{i=1}^{b} x_i y_i \right)^2}{\sum_{i=1}^{b} z_i x_i^2}.$$

**Lemma H.3.** Let Assumption 4.4 hold and let $S \subseteq [b]$. Then, for any vectors $X = [X_1, \ldots, X_b] \in \mathcal{X}$ and $\Gamma = [\Gamma_1, \ldots, \Gamma_b] \in \mathcal{X}$ such that $\Gamma_i = 0$ for all $i \notin S$,

$$|f(X + \Gamma) - f(X) - \langle \nabla f(X), \Gamma \rangle| \leq \sum_{i \in S} \frac{L_{i,S}^0 + L_{i,S}^1 \|\nabla_i f(X)\|_{(i)\star}}{2} \|\Gamma_i\|_{(i)}^2.$$

*Proof.* For all $X \in \mathcal{X}$ we have

$$f(X + \Gamma) = f(X) + \int_0^1 \langle \nabla f(X + \tau\Gamma), \Gamma \rangle \, d\tau$$

$$= f(X) + \int_0^1 \langle \nabla f(X + \tau\Gamma) - \nabla f(X), \Gamma \rangle \, d\tau + \langle \nabla f(X), \Gamma \rangle.$$

Therefore, using the Cauchy-Schwarz inequality

$$
\begin{aligned}
|f(X + \Gamma) - f(X) - \langle \nabla f(X), \Gamma \rangle| &= \left| \int_0^1 \sum_{i=1}^{b} \langle \nabla_i f(X + \tau\Gamma) - \nabla_i f(X), \Gamma_i \rangle_{(i)} \, d\tau \right| \\
&= \left| \int_0^1 \sum_{i \in S} \langle \nabla_i f(X + \tau\Gamma) - \nabla_i f(X), \Gamma_i \rangle_{(i)} \, d\tau \right| \\
&\leq \int_0^1 \sum_{i \in S} \left| \langle \nabla_i f(X + \tau\Gamma) - \nabla_i f(X), \Gamma_i \rangle_{(i)} \right| \, d\tau \\
&\leq \int_0^1 \sum_{i \in S} \|\nabla_i f(X + \tau\Gamma) - \nabla_i f(X)\|_{(i)\star} \|\Gamma_i\|_{(i)} \, d\tau \\
&\overset{(4.4)}{\leq} \int_0^1 \sum_{i \in S} \tau \left( L_{i,S}^0 + L_{i,S}^1 \|\nabla_i f(X)\|_{(i)\star} \right) \|\Gamma_i\|_{(i)}^2 \, d\tau \\
&= \sum_{i \in S} \frac{L_{i,S}^0 + L_{i,S}^1 \|\nabla_i f(X)\|_{(i)\star}}{2} \|\Gamma_i\|_{(i)}^2.
\end{aligned}
$$

$\square$

**Lemma H.4.** Let Assumptions 4.1 and Assumption 4.3 hold and let $S \subseteq [b]$. Then

$$\sum_{i \in S} \frac{\|\nabla_i f(X)\|_{(i)\star}^2}{2 \left( L_{i,S}^0 + L_{i,S}^1 \|\nabla_i f(X)\|_{(i)\star} \right)} \leq f(X) - f^\star$$

for all $X \in \mathcal{X}$.

*Proof.* Let $Y = [Y_1, \ldots, Y_b] \in \mathcal{X}$, where $Y_i = X_i - \frac{\|\nabla_i f(X)\|_{(i)\star}}{L_{i,S}^0 + L_{i,S}^1 \|\nabla_i f(X)\|_{(i)\star}} H_i$ for some $H_i \in \partial \|\cdot\|_{(i)\star} (\nabla_i f(X))$ for $i \in S$ and $Y_i = X_i$ otherwise. By Lemma H.3

$$f(Y) \leq f(X) + \langle \nabla f(X), Y - X \rangle + \sum_{i \in S} \frac{L_{i,S}^0 + L_{i,S}^1 \|\nabla_i f(X)\|_{(i)\star}}{2} \|X_i - Y_i\|_{(i)}^2$$

$$= f(X) + \sum_{i \in S} \langle \nabla_i f(X), Y_i - X_i \rangle_{(i)} + \sum_{i \in S} \frac{L_{i,S}^0 + L_{i,S}^1 \|\nabla_i f(X)\|_{(i)\star}}{2} \|X_i - Y_i\|_{(i)}^2$$

$$= f(X) - \sum_{i \in S} \frac{\|\nabla_i f(X)\|_{(i)\star}}{L_{i,S}^0 + L_{i,S}^1 \|\nabla_i f(X)\|_{(i)\star}} \langle \nabla_i f(X), H_i \rangle_{(i)}$$

$$+ \sum_{i \in S} \left( \frac{L_{i,S}^0 + L_{i,S}^1 \|\nabla_i f(X)\|_{(i)\star}}{2} \frac{\|\nabla_i f(X)\|_{(i)\star}^2}{\left(L_{i,S}^0 + L_{i,S}^1 \|\nabla_i f(X)\|_{(i)\star}\right)^2} \|H_i\|_{(i)}^2 \right)$$

$$\overset{(52)}{=} f(X) + \sum_{i \in S} \left( -\frac{\|\nabla_i f(X)\|_{(i)\star}^2}{L_{i,S}^0 + L_{i,S}^1 \|\nabla_i f(X)\|_{(i)\star}} + \frac{\|\nabla_i f(X)\|_{(i)\star}^2}{2\left(L_{i,S}^0 + L_{i,S}^1 \|\nabla_i f(X)\|_{(i)\star}\right)} \right)$$

$$= f(X) - \sum_{i \in S} \frac{\|\nabla_i f(X)\|_{(i)\star}^2}{2\left(L_{i,S}^0 + L_{i,S}^1 \|\nabla_i f(X)\|_{(i)\star}\right)},$$

and hence

$$\sum_{i \in S} \frac{\|\nabla_i f(X)\|_{(i)\star}^2}{2\left(L_{i,S}^0 + L_{i,S}^1 \|\nabla_i f(X)\|_{(i)\star}\right)} \le f(X) - f(Y) \le f(X) - f^\star.$$

$\square$

**Lemma H.5.** Let Assumptions 4.1 and Assumption 4.3 hold and let $S \subseteq [b]$. Then, for any $x_i > 0$, $i \in [b]$, we have

$$\sum_{i \in S} x_i \|\nabla_i f(X)\|_{(i)\star} \le 4 \max_{i \in S}(x_i L_{i,S}^1) \left(f(X) - f^\star\right) + \sum_{i \in S} \frac{x_i L_{i,S}^0}{L_{i,S}^1}$$

for all $X \in \mathcal{X}$.

*Proof.* Applying Lemma H.4 and Lemma H.2 with $y_i = \|\nabla_i f(X)\|_{(i)\star}$, $z_i = L_{i,S}^0 + L_{i,S}^1 \|\nabla_i f(X)\|_{(i)\star}$ and any $x_i > 0$, we have

$$2\left(f(X) - f^\star\right) \ge \sum_{i \in S} \frac{\|\nabla_i f(X)\|_{(i)\star}^2}{L_{i,S}^0 + L_{i,S}^1 \|\nabla_i f(X)\|_{(i)\star}}$$

$$\ge \frac{\left(\sum_{i \in S} x_i \|\nabla_i f(X)\|_{(i)\star}\right)^2}{\sum_{i \in S} x_i^2 L_{i,S}^0 + \sum_{i \in S} x_i^2 L_{i,S}^1 \|\nabla_i f(X)\|_{(i)\star}}$$

$$\ge \frac{\left(\sum_{i \in S} x_i \|\nabla_i f(X)\|_{(i)\star}\right)^2}{\sum_{i \in S} x_i^2 L_{i,S}^0 + \max_{i \in S}(x_i L_{i,S}^1) \sum_{i \in S} x_i \|\nabla_i f(X)\|_{(i)\star}}$$

$$\ge \begin{cases} \frac{\left(\sum_{i \in S} x_i \|\nabla_i f(X)\|_{(i)\star}\right)^2}{2 \sum_{i \in S} x_i^2 L_{i,S}^0} & \text{if } \frac{\sum_{i \in S} x_i^2 L_{i,S}^0}{\max_{i \in S}(x_i L_{i,S}^1)} \ge \sum_{i \in S} x_i \|\nabla_i f(X)\|_{(i)\star}, \\ \frac{\sum_{i \in S} x_i \|\nabla_i f(X)\|_{(i)\star}}{2 \max_{i \in S}(x_i L_{i,S}^1)} & \text{otherwise.} \end{cases}$$

Therefore,

$$\sum_{i \in S} x_i \|\nabla_i f(X)\|_{(i)\star} \le \max\left\{ 4 \max_{i \in S}(x_i L_{i,S}^1) \left(f(X) - f^\star\right), \frac{\sum_{i \in S} x_i^2 L_{i,S}^0}{\max_{i \in S}(x_i L_{i,S}^1)} \right\}$$

$$\le 4 \max_{i \in S}(x_i L_{i,S}^1) \left(f(X) - f^\star\right) + \frac{\sum_{i \in S} x_i^2 L_{i,S}^0}{\max_{i \in S}(x_i L_{i,S}^1)}$$

$$\leq \quad 4 \max_{i \in S}(x_i L^1_{i,S}) \left(f(X) - f^\star\right) + \sum_{i \in S} \frac{x_i L^0_{i,S}}{L^1_{i,S}}.$$

$\square$

**Note on LLM Usage.**   Large Language Models were used to assist in polishing the writing of the manuscript. LLM assistance did not contribute to the scientific content of the paper.

