# OpenReview forum: "Drop-Muon: Update Less, Converge Faster"
_ICLR.cc/2026/Conference — Submitted to ICLR 2026_

### Official Review · Reviewer_XX3C · 2025-10-16

**Soundness:** 2
**Presentation:** 2
**Contribution:** 2
**Rating:** 2
**Confidence:** 2

**Summary:**

In this work, the authors challenge the assumption of all-layer udpates, showing that fullnetwork updates can be fundamentally suboptimal, both in theory and in practice. They introduce a non-Euclidean Randomized Progressive Training method—DropMuon—a framework that updates only a subset of layers per step according to a randomized schedule, combining the efficiency of progressive training with layer-specific non-Euclidean updates for top-tier performance. They provide rigorous convergence guarantees under both layer-wise smoothness and layer-wise (L0 , L1 )-smoothness. They also do CNN experiments on MNIST or CIFAR, and show that Drop-Muon consistently outperforms standard full-network Muon.

**Strengths:**

MuonDrop samples a random subset of layers according to a user-defined distribution D and updates only the parameters of the selected layers , keeping all other layers frozen. This is an interesting idea.

Establishes convergence guarantees for Drop-Muon under two regimes: layer-wise smoothness (Theorem 4.1) and layer-wise (L0 , L1 )–smoothness (Theorem 4.2). And the authors claim that provide the first convergence guarantees for progressive training-style methods in the non-smooth setting.

Did CNN experiments on MNIST or CIFAR, and show that Drop-Muon consistently outperforms standard full-network Muon.

**Weaknesses:**

(1) Some parts of the writing is not friendly to non-theory background people like me, such as line 099 "admits two equivalent formulations" or line 130/131 "when ||.||_(i)=||.||_2….". And I don't quite get the motivation of only updating a subset of layers. And naturally, it is hard for me to grasp section 4.

(2) The empirical result is conducted on relatively small datasets (MINIST CIFAR), with CNN. It would be nice to try larger dataset, or adding language model experiments.

(3) Only MOUN is compared as baseline, it would be nice to add more baseline optimization methods.

**Questions:**

In addition to weakness (1), what do you mean by non-smooth setting in line 192?

Did you also search for hyper parameter for the baseline? (line 3220)

---

> ### Author Response · Authors · 2025-11-19
>
> We would like to thank the reviewer for their comments. We appreciate the recognition of the paper's strengths and the acknowledgment of the novelty and significance of our contributions. We are happy to provide clarifications to explain the results.
>
> ---
>
> **Weaknesses:**
> > Some parts of the writing is not friendly to non-theory background people like me, such as line 099 "admits two equivalent formulations" or line 130/131 "when $\|.\|_{(i)}=\|.\|_2$...". And I don't quite get the motivation of only updating a subset of layers. And naturally, it is hard for me to grasp section 4.
>
> Our paper is mainly theoretical in nature, and we understand that the concepts may not be immediately accessible to readers without such a background. We appreciate the reviewer's effort in going through these details. We believe that the paper strikes a good balance between intuitive explanations and rigorous mathematical derivations.
>
> **We do not consider the theoretical rigor and focus of our work to be a weakness. Difficulty in understanding the paper stems from a mismatch between the reviewer's background and the paper's intended audience, not from any deficiency in the work itself. The paper's contributions remain valid, significant, and well-supported.** There is no basis for dismissing our contributions or questioning the quality of the paper, and we hope the reviewer keeps this in mind when reassessing our work.
>
> We would like to clarify the main points and the motivation behind our Drop-Muon framework, which aims to provide evidence for the benefits of partial network updates both in theory and in practice:
>
> - In theory: We show that the cost function (8), where "cost" abstracts computational resources or runtime, is minimized by $(p_1, p_2, \ldots, p_b)=(1, 0, \ldots, 0)$ (i.e., full-network updates) if and only if a certain restrictive condition on the (generalized) smoothness constants is satisfied (Theorems 4.3 and E.2). When this condition does not hold, partial updates reduce the cost, motivating our randomization strategy.
>
> - In practice: Our experiments show that Drop-Muon outperforms Muon in terms of wall-clock time, as evidenced in Figures 1, 2, 3, 5, 6, 7, 9; Figure 8 shows comparable performance. This provides a clear support for only updating a subset of layers. Here, we would like to highlight that our experiments were deliberately designed to isolate the wall-clock *benefits of partial layer updates*, without introducing confounding differences in other hyperparameters. For this reason, both Drop-Muon and Muon use the same learning rate. However, both our theoretical analysis and prior work on coordinate-descent–style methods indicate that Drop-Muon could use larger stepsizes than the full-update Muon baseline, potentially making it even faster. Even without exploiting this advantage, Drop-Muon already achieves superior wall-clock performance.
>
> The intuition for the superiority of our method is the following: while Drop-Muon can be slightly worse than Muon in terms of *iteration count* (see the left panels of Figures 1 and 3 - this is expected: because Drop-Muon updates only a subset of the layers per iteration, it naturally makes less progress per iteration than a full-update method). However, iteration count is
> *not* the target metric: each Drop-Muon iteration is cheaper than a full Muon iteration. That is why our primary comparison is based on wall-clock time (e.g., right panels of Figures 1 and 3). Under this metric, **Drop-Muon consistently outperforms Muon, as predicted by our theory**. The key insight is that the modest reduction in per-iteration progress is more than offset by the reduced per-iteration computational cost, resulting in **faster overall convergence**, even with non-optimized parameters.
>
> We hope this clarifies the motivation for updating only a subset of layers.
>
> Regarding specific lines in the paper:
>
> - Line 099 ("admits two equivalent formulations") means that the update rule (3) can be written in two mathematically equivalent ways.
>
> - Line 130 "when $\|.\|_{(i)}=\|.\|_2$..." specifies that the norms defining the layer-wise ball constraints are taken to be Euclidean norms. Using different norms results in different algorithmic updates, (for example, Muon uses the spectral norm for hidden layers).

---

> > ### Author Response · Authors · 2025-11-19
> >
> > > The empirical result is conducted on relatively small datasets (MINIST CIFAR), with CNN...
> >
> > We appreciate the reviewer's suggestion. In this initial response, we focus on addressing all non-experimental comments in detail; additional comments on experiments will follow in subsequent rebuttal updates.
> >
> > We would like to briefly reiterate the main aim of our work. **This paper is primarily theoretical, and it should be judged on that basis The central contribution stands entirely independently of experiments.**: we identify a fundamental issue in deep model training and **rigorously prove that the predominant practice of full-network updates is not optimal** (Theorems 4.3 and E.2). This is the core of the paper. Even with no experiments, the theoretical result is strong, and the evaluation should be based on that, just as experimental papers are evaluated on experiments even when they contain no theory.
> >
> > That said, we did go a step further and provided empirical validation showing that our theory is reflected in practice on smaller but representative setups, demonstrating that our method is implementable and promising without requiring industrial-scale compute. Given the reviewer’s interest, we are already running additional larger experiments and will share the results once they complete. However, we would like to reiterate that the paper is **fundamentally theoretical**, and it should be judged as such.
> >
> > > Only MOUN is compared as baseline, it would be nice to add more baseline optimization methods.
> >
> > Our philosophy is to provide experiments that directly relate to and support the theory, demonstrating its predictive power. The goal of our paper was to provide, to our knowledge, the first systematic investigation of the common practice of updating all layers of a network at every iteration. We rigorously show, both theoretically and empirically, that this design choice is generally suboptimal. Therefore, the natural baseline for evaluating Drop-Muon is full-network Muon. While we could include additional optimizers (e.g., Adam or AdamW) for comparison, prior work has already established that Muon outperforms them, so we focused on the most relevant baseline.
> >
> > **Questions:**
> > > In addition to weakness (1), what do you mean by non-smooth setting in line 192?
> >
> > We understand that the terminology may be confusing for readers outside theoretical optimization research. To clarify: the standard Lipschitz smoothness assumption states that $\|\nabla f(x)-\nabla f(y)\|\leq L\|x-y\|$ for all $x,y$. This is a standard assumption widely used in optimization literature. However, as discussed in Section 4 and Appendix A.2, this assumption is often violated for deep learning objectives. To address this, the optimization community has proposed several relaxations that are better suited for such (non-smooth) settings. One such relaxation is $(L_0, L_1)$-smoothness, which assumes $\|\nabla f(x)-\nabla f(y)\|\leq (L_0+L_1\|\nabla f(y)\|)\|x-y\|$ for all $x,y$. This formulation is much more flexible and has been shown to better capture the behavior of deep learning loss landscapes.
> > Our work builds upon this assumption (see Assumptions 4.3 and 4.4, and Lemma H.3). We extend standard $(L_0, L_1)$-smoothness by allowing separate constants for different subsets of layers. This provides tighter, subset-specific bounds and more accurately reflects layer interactions within the network, without restricting the function class.
> >
> > Long story short, we do not assume smoothness, and hence the setting considered in Theorems 4.2 and 4.4 is non-smooth.
> >
> > > Did you also search for hyper parameter for the baseline? (line 3220)
> >
> > Our experiments were designed to isolate the wall-clock time benefits of partial layer updates. To ensure this, Drop-Muon and Muon use the same hyperparameters.
> >
> > ---
> >
> > We hope these explanations clarify the motivations, theoretical contributions, and empirical results of the paper. **All results are well-supported and highlight a new, efficient approach to training deep networks that challenges the status quo of full-network updates.**
> > While some theoretical aspects may be challenging for readers without a theoretical background, this does not reflect any shortcomings in the paper itself. We hope the reviewer will take this into account when reconsidering their score. The framework, derivations, and experiments are fully correct and valid. If any part remains unclear, we are happy to provide further explanations.
> >
> > We kindly ask for a reconsideration of the current evaluation, which we believe does not adequately reflect the quality and impact of our research.

---

> > > ### Comment · Reviewer_XX3C · 2025-11-24
> > >
> > > Thank you for your response. I apologize that I may lack the background to have a good grasp of your work.

---

> > > > ### Author Response · Authors · 2025-11-24
> > > >
> > > > Thank you for your reply. We fully understand that the theoretical nature of the submission may fall outside your expertise. However, awarding a score of 2 on that basis does not constitute a fair evaluation of our work. In light of your acknowledgement, we would be grateful if you could reconsider your score, as it does not reflect the strength of the paper's contributions.

---

> > > > > ### Author Response · Authors · 2025-12-03
> > > > >
> > > > > As promised, we have added larger-scale experiments: training NanoGPT on the FineWeb dataset and an image-classification task with ResNet50 (Section 6). These results confirm the conclusions of the previous smaller experiments (now moved to the appendix) and clearly demonstrate that **Drop-Muon consistently outperforms Muon in wall-clock time**. The paper has been updated accordingly.
> > > > >
> > > > > Overall, we believe that all points raised in the review have been addressed. We hope that the significance of our contributions is now clear.

---

### Official Review · Reviewer_kUBn · 2025-10-23

**Soundness:** 2
**Presentation:** 3
**Contribution:** 2
**Rating:** 2
**Confidence:** 3

**Summary:**

This paper introduces Drop-Muon, a new randomized layer-wise optimizer that updates only a subset of layers at each iteration rather than the full network. The method extends the Muon optimizer framework by incorporating Randomized Progressive Training (RPT), sampling a random minimal index $s_{k}$ and updating all layers from $s_{k}$ to $b$, thereby reducing computational cost while maintaining convergence guarantees.

The authors make three key claims:
* Full-network updates are not theoretically optimal unless layer smoothness constants satisfy a rare equality.
* Theoretical results include convergence guarantees under both layer-wise smoothness and generalized $(L_{0}, L_{1})$-smoothness, for both deterministic and stochastic settings.
* Empirical results on CNNs (MNIST, Fashion-MNIST, CIFAR-10) show Drop-Muon matches Muon’s accuracy while converging up to 1.4× faster in wall-clock time.

**Strengths:**

* Novel perspective: The work challenges a long-standing assumption that updating all layers per step is necessary for optimal convergence, providing a fresh theoretical and empirical lens.

* Rigorous theoretical analysis: The paper presents detailed convergence proofs across multiple smoothness regimes and establishes a cost model linking layer sampling distribution to compute efficiency.

* Practical implementation: The method is conceptually simple, compatible with existing architectures, and demonstrated to reduce wall-clock time without loss of accuracy.

* Theoretical-experimental coherence: The experiments, though small, confirm the theoretical insight that selective layer updates can outperform full-network training.

**Weaknesses:**

$\textbf{Weaknesses in theoretical results}$
* $\textbf{Layer-wise optimal tuning.}$ Likewise, most theoretical results count for optimally tuning the learning rate as some problem property-dependent values. It makes sense if a single learning rate is the only (or dominant) term that requires such tuning.
  * Theorem 4.1 assumes layer-wise optimal learning rates, which depend on unknown smoothness constants. This assumption is rarely satisfied in practice and undermines the practical interpretability of the theoretical results. Moreover, layer-wise learning rate tuning greatly reduces usability compared to single global tuning.
* $\textbf{No clear theoretical advantage for layer-wise randomization.}$
   * The results in Theorem 4.1–4.3 effectively decompose into independent layer analyses with double summations $(K,b)$, suggesting the benefits come from parallel or reduced-cost computation rather than a deeper coupling among layers.
   * The proof structure reads more like “an analysis per-layer, concatenated,” rather than showing true synergy among layers.

---

$\textbf{Cost-optimality justification, in section 4.1.1 COST OPTIMIZATION, is fragile.}$
* I believe a faithful result of such a claim, “full-network updates are not in general optimal”, is significant. However, the justification in this study is rather fragile.
* The claim is only supported by an idealized cost model. While it proves non-optimality theoretically, it does not guarantee that the proposed RPT sampling is superior.
* The argument seems to restate that the largest smoothness constant limits the global learning rate, a well-known result, without clearly showing why randomization improves optimization.
   * For me, it seems to restate that "ill-conditioned problems, e.g., quadratic with Hessian $H$, the largest learning rate is limited by the largest $L_{\max}$, i.e., $LR = 1/L_{\max} = 1/\lambda_{\max}(H)$". Here, the "cost" is also limited by the largest $L$.

---

$\textbf{Empirical Verification}$
* I fully understand that evaluating a new optimizer is inherently challenging. However, a reasonably comprehensive assessment helps prevent superficial or unconvincing claims and demonstrates respect for the reviewers. Moreover, a study that presents rigorous theoretical results deserves an equally thorough empirical verification.
   * The experiments are limited to small CNNs on MNIST, Fashion-MNIST, and CIFAR-10, which are far from the common deep learning setups (e.g., ViT, BERT, GPT-2), motivating the paper. The current benchmarks are too lightweight to demonstrate practical scalability.
   * I am not asking for months of training, but rather for experiments that extend beyond the few hours reported in this work, ideally to a few days, to better reflect realistic training scenarios.

---

$\textbf{Conceptual Weaknesses}$
* The justification of the “randomized progressive” sampling rule (updating all layers from $s_{k}$ to $b$ appears primarily motivated by backprop efficiency, not theoretical necessity. A completely random subset could better illustrate the claimed generality of the framework.
* The paper’s theoretical innovation largely builds on the existing Muon and RPT frameworks with minimal new structural design.

---

Overall, this paper presents an interesting theoretical argument and a clean framework questioning full-network updates, with rigorous proofs and a well-structured manuscript. However, the practical implications and empirical validation are too weak for such a bold claim.

With stronger empirical demonstrations and a clearer justification for how randomization improves beyond computational convenience, this work could become an impactful contribution to the optimizer design literature.

**Questions:**

* In Randomized Progressive Training, the expected number of updated layers is roughly $b/2$. Is this ratio fixed or tunable? Would different expected update ratios affect convergence and cost trade-off?

* Why does Drop-Muon require updating from the sampled $s_{k}$ to the last layer? Would fully random subset updates (non-contiguous) provide stronger or weaker convergence in practice?

---

> ### Author Response · Authors · 2025-11-19
>
> We thank the reviewer for the time spent evaluating our work and acknowledging its strengths: novelty, practicality and rigorousness. We appreciate the feedback and the opportunity to clarify several points about our method and results.
>
> ---
>
> **Weaknesses:**
>
> > Layerwise optimal tuning...
>
> Yes, all algorithms in the Muon family allow for per-layer learning rate tuning. This is their strength rather than a limitation. As argued in our paper and prior works, different layers in a neural network serve fundamentally different roles and interact with the loss landscape in different ways. Treating them identically leads to suboptimal behavior, whereas using layer-specific control aligns better with the underlying geometry of the problem and has been shown to yield superior empirical performance (for Muon and related optimizers). Importantly, the framework allows but does not require per-layer learning rates: users can still employ a single global learning rate or grouped learning rates if they prefer.
>
> It is also correct that achieving theoretically optimal performance requires access to problem-dependent constants (here, generalized smoothness parameters). We agree that this is not ideal; however, this is a universal limitation across optimization theory. The theory behind all standard optimizers, e.g., SGD, Adam, AdamW, Muon, depends on such unknown constants. Nonetheless, theoretical analyses of these methods remain highly influential and valuable for understanding their behavior. Our work is no different.
>
> > No clear theoretical advantage for layer-wise randomization...
>
> Reiterating from the previous point, layer-wise treatment is a strength, not a limitation of our analysis. It reflects the inherent modular structure of neural networks and allows us to incorporate the mechanics of backpropagation directly into the theoretical framework. Unlike analyses that flatten the network into a single parameter vector, our approach models the interdependence of layers in a principled way: computing the gradient for layer $i$ necessarily requires computing the gradient for layer $i+1$, and our theoretical framework captures this relationship explicitly. The double summations are a natural consequence of this structure, not an artifact of "analysis per-layer, concatenated" proofs. The suggestion that the proofs lack "true synergy among layers" is also not accurate. The layer-wise, subset-specific generalized smoothness assumptions and the cost model all interact in a way that couples the behavior of different layers. Our results depend on these interactions and cannot be decomposed into independent per-layer analyses.
>
> Most importantly, **the theoretical superiority of layer-wise randomization is clearly established in Theorems 4.3 and E.2**, where we prove that **the cost function (8)** (where "cost" abstracts computational resources or runtime) **is minimized by $(p_1, p_2, \ldots, p_b)=(1, 0, \ldots, 0)$ (i.e., full-network updates) if and only if a certain restrictive condition on the (generalized) smoothness constants is satisfied**. When this condition does not hold, partial updates reduce the cost, providing a rigorous theoretical motivation for our randomization strategy.
>
> Overall, we respectfully disagree that this criticism has any merit.
>
> > Cost-optimality justification in section 4.1.1 is fragile...
>
> All models are wrong, but we believe that our cost model is not only useful but also realistic. It accounts for all major components of neural network training: forward pass, backward pass, gradient transformation through sharp operator, parameter updates, and fixed per-iteration overhead (e.g., data loading). If the reviewer believes any essential component is missing, we would greatly appreciate the opportunity to incorporate it.
>
> By design, **every theoretical analysis relies on assumptions**, whether about the cost model, the function class, or other. There is simply no way to prove meaningful results otherwise. **Our assumptions are realistic and minimal**, and they enable us to establish provable superiority of our method.
>
> Our model enables us to **prove that the widely adopted practice of updating all layers at every iteration is not generally optimal** (again, see Theorems 4.3 and E.2). We strongly disagree with the claim that "it does not guarantee that the proposed RPT sampling is superior". Yes, it does. Theorems 4.3 and E.2 explicitly establish this. To describe this result as "fragile" is simply incorrect.

---

> > ### Author Response · Authors · 2025-11-19
> >
> > > The argument seems to restate that the largest smoothness constant limits the global learning rate...
> >
> > Again, Theorems 4.3 and E.2 clearly explain why randomization improves optimization performance. Of course, the optimal parameters cannot be selected freely, as optimizers require parameters within specific regimes. In this framework, the optimal parameter values are determined by the (generalized) smoothness constants, consistent with other methods analyzed under similar conditions.
> >
> > Having said that, **our results extend far beyond the classical setting**. Our results show not only that smoothness limits stepsizes (which is indeed well known), but that the optimal allocation of computation across layers depends on the interplay between layer-specific generalized smoothness constants. This mechanism is **fundamentally different from the standard conditioning argument in quadratic optimization**.
> >
> > > Empirical verification...
> >
> > We appreciate the reviewer's suggestion. In this initial response, we focus on addressing all non-experimental comments in detail; additional comments on experiments will follow in subsequent rebuttal updates.
> >
> > We would like to briefly reiterate the main aim of our work. **This paper is primarily theoretical, and it should be judged on that basis The central contribution stands entirely independently of experiments.**: we identify a fundamental issue in deep model training and **rigorously prove that the predominant practice of full-network updates is not optimal** (Theorems 4.3 and E.2). This is the core of the paper. Even with no experiments, the theoretical result is strong, and the evaluation should be based on that, just as experimental papers are evaluated on experiments even when they contain no theory.
> >
> > That said, we did go a step further and provided empirical validation showing that our theory is reflected in practice on smaller but representative setups, demonstrating that our method is implementable and promising without requiring industrial-scale compute. Given the reviewer’s interest, we are already running additional larger experiments and will share the results once they complete. However, we would like to reiterate that the paper is **fundamentally theoretical**, and it should be judged as such.
> >
> > > Conceptual weaknesses. The justification of the "randomized progressive" sampling rule...
> >
> > Updating all layers from the sampled one to the end of the network is the most natural choice in a deep learning setting, given the mechanics of backpropagation. However, **our paper already provides a general framework that accommodates arbitrary sampling schemes**. As stated in the contributions section, *Drop-Muon supports virtually any layer sampling scheme (Appendix C). In the main part of this paper, we focus on Randomized Progressive Training (RPT), a natural strategy aligned with backpropagation mechanics*. The RPT results are special cases of the general theory that can be found in the appendix.
> >
> > > The paper's theoretical innovation largely builds on the existing Muon and RPT frameworks...
> >
> > Our method is inspired by Muon/Gluon and RPT, but the combination of these ingredients is entirely novel. **No prior work investigates partial network updates within the Muon/Gluon family, and obviously no existing papers demonstrate that such partial updates can be superior**. We present **the first theoretical and empirical results showing that this approach outperforms the full-network baseline**. We also introduce **new generalized smoothness assumptions tailored to deep learning** and provide **the first convergence results for progressive training in the stochastic and non-smooth regime**. All of these represent **substantial contributions beyond prior work**.

---

> > > ### Author Response · Authors · 2025-11-19
> > >
> > > **Questions:**
> > >
> > > > In Randomized Progressive Training, the expected number of updated layers is roughly $b/2$. Is this ratio fixed or tunable? Would different expected update ratios affect convergence and cost trade-off?}
> > >
> > > If we only consider uniform sampling, then the expected number of updated layers is indeed $b/2$. For other sampling strategies, this is not necessarily the case. The only requirement on the probabilities defining the layer sampling distribution is that $p_1>0$, i.e., that all layers are updated with nonzero probability. Users can choose these probabilities freely. Of course, not all configurations will work equally well.
> > > As an extreme example, suppose we assign probabilities $p_1=\ldots=p_{b-1}\approx 0$ and $p_b=1- \sum_{i=1}^{b-1} p_i \approx 1$, so that we almost always update only the last layer. One clearly should not expect fast convergence under such a sampling strategy.
> > > Our theory captures this behavior: consider Theorem 4.1. In this extreme case, we have $w_i = \sum_{s=1}^i \frac{p_s}{2 L_{i, \{s,\dots,b\}}^0} \approx 0$ for $i=1, \ldots, b-1$ and $w_b = \sum_{s=1}^b \frac{p_s}{2 L_{i, \{s,\dots,b\}}^0} \approx \frac{p_b}{2 L_{i, \{b\}}^0}$.
> > > Thus, the gradients for layers $i=1, \ldots, b-1$ receive weights close to zero. Ensuring that all gradient norms reach the desired accuracy therefore requires substantially more iterations compared with using a more balanced sampling distribution (as discussed in Appendix E.1.1).
> > >
> > > > Why does Drop-Muon require updating from the sampled $s_k$ to the last layer? Would fully random subset updates (non-contiguous) provide stronger or weaker convergence in practice?}
> > >
> > > As noted above, **Drop-Muon already supports arbitrary parameter sampling schemes**. However, updating only a non-contiguous subset of layers is not effective in deep learning settings: because of backpropagation, gradients must be computed for all layers from $s_k$ to $b$ anyway. Ignoring updates for these layers would therefore waste computation. Progressive updates simply leverage the computation that is already unavoidable.
> > >
> > > ---
> > >
> > > We thank the reviewer again for their feedback. We hope that our detailed responses have thoroughly addressed all the concerns raised. We have demonstrated that **the perceived weaknesses either do not apply or are already accounted for within our framework**, and that **our theoretical contributions remain strong and well-supported**. We appreciate the reviewer's engagement with our work and hope that, in light of these clarifications, it is clear that there are no remaining concerns. Given the **reviewer’s own acknowledgment that our work "presents an interesting theoretical argument and a clean framework questioning full-network updates, with rigorous proofs and a well-structured manuscript"**, we kindly ask for a reconsideration of the current evaluation, which we believe does not adequately capture the quality and impact of our research. **As the reviewer themselves noted in the "Strengths" section, our work is novel, rigorous, and practically relevant. We would be extremely grateful if the evaluation could reflect this.**

---

> > ### Comment · Reviewer_kUBn · 2025-11-24
> >
> > Thank you for the clarifications. I’ll keep the score for the following reasons.
> >
> > ---
> >
> > $\textbf{Layerwise optimal tuning...}$
> >
> > "Layer-wise treatment is a strength": this claim is not convincing. In practice, nobody can afford to tune a separate learning rate for each layer, as the hyperparameter search space quickly becomes unmanageable.
> > * As a result, the proposed method would likely collapse to a single global learning rate in any realistic setting.
> > * Manually tuning layer-wise learning rates is not practical. A more meaningful advantage would be a tuning-free mechanism, or an approach where learning rates are automatically adapted based on layer properties, rather than requiring explicit per-layer tuning.
> >
> > ---
> >
> > $\textbf{No clear theoretical advantage for layer-wise randomization and fragile cost-optimality argument (Section 4.1.1)}$
> >
> > While the paper states (and I agree) that always updating all layers may be sub-optimal, Theorems 4.3 and E.2 do not establish a positive guarantee for the proposed alternative:
> > * These theorems only suggest that full-layer updates can be flawed in some cases.
> > * They do not show that partial updates are better or even convergent. In fact, partial updates could be worse.
> >
> > An analogy: suppose there are three possible investment strategies. You prove that strategy 1 leads to a 10% loss. This alone does not justify choosing strategy 2 or 3, either of which may lead to a 20% loss. Rationally, one would still choose strategy 1.
> >
> > Similarly, $\textbf{disproving the optimality of full-layer updating does not imply the advantage of layer-wise randomization }$. The current motivation contains a logical gap.
> >
> > ---
> >
> > Because of this logical gap, the theoretical significance of the work is substantially weakened. What is missing is a result that shows:
> > >**Layer-wise randomization can outperform full-layer updating under certain conditions.**
> >
> > No such conclusion is provided.
> >
> > ---
> >
> > $\textbf{Empirical verification...}$
> > * Overall, the experimental section is relatively weak and under-supported. The analysis of the number of updated layers, the sampling strategy, and the resulting performance does not sufficiently substantiate the empirical contribution of this study.

---

> > > ### Author Response · Authors · 2025-12-03
> > >
> > > We thank the reviewer for the response. However, we cannot agree with the points raised, and we believe the reviewer has misunderstood both the paper and the rebuttal, as evidenced by the initial response and the followup.
> > >
> > > > "Layer-wise treatment is a strength": this claim is not convincing...
> > >
> > > We understand the reviewer's concern. However, the optimization community widely recognizes that Muon and related methods discussed in our paper (e.g., Scion, Gluon [1,2]) are among the most promising current optimizers. These algorithms are applied in a *layer-wise* manner, which is a key design principle. We refer the reviewer to the original references for details.
> > > Consequently, each of these variants naturally supports per-layer tuning. There exist lines of work, such as [2], that aim to determine theoretically optimal layer-wise stepsizes.
> > >
> > > > While the paper states (and I agree) that always updating all layers may be sub-optimal...
> > >
> > > This is not true. Let us address each point one by one.
> > >
> > > > They do not show that partial updates are better or even convergent
> > >
> > > Yes, Theorems 4.3 and E.2 do not show that Drop-Muon is convergent, because it is not what these theorems are about. **Convergence results are clearly established in Theorems 4.1, 4.2, 4.4**.
> > >
> > > Now, before we move on, let us restate the non-optimality results. As an example, consider Theorems 4.3:
> > > *The cost (8) is minimized by $(p_1, p_2, \ldots, p_b)=(1, 0, \ldots, 0)$ **if and only if** $L_{1,\{1,\ldots,b\}}=\max_{i\in[b]} L_{i,\{1,\ldots,b\}}$*.
> > >
> > > > An analogy: suppose there are three possible investment strategies...
> > >
> > > No, this is not a correct analogy. Theorems 4.3 and E.2 do not claim that the algorithm with full sampling is suboptimal among, say, all first-order methods. Rather, they show that full sampling is not optimal *within* Drop-Muon framework (except under an unlikely condition on the layer-wise smoothness constants). In other words, among all possible sampling strategies that Drop-Muon could employ, full updates are not the best choice. This rigorously and directly addresses the reviewer's concern. We hope this clarifies that there is no logical gap.
> > >
> > > > What is missing is a result that shows: Layer-wise randomization can outperform full-layer updating under certain conditions. No such conclusion is provided.
> > >
> > > Yes, it is. This is exactly what the theorems say, as explained above.
> > >
> > > > Overall, the experimental section is relatively weak and under-supported...
> > >
> > > As promised, we have added larger-scale experiments: training NanoGPT on the FineWeb dataset and an image-classification task with ResNet50 (Section 6). These results confirm the conclusions of the previous smaller experiments (now moved to the appendix) and clearly demonstrate that **Drop-Muon consistently outperforms Muon in wall-clock time**. The paper has been updated accordingly.
> > >
> > > [1] Thomas Pethick, Wanyun Xie, Kimon Antonakopoulos, Zhenyu Zhu, Antonio Silveti-Falls, and
> > > Volkan Cevher. Training deep learning models with norm-constrained LMOs. arXiv preprint
> > > arXiv:2502.07529, 2025.
> > >
> > > [2] Artem Riabinin, Egor Shulgin, Kaja Gruntkowska, and Peter Richt´ arik. Gluon: Making Muon &
> > > Scion great again! (Bridging theory and practice of LMO-based optimizers for LLMs). arXiv
> > > preprint arXiv:2505.13416, 2025b.
> > >
> > > ---
> > >
> > > We hope that these additional clarifications convinced the reviewer that our claims are fully supported and there are no gaps in our reasoning.
> > >
> > > Overall, we believe that all points raised in the review have been addressed. We hope that the significance of our contributions is now clear.

---

### Official Review · Reviewer_Su4K · 2025-10-31

**Soundness:** 3
**Presentation:** 2
**Contribution:** 3
**Rating:** 4
**Confidence:** 4

**Summary:**

This paper claims that optimizers don't need to update all network layers at every step, arguing this is computationally suboptimal for modern non-Euclidean optimizers like Muon. The authors propose Drop-Muon, a framework that instead updates a random subset of layers. Their core contribution is Randomized Progressive Training (RPT), a computationally efficient strategy that samples a "cutoff" layer and only updates layers from the cutoff layer to the end, which aligns with the backpropagation algorithm. The paper provides rigorous theoretical convergence guarantees for this method under generalized layer-wise $(L^0, L^1)$-smoothness and shows that full-network updates are theoretically inefficient. Empirically, Drop-Muon shows wall-clock time speedup over standard Muon on shallow CNNs trained on MNIST, Fashion-MNIST, and CIFAR-10.

**Strengths:**

1. The paper notices a simple but fundamental question about a new class of optimizers (Muon, Gluon, Scion). Questioning the "dense update" assumption is a valid and interesting line of inquiry.
2. The paper introduces progressive training into non-Euclidean optimizers, that's an interesting idea.
3. The paper provides the first convergence guarantees for a progressive, non-Euclidean, stochastic optimizer under generalized $(L^0, L^1)$-smoothness, which is a significant technical achievement.
4. The paper provides empirical results showing Drop-Muon can show speedup in wall-clock time.

**Weaknesses:**

1. The paper uses Randomized Progressive Training (RPT) method for frozen layer selection, but lacks discussion about the effect of RPT to performance. I understand that it's mainly for efficiency. However, not updating model evenly may lead to performance sacrifice. As shown in Appendix G.1, updating deeper layers with higher probability shows worse performance. More discussion is necessary.
2. There is a gap between the theoretical conclusion and experimental setups. Although the authors show optimal probability for Progressive Training, they just use random version in experiments.
3. My main concern is that we can't see any performance improvement (final accuracy) empirically and Drop-Muon is even worse than Muon in most cases. This is a bit opposite with theoretical conclusion -- subset-network updating can achieve optimal while full-network updating cannot.
4. Experimental setup is limited. Section 6 only shows results of 3-layer CNN on simple tasks. It doesn't show more realistic results for modern model architectures.
5. Experiments is limited in image tasks. Why not show results on language tasks as well? As the authors claimed in line 333, results on nanoGPT have shown that full-network updates is hard to achieve optimal. Why not test if Drop-Muon can help in this case?
6. Why do all experimental results report training accuracy rather than validation/test accuracy that is a more standard and promising metric?

**Questions:**

1. The paper only mentions Gluon when discussing about $(L^0,L^1)$-smoothness for Muon/Scion. I notice another previous paper name ClippedScion [1] also contributes on the same non-Euclidean $(L^0, L^1)$-smoothness framework. It's better to discuss this work as well.


[1] Pethick, Thomas, Wanyun Xie, Mete Erdogan, Kimon Antonakopoulos, Tony Silveti-Falls, and Volkan Cevher. "Generalized Gradient Norm Clipping & Non-Euclidean $(L_0, L_1)$-Smoothness." arXiv preprint arXiv:2506.01913 (2025).

---

> ### Author Response · Authors · 2025-11-19
>
> We thank the reviewer for their constructive feedback. We appreciate the recognition of the novelty and technical contributions of our work and we are happy that the reviewer found our ideas interesting. We now address the reviewer's concerns in detail.
>
> ---
>
> **Weaknesses:**
> > The paper uses Randomized Progressive Training (RPT) method for frozen layer selection, but lacks discussion about the effect of RPT to performance...
>
> We respectfully disagree with the statement that our paper "lacks discussion about the effect of RPT to performance". The entire paper is effectively a discussion of how partial layer updates affect both theoretical and practical performance. Performance can be measured in many ways. In our work, we focus on iteration complexity and update "cost" in the theoretical results, and iteration and wall-clock time in the empirical results. These metrics are widely accepted and directly translate into the efficiency of training.
>
> Regarding Appendix G.1, this section describes the training parameters and experimental setup. We outline several possible layer-sampling schemes that one might consider when designing experiments. We do not claim that *any* sampling scheme will yield improved performance.  It is indeed true that heavily biasing sampling toward deeper layers slows the algorithm down. Early layers play a crucial role, especially at the beginning of training, and therefore cannot simply be ignored. As an extreme example, suppose we assign probabilities $p_1=\ldots=p_{b-1}\approx 0$ and $p_b=1- \sum_{i=1}^{b-1} p_i \approx 1$, so that we almost always update only the last layer. One clearly should not expect fast convergence under such a sampling strategy.
> Our theory captures this behavior: consider Theorem 4.1. In this extreme case, we have $w_i = \sum_{s=1}^i \frac{p_s}{2 L_{i, \{s,\dots,b\}}^0} \approx 0$ for $i=1, \ldots, b-1$ and $w_b = \sum_{s=1}^b \frac{p_s}{2 L_{i, \{s,\dots,b\}}^0} \approx \frac{p_b}{2 L_{i, \{b\}}^0}$.
> Thus, the gradients for layers $i=1, \ldots, b-1$ receive weights close to zero. Ensuring that all gradient norms reach the desired accuracy therefore requires substantially more iterations compared with using a more balanced sampling distribution (as discussed in Appendix E.1.1).
> Overall, our paper argues that efficiency gains can be achieved by performing partial network updates when using our proposed randomization strategy, but *not* that arbitrary parameter choices will yield such improvements. Just as with the learning rate, the sampling probabilities are algorithmic parameters that must be selected in a reasonable manner. We provide guidance on how to choose them both in the paper and in the responses below.
>
> > There is a gap between the theoretical conclusion and experimental setups...
>
> It is true that achieving theoretically optimal performance with our method, just as with virtually any optimizer, requires access to certain problem-specific constants (in our case, smoothness or generalized smoothness parameters). We acknowledge that this requirement is not ideal. However, it is a common limitation in optimization theory, and all standard methods (e.g., SGD, Adam, or Muon) also suffer from it (which is why in practice these parameters are tuned).
>
> In deep learning, global smoothness/generalized smoothness constants are typically unknown. For this reason, in our experiments we adopt several simple, off-the-shelf layer-sampling strategies that require no knowledge of problem-specific constants. We do not view this as a weakness of our framework; rather, it is a strength. The fact that the algorithm can outperform the full-network baseline using the most basic uniform sampling demonstrates its robustness and practical viability, even without access to theoretically optimal parameters.

---

> > ### Author Response · Authors · 2025-11-19
> >
> > > My main concern is that we can't see any performance improvement (final accuracy) empirically and Drop-Muon is even worse than Muon in most cases...
> >
> > We would like to clarify a potential misunderstanding of our work. We never claim that Drop-Muon improves upon Muon in terms of the best achievable accuracy of the trained model. This is not our objective. **Our goal is to design an optimizer that improves upon Muon** (which is one of the most promising recently proposed methods) **in terms of total training *time*: achieving the same level of accuracy as Muon, but *faster*. This is precisely what our results demonstrate, both theoretically and empirically**:
> >
> > - In theory: We show that the cost function (8), where "cost" abstracts computational resources or runtime, is minimized by $(p_1, p_2, \ldots, p_b)=(1, 0, \ldots, 0)$ (i.e., full-network updates) if and only if a certain restrictive condition on the (generalized) smoothness constants is satisfied (Theorems 4.3 and E.2). When this condition does not hold, partial updates reduce the cost, motivating our randomization strategy.
> >
> > - In practice: Our experiments show that Drop-Muon outperforms Muon in terms of wall-clock time, as evidenced in Figures 1, 2, 3, 5, 6, 7, 9; Figure 8 shows comparable performance. The assertion that "Drop-Muon is even worse in most cases" is directly contradicted by the empirical evidence and is simply factually incorrect.
> >
> > Here, we would also like to highlight that our experiments were deliberately designed to *isolate the wall-clock benefits of partial layer updates*, without introducing confounding differences in other hyperparameters. For this reason, both Drop-Muon and Muon use the same learning rate. However, both our theoretical analysis and prior work on coordinate-descent–style methods indicate that Drop-Muon could use larger stepsizes than the full-update Muon baseline, potentially making it even faster. Even without exploiting this advantage, Drop-Muon already achieves superior wall-clock performance.
> >
> > It is true that Drop-Muon can appear slightly worse than Muon in terms of *iteration count* (see the left panels of Figures 1 and 3). This is expected: because Drop-Muon updates only a subset of the layers per iteration, it naturally makes less progress per iteration than a full-update method. However, iteration count is not the most informative metric for comparing the two algorithms. A single Drop-Muon iteration is cheaper than a Muon iteration. For this reason, our primary comparison is based on wall-clock time (right panels of Figures 1 and 3). Under this metric, Drop-Muon consistently outperforms Muon, as predicted by our theory. The key point is that the modest reduction in per-iteration progress is more than offset by the reduced per-iteration computational cost, leading to faster overall convergence (even with non-optimized parameter choices).
> >
> > > Experiments
> >
> > We appreciate the reviewer's suggestion. In this initial response, we focus on addressing all non-experimental comments in detail; additional comments on experiments will follow in subsequent rebuttal updates.
> >
> > We would like to briefly reiterate the main aim of our work. **This paper is primarily theoretical, and it should be judged on that basis The central contribution stands entirely independently of experiments.**: we identify a fundamental issue in deep model training and **rigorously prove that the predominant practice of full-network updates is not optimal** (Theorems 4.3 and E.2). This is the core of the paper. Even with no experiments, the theoretical result is strong, and the evaluation should be based on that, just as experimental papers are evaluated on experiments even when they contain no theory.
> >
> > That said, we did go a step further and provided empirical validation showing that our theory is reflected in practice on smaller but representative setups, demonstrating that our method is implementable and promising without requiring industrial-scale compute. Given the reviewer's interest, we are already running additional larger experiments and will share the results once they complete. However, we would like to reiterate that the paper is **fundamentally theoretical**, and it should be judged as such.

---

> > > ### Author Response · Authors · 2025-11-19
> > >
> > > **Questions:**
> > > > The paper only mentions Gluon when discussing about $(L^0,L^1)$-smoothness for Muon/Scion...
> > >
> > > We thank the reviewer for suggesting this additional reference. The difference between our approach and ClippedScion is that ClippedScion does not explicitly model the layer-wise structure in the smoothness assumption. Gluon is more closely related to our setting. Nevertheless, we will include ClippedScion in the discussion to acknowledge related work.
> > >
> > > ---
> > >
> > > We hope that the explanations above make our results and contributions clearer. We have addressed each concern in detail and showed that our theory, experiments, and conclusions are consistent and well supported. With these clarifications, we believe the strengths of Drop-Muon, both in theory and in practice, should be evident.
> > >
> > > We thank the reviewer once more for their helpful comments. Should any aspect of our work require further clarification, we are happy to elaborate. In light of the clarifications provided, we would greatly appreciate the reviewer's consideration in raising the score.

---

> > ### Comment · Reviewer_Su4K · 2025-11-28
> >
> > Thanks for the author's quick reply. However, my concerns regarding the trade-off between speed and final performance remain unaddressed.
> >
> > While Drop-Muon accelerates training in the early stages, Figure 8 (highlighted by the authors) reveals a concerning **performance ceiling**: the accuracy curve for Drop-Muon flattens out noticeably below that of the Muon baseline after reaching 90% on CIFAR-10. This raises a critical question: Can Drop-Muon reliably reach higher accuracy thresholds (e.g., 95% or 97%)? The current data in Figures 5, 7, and 8 suggest it cannot match the baseline's peak performance.
> >
> > Furthermore, the abstract explicitly claims a "shift in how **large-scale models** can be efficiently trained." Such a strong practical claim can't be supported solely by training accuracy on small datasets. Reporting validation/test accuracy is necessary.

---

> ### Author Response · Authors · 2025-12-03
>
> We thank the reviewer for their reply.
>
> As promised, we have added larger-scale experiments: training NanoGPT on the FineWeb dataset and an image-classification task with ResNet50 (Section 6). These results confirm the conclusions of the previous smaller experiments (now moved to the appendix) and clearly demonstrate that **Drop-Muon consistently outperforms Muon in wall-clock time**. The paper has been updated accordingly.
>
> Regarding the concern about a potential "performance ceiling", we would like to clarify that **this is not the case**. The figure mentioned by the reviewer (now Figure 11) appears to flatten toward the end not because the optimizer has reached a ceiling, but because training has completed sooner (both Drop-Muon and Muon are run for the same number of epochs). If training were continued, Drop-Muon would achieve the same accuracy as the full-network baseline.
>
> Overall, we believe that all points raised in the review have been addressed. We hope that the significance of our contributions is now clear.

---

### Official Review · Reviewer_45Me · 2025-11-01

**Soundness:** 3
**Presentation:** 3
**Contribution:** 4
**Rating:** 8
**Confidence:** 3

**Summary:**

This paper proposes a novel claim that updating only part of neural network results in better computation complexity compared to updating the entire model under most scenarios when optimizing a neural network with Muon (or in general LMO-type updates), and provides both theoretical and empirical justifications. Instead of considering only the iteration complexity of an optimizer, this paper focuses on optimizing the computation complexity by carefully analyzing the total computation cost, which is the expected computation cost per iteration times expected number of iteration for convergence. Moreover, this paper proposes Drop-Muon, a general framework that only updates a random sub-model using Muon (or LMO-type) optimizers. While the random sub-model can be sampled from any arbitrary distributions, this paper provides theoretical analysis of a certain distribution, randomized progressive training (RPT), which samples a random index $s^k\in [b]$ in iteration $k$ and only updates the $s^k,\ldots,b$-th layer (where $b$ denotes the total number of layers) while keeping the first $s^k-1$ layers frozen. Combining Drop-Muon with RPT, this paper provides convergence analysis of the resulting optimizer under both deterministic and stochastic loss scenarios under various smoothness assumptions. Specifically, the convergence rates match the optimal rate in corresponding settings. Notably, the convergence analysis and the computation cost optimization analysis together suggest that updating the entire model every step is only optimal when certain smoothness condition is met, which is unlikely in practice. Finally, this paper also provides experiment evidence to show that partially updating the model is indeed faster in computation compared to updating the entire model.

**Strengths:**

For me, the major strength of this paper lies in its novelty. It studies an practically important questions of how to accelerate training computation while maintaining the same performance, and it proposes one novel possible solution--partially updating neural networks. Moreover, this paper provides a systematic justification, both theoretically and empirically, to support this idea. For example, it provides theoretical analysis of Drop-Muon with RPT sampling under both deterministic and stochastic gradient settings and under different smoothness assumptions. It also provides alternative sampling distributions to choose submodels. Overall, I find the theoretical analysis of this paper very concrete. Furthermore, the empirical experiments also provide strong evidence that Drop-Muon is not only theoretically better but also outperforms full model update in practice.

**Weaknesses:**

- Although theoretically Drop-Muon achieves the same theoretical iteration complexity (i.e., dependence on total iteration $K$) as other optimizers without partial updates, I noticed from experiments (e.g. Figure 1 and 3) that the iteration complexity of Drop-Muon is worse than Muon in practice. This suggests a potential limitation of the tradeoff between iteration complexity (or data complexity) and computation cost. Specifically, in certain scenarios where the data size is the limiting constraint, Drop-Muon might have worse performance than full update Muon.
- The experiment section only includes rather simple tasks and dataset such as MNIST and fashion-MNIST. Experiments on more complicated tasks such as pre-training language models will be a stronger evidence to support the empirical performance of Drop-Muon. I personally think some relatively small models such as nano-GPT would be sufficient for this goal.
- A minor opinion that might improve the readability: could authors elaborate a bit further on the layerwise smoothness, especially since such constants are essential in both the convergence rate and the sampling distribution. For example, I think providing an example of layerwise smoothness set of a $b$-layer dense neural network will help me better understand it.

**Questions:**

- Could the authors elaborate how the activation of the first $s^k-1$ layers can be cached? If I understand correctly, each iterations samples a fresh data batch $\xi_k$, so the forward pass requires re-computing all activations from layer 1 to b. For example, in iteration $k-1$, the first layer computed activation $X^{(1)}\_{k-1}\xi\_{k-1}$, but it's required to compute $X^{(1)}_k\xi_k$ in iteration $k$ and $\xi\_{k-1}\ne \xi_k$.

---

> ### Author Response · Authors · 2025-11-19
>
> We would like to thank the reviewer for thorough and constructive feedback. We appreciate the recognition of the strengths and contributions of our work. We now address the comments and provide additional clarifications.
>
> ---
>
> **Weaknesses:**
> > I noticed from experiments (e.g. Figure 1 and 3) that the iteration complexity of Drop-Muon is worse than Muon in practice...
>
> We would like to highlight that our experiments were designed to isolate the wall-clock benefits of partial layer updates without introducing differences in other algorithmic components. For this reason, we used the same learning rate for both Drop-Muon and Muon. Nevertheless, both our theoretical analysis and the established practice of coordinate-descent–style methods suggest that Drop-Muon could safely use larger stepsizes than the full-network Muon baseline, potentially leading to faster convergence.
>
> That said, when both algorithms use identical learning rates, Drop-Muon can appear slightly worse than Muon in terms of *iteration count* (as seen in the left panels of Figures 1 and 3). This behavior is expected: because Drop-Muon updates only a subset of the network at each iteration, it naturally achieves less progress per step than the full-update Muon variant.
> However, iteration count alone is not the most fair metric for comparison. A Drop-Muon iteration can be substantially cheaper than a Muon iteration, which is why **our primary comparison is based on wall-clock time** (right panels of Figures 1 and 3). **Under this metric, Drop-Muon consistently outperforms Muon**. The key point is that the modest reduction in per-iteration progress is more than offset by the reduced per-iteration computational cost, resulting in faster overall convergence.
>
> > The experiment section only includes rather simple tasks and dataset such as MNIST and fashion-MNIST...
>
> We appreciate the reviewer's suggestion. In this initial response, we wanted to mainly focus on addressing all non-experimental comments in detail; additional comments on experiments will follow in subsequent rebuttal updates.
>
> Let us however briefly reiterate the main aim of our work.**This paper is primarily theoretical, and it should be judged on that basis. The central contribution stands entirely independently of experiments.**: we identify a fundamental issue in deep model training and **rigorously prove that the predominant practice of full-network updates is not optimal** (Theorems 4.3 and E.2). This is the core of the paper. Even with no experiments, the theoretical result is strong, and the evaluation should be based on that, just as experimental papers are evaluated on experiments even when they contain no theory.
>
> That said, we did go a step further and provided empirical validation showing that our theory is reflected in practice on smaller but representative setups, demonstrating that our method is implementable and promising. Given the reviewer's interest, we are already running additional larger experiments and will share the results once they complete. However, our paper is **fundamentally theoretical**, and it should be judged as such.

---

> > ### Author Response · Authors · 2025-11-19
> >
> > > could authors elaborate a bit further on the layerwise smoothness...
> >
> > We understand that, on first reading, the assumption may be difficult to grasp. In deep learning, it is generally challenging (or impossible) to compute global smoothness or generalized smoothness constants exactly, so closed-form values are typically unavailable. However, we can offer additional intuition.
> > Most analyses of gradient-based methods rely on standard (or generalized) smoothness assumptions. Our Assumptions 4.2 and 4.3 can be viewed as a more fine-grained versions of these classical conditions. In the special case $b=1$ (i.e., a single layer), they reduce to (generalized) smoothness. For multi-layer networks, our layer-wise assumptions capture the intuition that different subsets of layers may have different effective smoothness constants. This allows us to use tighter bounds on the local curvature of $f$, better reflecting the structure of the model.
> > Importantly, our Assumptions 4.2 and 4.3 are not more restrictive than standard (generalized) smoothness. Rather, they provide a richer parameterization by assigning separate constants to different subsets of layers, enabling a more precise analysis without narrowing the function class.
> >
> > Intuitively: while the full network admits certain global smoothness constants, it is natural to expect that individual layers (or smaller groups of layers) may admit tighter constants than those that apply to the model as a whole. This is precisely what our assumptions formalize, ultimately giving the improved bounds established in Theorems 4.1, 4.2, and 4.4.
> > As discussed in Section 4.1 and Appendices A and B, these assumptions are inspired by the coordinate descent literature. In fact, Assumption 4.2 reduces to block-wise Lipschitz continuity of the gradient when $\text{supp}(\mathcal{D}) = \{\{1\}, \{2\}, \ldots, \{b\}\}$.
> >
> > We appreciate the suggestion and will include an expanded discussion of these assumptions in the main text when allowed the additional page in the camera-ready version.
> >
> > **Questions:**
> > > Could the authors elaborate how the activation of the first $s^k-1$ layers can be cached? If I understand correctly, each iterations samples a fresh data batch $\xi_k$, so the forward pass requires re-computing all activations from layer $1$ to $b$...
> >
> > The cost model in Section 3 applies to Algorithm 2, which uses deterministic gradients. In this regime, the only source of randomness enters through the layer sampling. Thus, intermediate activations can be reused without recomputing them. In the stochastic setting, where fresh stochastic gradients are sampled at each iteration, the forward pass indeed changes with each batch. Therefore, the cost of the forward pass can be absorbed into $c_{text{ov}}$, since it remains the same regardless of which layers are sampled.
> >
> > ---
> >
> > We thank the reviewer once again for the thoughtful feedback and for recognizing the novelty and potential impact of our work. Should any additional questions arise, we are more than happy to provide further clarification.

---

### Meta-Review · Area_Chair_cEof · 2026-01-06

**Summary:**

The paper proposes Drop-Muon, a randomized progressive training variant of Muon that updates only subsets of layers per step to reduce compute cost. The core claim is that full-network updates are theoretically suboptimal under realistic layer-wise smoothness conditions, and that partial updates can achieve the same accuracy faster in wall-clock time. Reviewers agree the theory is non-trivial and novel. The main disagreements are about (i) whether the theory truly establishes an advantage for the proposed randomization, (ii) the realism and fragility of the cost model and layer-wise tuning assumptions, and (iii) whether the empirical evidence—especially regarding final accuracy and scale—is sufficient to support the practical claims.

**Reviewer Concerns:**

1. Does the theory really justify partial updates as better, not just full updates as suboptimal? Partially addressed.
The rebuttal clarifies that Theorems 4.3/E.2 establish optimality within the Drop-Muon sampling family and that convergence is handled separately. This resolves misunderstandings for some reviewers, but at least one reviewer remains unconvinced that the theory provides a strong positive guarantee rather than a relative non-optimality result.

2. Practical realism of assumptions (layer-wise smoothness, tuning, cost model) Partially addressed.
Authors argue these assumptions are standard in Muon-family theory and optional in practice, and defend the cost model as reasonable. This is coherent but does not fully satisfy concerns about usability, fragility, and interpretability raised by skeptical reviewers.

3. Empirical evidence and strength of practical claims. Largely addressed.
Initial concerns about small-scale CNN experiments were valid. The added NanoGPT and ResNet50 experiments significantly strengthen the empirical case for wall-clock speedups. However, some concerns remain about final accuracy ceilings, reliance on training accuracy, and the strength of claims about large-scale training.

**Reviewer Scores:**

Reviewer 45Me (8 > 8/9): Rebuttal addresses all technical and experimental questions; main concerns resolved.
Reviewer Su4K: (4 > Likely 5/6): Added large-scale experiments and clarification on objectives help, but lingering concern about final accuracy and strong claims likely remains.
Reviewer kUBn: (2 > 2): Core theoretical skepticism (logical gap, cost model, tuning practicality) remains despite extensive rebuttal. Reviewer explicitly states score stays.
Reviewer XX3C: (2 > 2)
Reviewer admits lack of background; rebuttal clarifies theory and adds experiments,

---

### Decision · Program_Chairs · 2026-01-26

Reject